# Genetic disruption of WASHC4 drives endo-lysosomal dysfunction and cognitive-movement impairments in mice and humans

Jamie L Courtland[1†], Tyler WA Bradshaw[1†], Greg Waitt[2], Erik J Soderblom[2,3], Tricia Ho[2], Anna Rajab[4], Ricardo Vancini[5], Il Hwan Kim[3,6*], Scott H Soderling[1,3*]

[1]Department of Neurobiology, Duke University School of Medicine, Durham, United States; [2]Proteomics and Metabolomics Shared Resource, Duke University School of Medicine, Durham, United States; [3]Department of Cell Biology, Duke University School of Medicine, Durham, United States; [4]Burjeel Hospital, VPS Healthcare, Muscat, Oman; [5]Department of Pathology, Duke University School of Medicine, Durham, United States; [6]Department of Anatomy and Neurobiology, University of Tennessee Heath Science Center, Memphis, United States

*For correspondence:
ikim9@uthsc.edu (IHK);
scott.soderling@duke.edu (SHS)

†These authors contributed equally to this work

Competing interests: The authors declare that no competing interests exist.

**Abstract** Mutation of the Wiskott–Aldrich syndrome protein and SCAR homology (WASH) complex subunit, SWIP, is implicated in human intellectual disability, but the cellular etiology of this association is unknown. We identify the neuronal WASH complex proteome, revealing a network of endosomal proteins. To uncover how dysfunction of endosomal SWIP leads to disease, we generate a mouse model of the human *WASHC4[c.3056C>G]* mutation. Quantitative spatial proteomics analysis of SWIP[P1019R] mouse brain reveals that this mutation destabilizes the WASH complex and uncovers significant perturbations in both endosomal and lysosomal pathways. Cellular and histological analyses confirm that SWIP[P1019R] results in endo-lysosomal disruption and uncover indicators of neurodegeneration. We find that SWIP[P1019R] not only impacts cognition, but also causes significant progressive motor deficits in mice. A retrospective analysis of SWIP[P1019R] patients reveals similar movement deficits in humans. Combined, these findings support the model that WASH complex destabilization, resulting from SWIP[P1019R], drives cognitive and motor impairments via endo-lysosomal dysfunction in the brain.

## Introduction

Neurons maintain precise control of their subcellular proteome using a sophisticated network of vesicular trafficking pathways that shuttle cargo throughout the cell. Endosomes function as a central hub in this vesicular relay system by coordinating protein sorting between multiple cellular compartments, including surface receptor endocytosis and recycling, as well as degradative shunting to the lysosome (*Chiu et al., 2017*; *Cullen and Steinberg, 2018*; *Raiborg et al., 2015*; *Simonetti et al., 2019*). How endosomal trafficking is modulated in neurons remains a vital area of research due to the unique degree of spatial segregation between organelles in neurons, and its strong implication in neurodevelopmental and neurodegenerative diseases (*Follett et al., 2014*; *Lane et al., 2012*; *Mukherjee et al., 2019*; *Poët et al., 2006*; *Zimprich et al., 2011*).

In non-neuronal cells, an evolutionarily conserved complex, the Wiskott–Aldrich syndrome protein and SCAR homology (WASH) complex, coordinates endosomal trafficking (*Derivery and Gautreau, 2010*; *Linardopoulou et al., 2007*). WASH is composed of five core protein components: WASHC1 (aka WASH1), WASHC2 (aka FAM21), WASHC3 (aka CCDC53), WASHC4 (aka SWIP), and WASHC5

**eLife digest** Cells in the brain need to regulate and transport the proteins and nutrients stored inside them. They do this by sorting and packaging the contents they want to move in compartments called endosomes, which then send these packages to other parts of the cell. If the components involved in endosome trafficking mutate, this can lead to 'traffic jams' where proteins pile up inside the cell and stop it from working normally.

In 2011, researchers found that children who had a mutation in the gene for WASHC4 – a protein involved in endosome trafficking – had trouble learning. However, it remained unclear how this mutation affects the role of WASCH4 and impacts the behavior of brain cells.

To answer this question, Courtland, Bradshaw et al. genetically engineered mice to carry an equivalent mutation to the one identified in humans. Experiments showed that the brain cells of the mutant mice had fewer WASHC4 proteins, and lower levels of other proteins involved in endosome trafficking. The mutant mice also had abnormally large endosomes in their brain cells and elevated levels of proteins that break down the cell's contents, resulting in a build-up of cellular debris. Together, these findings suggest that the mutation causes abnormal trafficking in brain cells.

Next, Courtland, Bradshaw et al. compared the behavior of adult and young mice with and without the mutation. Mice carrying the mutation were found to have learning difficulties and showed abnormal movements which became more exaggerated as they aged, similar to people with Parkinson's disease. With this result, Courtland, Bradshaw et al. reviewed the medical records of the patients with the mutation and discovered that these children also had problems with their movement.

These findings help explain what is happening inside brain cells when the gene for WASHC4 is mutated, and how disrupting endosome trafficking can lead to behavioral changes. Ultimately, understanding how learning and movement difficulties arise, on a molecular level, could lead to new therapeutic strategies to prevent, manage or treat them in the future.

(aka Strumpellin) (encoded by genes *Washc1-Washc5*, respectively), which are broadly expressed in multiple organ systems (*Alekhina et al., 2017*; *Kustermann et al., 2018*; *McNally et al., 2017*; *Simonetti and Cullen, 2019*; *Thul et al., 2017*). The WASH complex plays a central role in non-neuronal endosomal trafficking by activating Arp2/3-dependent actin branching at the outer surface of endosomes to influence cargo sorting and vesicular scission (*Gomez and Billadeau, 2009*; *Lee et al., 2016*; *Phillips-Krawczak et al., 2015*; *Piotrowski et al., 2013*; *Simonetti and Cullen, 2019*). WASH also interacts with at least three main cargo adaptor complexes – the Retromer, Retriever, and COMMD/CCDC22/CCDC93 (CCC) complexes – all of which associate with distinct sorting nexins to select specific cargo and enable their trafficking to other cellular locations (*Binda et al., 2019*; *Farfán et al., 2013*; *McNally et al., 2017*; *Phillips-Krawczak et al., 2015*; *Seaman and Freeman, 2014*; *Singla et al., 2019*). Loss of the WASH complex in non-neuronal cells has detrimental effects on endosomal structure and function, as its loss results in aberrant endosomal tubule elongation and cargo mislocalization (*Bartuzi et al., 2016*; *Derivery et al., 2009*; *Gomez et al., 2012*; *Gomez and Billadeau, 2009*; *Phillips-Krawczak et al., 2015*; *Piotrowski et al., 2013*). However, whether the WASH complex performs an endosomal trafficking role in neurons remains an open question, as no studies have addressed neuronal WASH function to date.

Consistent with the association between the endosomal trafficking system and pathology, dominant missense mutations in *WASHC5* (protein: Strumpellin) are associated with hereditary spastic paraplegia (SPG8) (*de Bot et al., 2013*; *Valdmanis et al., 2007*), and autosomal recessive point mutations in *WASHC4* (protein: SWIP) and *WASHC5* are associated with syndromic and non-syndromic intellectual disabilities (*Assoum et al., 2020*; *Elliott et al., 2013*; *Ropers et al., 2011*). In particular, an autosomal recessive mutation in *WASHC4* (c.3056C>G; p.Pro1019Arg) was identified in a cohort of children with non-syndromic intellectual disability (*Ropers et al., 2011*). Cell lines derived from these patients exhibited decreased abundance of WASH proteins, leading the authors to hypothesize that the observed cognitive deficits in SWIP[P1019R] patients resulted from disruption of neuronal WASH signaling (*Ropers et al., 2011*). However, whether this mutation leads to

perturbations in neuronal endosomal integrity, or how this might result in cellular changes associated with disease, are unknown.

Here we report the analysis of neuronal WASH and its molecular role in disease pathogenesis. We use in vivo proximity proteomics (iBioID) to uncover the neuronal WASH proteome and demonstrate that it is highly enriched for components of endosomal trafficking. We then generate a mouse model of the human *WASHC4*$^{c.3056c>g}$ mutation (SWIP$^{P1019R}$) (*Ropers et al., 2011*) to discover how this mutation may alter neuronal trafficking pathways and test whether it leads to phenotypes congruent with human patients. Using an adapted spatial proteomics approach (*Davies et al., 2018*; *Geladaki et al., 2019*; *Hirst et al., 2018*; *Shin et al., 2019*), coupled with a system-level analysis of protein covariation networks, we find strong evidence for substantial disruption of neuronal endosomal and lysosomal pathways in vivo. Cellular analyses confirm a significant impact on neuronal endo-lysosomal trafficking in vitro and in vivo, with evidence of lipofuscin accumulation and progressive apoptosis activation, molecular phenotypes that are indicative of neurodegenerative pathology. Behavioral analyses of SWIP$^{P1019R}$ mice at adolescence and adulthood confirm a role of WASH in cognitive processes and reveal profound, progressive motor dysfunction. Importantly, retrospective examination of SWIP$^{P1019R}$ patient data highlights parallel clinical phenotypes of motor dysfunction coincident with cognitive impairments in humans. Our results establish that loss of WASH complex function leads to alterations in the neuronal endo-lysosomal axis, which manifest behaviorally as cognitive and movement impairments in mice.

## Results

### Identification of the WASH complex proteome in vivo confirms a neuronal role in endosomal trafficking

While multiple mutations within the WASH complex have been identified in humans (*Assoum et al., 2020*; *Elliott et al., 2013*; *Ropers et al., 2011*; *Valdmanis et al., 2007*), how these mutations lead to neurological dysfunction remains unknown (*Figure 1A*). Given that previous work in non-neuronal cultured cells and non-mammalian organisms have established that the WASH complex functions in endosomal trafficking, we first aimed to determine whether this role was conserved in the mouse nervous system (*Alekhina et al., 2017*; *Jia et al., 2010*; *Derivery et al., 2009*; *Gomez et al., 2012*; *Gomez and Billadeau, 2009*). To discover the likely molecular functions of the neuronal WASH complex, we utilized an in vivo BioID (iBioID) paradigm developed in our laboratory to identify the WASH complex proteome from brain tissue (*Uezu et al., 2016*). BioID probes were generated by fusing a component of the WASH complex, WASH1 (gene: *Washc1*), with the promiscuous biotin ligase, BioID2 (WASH1-BioID2, *Figure 1B*), or by expressing BioID2 alone (negative control, soluble-BioID2) under the neuron-specific, human Synapsin-1 promoter (*Kim et al., 2016*). We injected adenoviruses (AAV) expressing these constructs into the cortex of wild-type postnatal day zero (P0) mice (*Figure 1B*). Two weeks post-injection, we administered daily subcutaneous biotin for 7 days to biotinylate in vivo substrates. The viruses displayed efficient expression and activity in brain tissue, as evidenced by colocalization of the WASH1-BioID2 viral epitope (HA) and biotinylated proteins (Streptavidin) (*Figure 1C–F*). For label-free quantitative LC-MS/MS analyses, whole-brain samples were collected at P22, snap frozen, and processed as previously described (*Uezu et al., 2016*). A total of 2102 proteins were identified across all three experimental replicates, which were further analyzed for those with significant enrichment in WASH1-BioID2 samples over solubleBioID2 negative controls (*Figure 1—figure supplement 1D*, *Supplementary file 1*).

The resulting neuronal WASH proteome included 175 proteins that were significantly enriched (fold-change≥4.0, Benjamini–Hochberg FDR<0.05, *Figure 1G*; *Benjamini and Hochberg, 1995*). Of these proteins, we identified all five WASH complex components (*Figure 1H*), as well as 13 previously reported WASH complex interactors (*Figure 1I*; *McNally et al., 2017*; *Phillips-Krawczak et al., 2015*; *Simonetti and Cullen, 2019*; *Singla et al., 2019*), which provided strong validity for our proteomic approach and analyses. Additional bioinformatic analyses of the neuronal WASH proteome identified a network of proteins implicated in vesicular trafficking, including 23 proteins enriched for endosomal functions (*Figure 1J*) and 24 proteins enriched for endocytic functions (*Figure 1K*). Among these endosomal and endocytic proteins were components of the recently identified endosomal sorting complexes, CCC (CCDC93 and COMMD9) and Retriever (VPS35L) (*Phillips-*

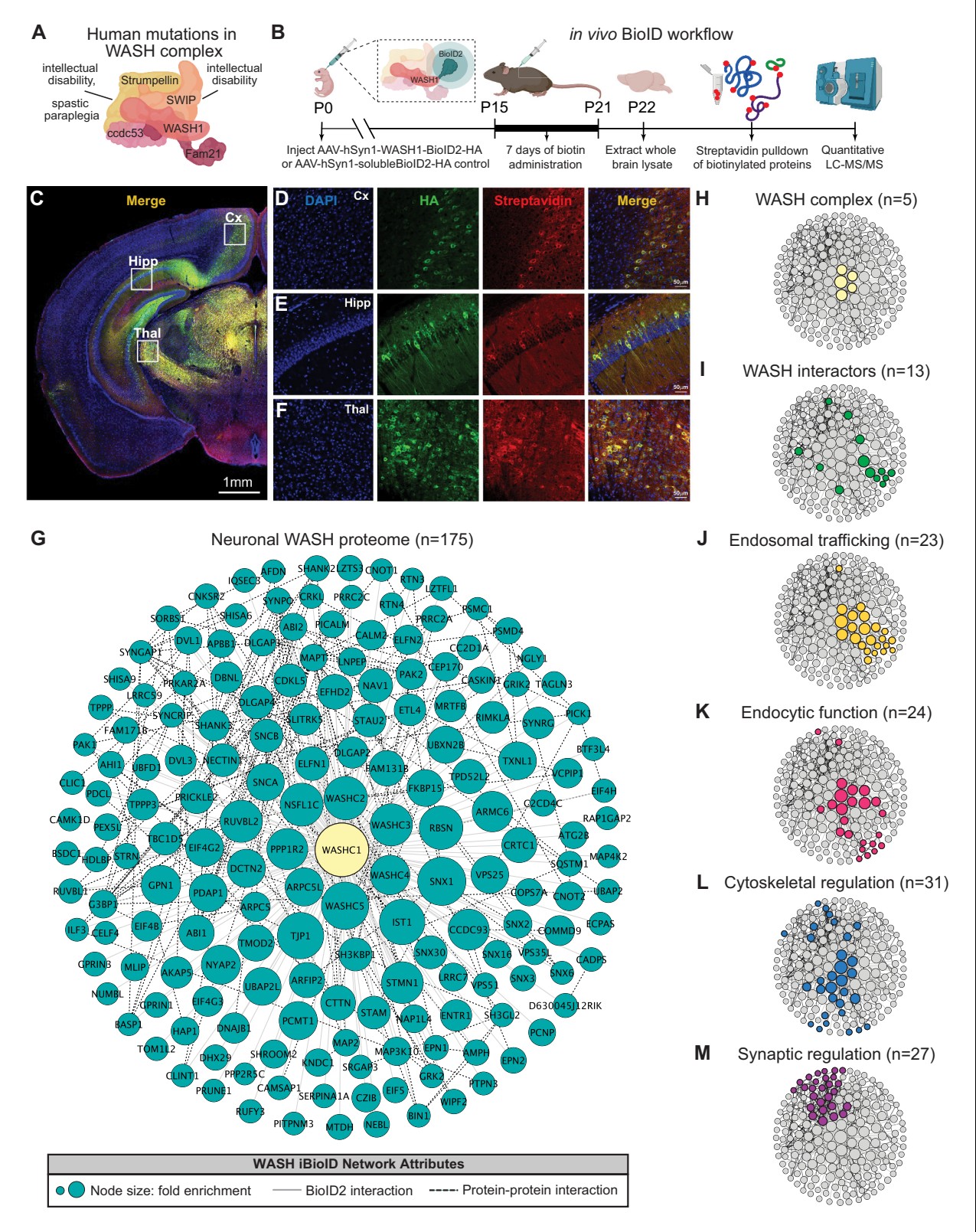

**Figure 1.** Identification of the WASH complex proteome in vivo. (**A**) The WASH complex is composed of five subunits: *Washc1* (WASH1), *Washc2* (FAM21), *Washc3* (CCDC53), *Washc4* (SWIP), and *Washc5* (Strumpellin). Human mutations in these components are associated with spastic paraplegia (*de Bot et al., 2013*; *Jahic et al., 2015*; *Valdmanis et al., 2007*), Ritscher–Schinzel syndrome (*Elliott et al., 2013*), and intellectual disability (*Assoum et al., 2020*; *Ropers et al., 2011*). (**B**) A BioID2 probe was attached to the c-terminus of WASH1 and expressed under the human synapsin-1

*Figure 1 continued on next page*

*Figure 1 continued*

(hSyn1) promoter in an AAV construct for in vivo BioID (iBioID). iBioID probes (WASH1-BioID2-HA or negative control solubleBioID2-HA) were injected into wild-type mouse brain at P0 and allowed to express for 2 weeks. Subcutaneous biotin injections (24 mg/kg) were administered over 7 days for biotinylation, and then brains were harvested for isolation and purification of biotinylated proteins. LC–MS/MS identified proteins significantly enriched in all three replicates of WASH1-BioID2 samples over soluble-BioID2 controls. (C) Representative image of WASH1-BioID2-HA expression in a mouse coronal brain section (Cx=cortex, Hipp=hippocampus, Thal=thalamus). Scale bar, 1 mm. (D) Representative image of WASH1-BioID2-HA expression in mouse cortex (inset from C). Individual panels show nuclei (DAPI, blue), AAV construct HA epitope (green), and biotinylated proteins (Streptavidin, red). Merged image shows colocalization of HA and Streptavidin (yellow). Scale bar, 50 μm. (E) Representative image of WASH1-BioID2-HA expression in mouse hippocampus (inset from C). Scale bar, 50 μm. (F) Representative image of WASH1-BioID2-HA expression in mouse thalamus (inset from C). Scale bar, 50 μm. (G) iBioID identified 175 proteins in the WASH interactome (fold-enrichment>4; FDR<0.05). Node size represents protein abundance fold-enrichment over negative control (range: 4–7.5), solid gray edges delineate iBioID interactions between the WASHC1 probe (seen in yellow at the center) and identified proteins, and dashed edges indicate known protein–protein interactions from HitPredict database (*López et al., 2015*). (H,I) Clustergrams of (H) all five WASH complex proteins identified by iBioID. (I) Previously reported WASH interactors (13/175), including the CCC and Retriever complexes. (J) Endosomal trafficking proteins (23/175 proteins). (K) Endocytic proteins (24/175). (L) Proteins involved in cytoskeletal regulation (31/175), including Arp2/3 subunit, ARPC5. (M) Synaptic proteins (27/175). Clustergrams were annotated by hand and cross-referenced with Metascape GO enrichment (*Zhou et al., 2019*) of WASH1 proteome constituents over all proteins identified in the BioID experiment.

The online version of this article includes the following figure supplement(s) for figure 1:

**Figure supplement 1.** In vivo BioID data normalization and analysis parameters.

*Krawczak et al., 2015*; *Singla et al., 2019*), as well as multiple sorting nexins important for recruitment of trafficking regulators to the endosome and cargo selection, such as SNX1-3 and SNX16 (*Kvainickas et al., 2017*; *Maruzs et al., 2015*; *Shin et al., 2019*; *Simonetti et al., 2017*). These data demonstrated that the WASH complex interacts with many of the same proteins in neurons as it does in yeast, amoebae, flies, and mammalian cell lines. Furthermore, there were 31 proteins enriched for cytoskeletal regulatory functions (*Figure 1L*), including actin-modulatory molecules such as the Arp2/3 complex subunit ARPC5, which is consistent with WASH's role in activating this complex to stimulate actin polymerization at endosomes for vesicular scission (*Jia et al., 2010*; *Derivery et al., 2009*). The WASH1-BioID2 isolated complex also contained 27 proteins known to localize to the excitatory post-synapse (*Figure 1M*). This included many core synaptic scaffolding proteins, such as SHANK2-3 and DLGAP2-4 (*Chen et al., 2011*; *Mao et al., 2015*; *Monteiro and Feng, 2017*; *Wan et al., 2011*), as well as modulators of synaptic receptors such as SYNGAP1 and SHISA6 (*Barnett et al., 2006*; *Clement et al., 2012*; *Kim et al., 2003*; *Klaassen et al., 2016*), which was consistent with the idea that vesicular trafficking plays an important part in synaptic function and regulation. Taken together, these results support a major endosomal trafficking role of the WASH complex in mouse brain.

## SWIP[P1019R] does not incorporate into the WASH complex, reducing its stability and levels in vivo

To determine how disruption of the WASH complex may lead to disease, we generated a mouse model of a human missense mutation found in children with intellectual disability, *WASHC4[c.3056c>g]* (protein: SWIP[P1019R]) (*Ropers et al., 2011*). Due to the sequence homology of human and mouse *Washc4* genes, we were able to introduce the same point mutation in exon 29 of murine *Washc4* using CRISPR (*Derivery and Gautreau, 2010*; *Ropers et al., 2011*). This C>G point mutation results in a Proline>Arginine substitution at position 1019 of SWIP's amino acid sequence (*Figure 2A*), a region thought to be critical for its binding to the WASH component, Strumpellin (*Jia et al., 2010*; *Ropers et al., 2011*). Western blot analysis of brain lysate from adult homozygous SWIP[P1019R] mutant mice (referred to from here on as MUT mice) displayed significantly decreased abundance of two WASH complex members, Strumpellin and WASH1 (*Figure 2B*). These results phenocopied data from the human patients (*Ropers et al., 2011*) and suggested that the WASH complex is unstable in the presence of this SWIP point mutation in vivo. To test whether this mutation disrupted interactions between WASH complex subunits, we compared the ability of wild-type SWIP (WT) and SWIP[P1019R] (MUT) to co-immunoprecipitate with Strumpellin and WASH1 in HEK cells. Compared to WT, MUT SWIP co-immunoprecipitated significantly less Strumpellin and WASH1 (IP: 54.8% and 41.4% of WT SWIP, respectively), suggesting that the SWIP[P1019R] mutation hinders WASH complex formation (*Figure 2—figure supplement 1*). Together these data support the notion that

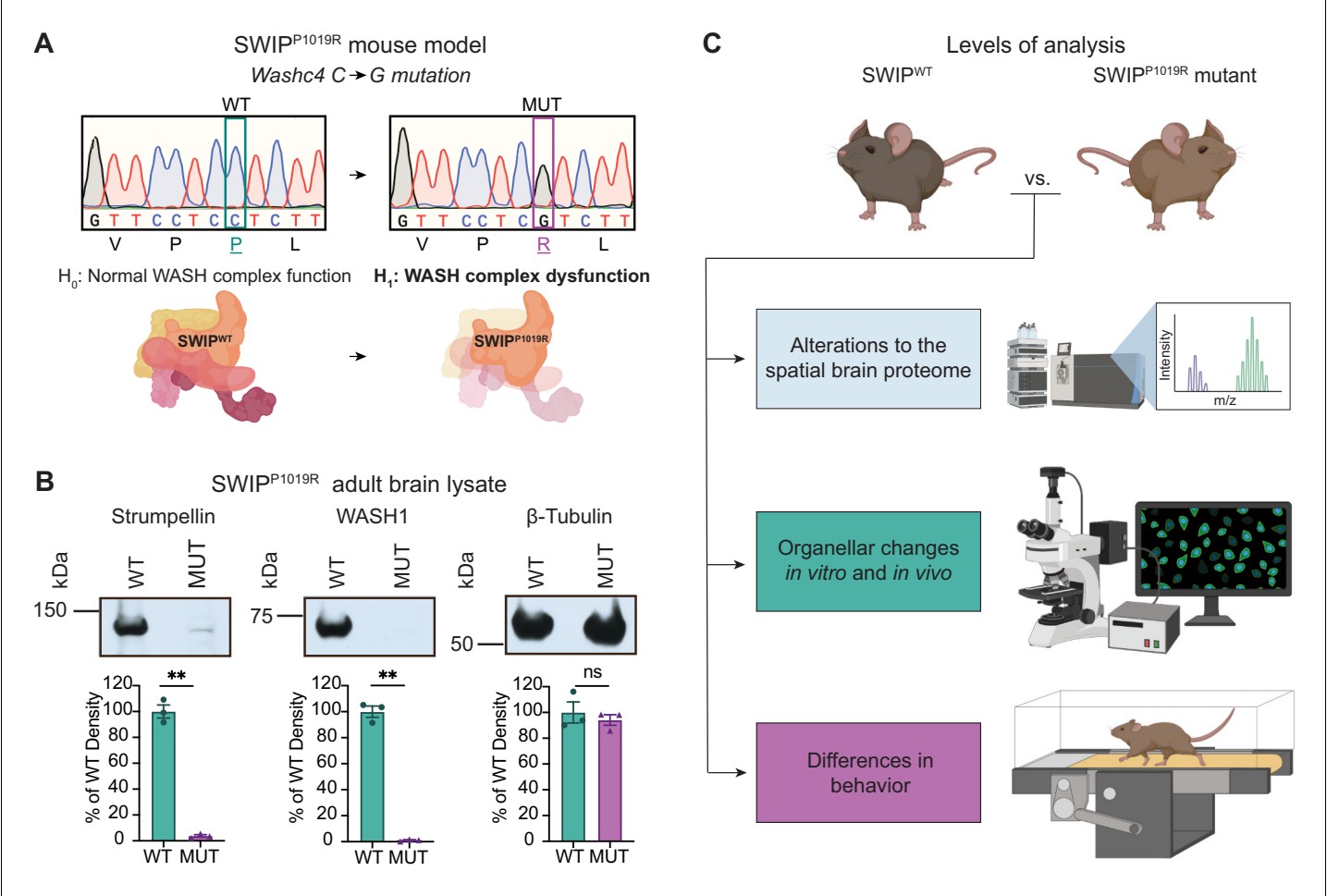

**Figure 2.** Mouse model of the human SWIP[P1019R] mutation displays decreases in WASH complex components. (**A**) Mouse model of the human SWIP[P1019R] missense mutation created using CRISPR. A C>G point mutation was introduced into exon29 of murine *Washc4*, leading to a P1019R amino acid substitution. We hypothesize (H[1]) that this mutation causes instability of the WASH complex. (**B**) Representative western blot and quantification of WASH components, Strumpellin and WASH1 (predicted sizes in kDa: 134 and 72, respectively), as well as loading control β-tubulin (55 kDa) from adult whole-brain lysate prepared from SWIP WT (*Washc4*[C/C]) and SWIP homozygous MUT (*Washc4*[G/G]) mice. Bar plots show quantification of band intensities normalized to β-tubulin, expressed as a % of WT (n=3 mice per genotype). Strumpellin (WT 100.0±5.1%, MUT 3.8±0.9%, $t_{2.14}$=18.60, p=0.0021) and WASH1 (WT 100.0±4.3%, MUT 1.1±0.4%, $t_{2.1}$=22.77, p=0.0018) were significantly decreased. Equivalent amounts of protein were analyzed in each condition (β-tubulin: WT 100.0±8.2%, MUT 94.1±4.1%, U=4, p>0.99). (**C**) Schematic of experimental techniques used to interrogate the effect of the SWIP[P1019R] mutation in subsequent figures: spatial proteomics (top), confocal and electron microscopy (middle), and mouse behavioral tasks (bottom).

The online version of this article includes the following figure supplement(s) for figure 2:

**Figure supplement 1.** Overexpression of SWIP[P1019R] decreases WASH complex binding in cultured cells.

SWIP[P1019R] is a damaging mutation that not only impairs its function, but also results in significant reductions of the WASH complex as a whole.

## Unbiased spatial proteomics analysis of SWIP[P1019R] mutant mouse brain reveals significant disruptions in endo-lysosomal pathways

Next, we aimed to understand the impact of the SWIP[P1019R] mutation on the subcellular organization of the mouse brain proteome using spatial proteomics. Conceptually, spatial proteomics encompasses a variety of methodological and analytical approaches, which share a common goal: predicting the subcellular localization of proteins. Most often this is done by combining subcellular fractionation of a biological sample with proteomic profiling of the resultant fractions (*Breckels et al., 2016*; *Crook et al., 2019*; *Crook et al., 2018*; *Geladaki et al., 2019*; *Itzhak et al.,*

*2017*; *Itzhak et al., 2016*; *Jean Beltran et al., 2016*). We performed spatial proteomics by subcellular fractionation, MS profiling, and subsequent clustering analysis. Clusters (modules) in the spatial proteomics network represent predicted subcellular compartments composed of proteins whose abundance covaries together in subcellular space (*Geladaki et al., 2019*; *Mulvey et al., 2017*). We analyzed differential abundance of individual proteins, as well as of protein groups (modules) identified in the spatial proteomics network to evaluate how the pathogenic SWIP$^{P1019R}$ mutation may perturb the organization of the neuronal subcellular proteome. This approach enabled us to study protein changes at a network level, which provided more biologically relevant insight than would be possible by assessing only protein-level differences.

Brains from 10-month-old mice were gently homogenized to release intact organelles, followed by successive centrifugation steps to enrich subcellular compartments into different biological fractions (BioFractions) based on their density (*Figure 3A*; *Geladaki et al., 2019*). Seven WT and seven MUT fractions (each prepared from one brain, 14 samples total) were labeled with unique isobaric tandem-mass tags and concatenated. We also included two sample pooled quality controls (SPQCs), which allowed us to assess experimental variability and perform normalization between experiments. By performing this experiment in triplicate, deep coverage of the mouse brain proteome was obtained – across all 48 samples we quantified 86,551 peptides, corresponding to 7488 proteins. After data pre-processing, normalization, and filtering, we retained 6919 reproducibly quantified proteins in the final dataset (*Supplementary file 2*).

We used MSstatsTMT to assess differential protein abundance for intra-fraction comparisons between WT and MUT genotypes and for overall comparisons between WT and MUT groups across all BioFractions (*Figure 3—figure supplement 4F*; *Huang et al., 2020*). In the first analysis, there were 65 proteins with significantly altered abundance in at least one of the seven subcellular fractions (Benjamini–Hochberg FDR<0.05, *Supplementary file 2*). Five proteins were differentially abundant between WT and MUT in all seven fractions, including four WASH proteins (WASHC1, WASHC2, WASHC4, WASHC5) and RAB21A – a known WASH interactor that functions in early endosomal trafficking ( *Figure 3D*; *Del Olmo et al., 2019*; *Simpson et al., 2004*). The abundance of the remaining WASH complex protein, WASHC3, was found to be very low and was only significantly reduced in BioFraction 10 (F10) and the overall ('Mutant-Control') comparison. These data affirm that the SWIP$^{P1019R}$ mutation destabilizes the WASH complex. Next, to evaluate global differences between WT and MUT brain, we analyzed the average effect of genotype on protein abundance across all fractions using MSstatsTMT (*Huang et al., 2020*). At this level, there were 728 differentially abundant proteins between WT and MUT brain (Benjamini–Hochberg FDR<0.05) (*Supplementary file 2*). We then aimed to place these differentially abundant proteins into a more meaningful biological context using a spatial proteomics approach.

Network-level analyses of spatial proteomic datasets can generally be performed in one of two ways: a top-down approach where proteins are grouped into organellar compartments learned from a predefined set of marker proteins, or a bottom-up approach where proteins are first clustered together based on covariation across biological fractions, and then analyzed for organellar enrichment (*Breckels et al., 2016*; *Crook et al., 2019*; *Crook et al., 2018*; *Itzhak et al., 2019*; *Itzhak et al., 2017*; *Jean Beltran et al., 2016*; *Orre et al., 2019*). For our network-based analyses, we chose to use a bottom-up approach, where we clustered the protein covariation network defined by the pairwise Pearson correlations between all proteins (*Freedman et al., 2007*). Our data-driven, quality-based approach used Network Enhancement (*Wang et al., 2018*) to remove biological noise from the covariation network and optimized partitions of the graph by maximizing the Surprise quality statistic (*Aldecoa and Marín, 2013*; *Traag et al., 2015*). Clustering of the protein covariation graph identified 49 modules of proteins that strongly covaried together (see Materials and methods for complete description of clustering approach).

To test for module-level differences between WT and MUT brain, we extended the LMM framework provided by MSstatsTMT to perform statistical inference at the level of protein groups (*Huang et al., 2020*). To identify systematic differences in the abundance of protein groups (modules), we fit the protein-level data for each module with a linear mixed-model expressing the mixed effect term, Protein, representing variation among a module's constituent proteins. We then performed a contrast of condition means given the fitted model, as described by *Huang et al., 2020*. Twenty-three of the 49 modules exhibited significant differences in WT versus MUT brain (Bonferroni p-adjust < 0.05; *Supplementary file 3*; *Benjamini and Hochberg, 1995*; *Hochberg, 1988*). Of note,

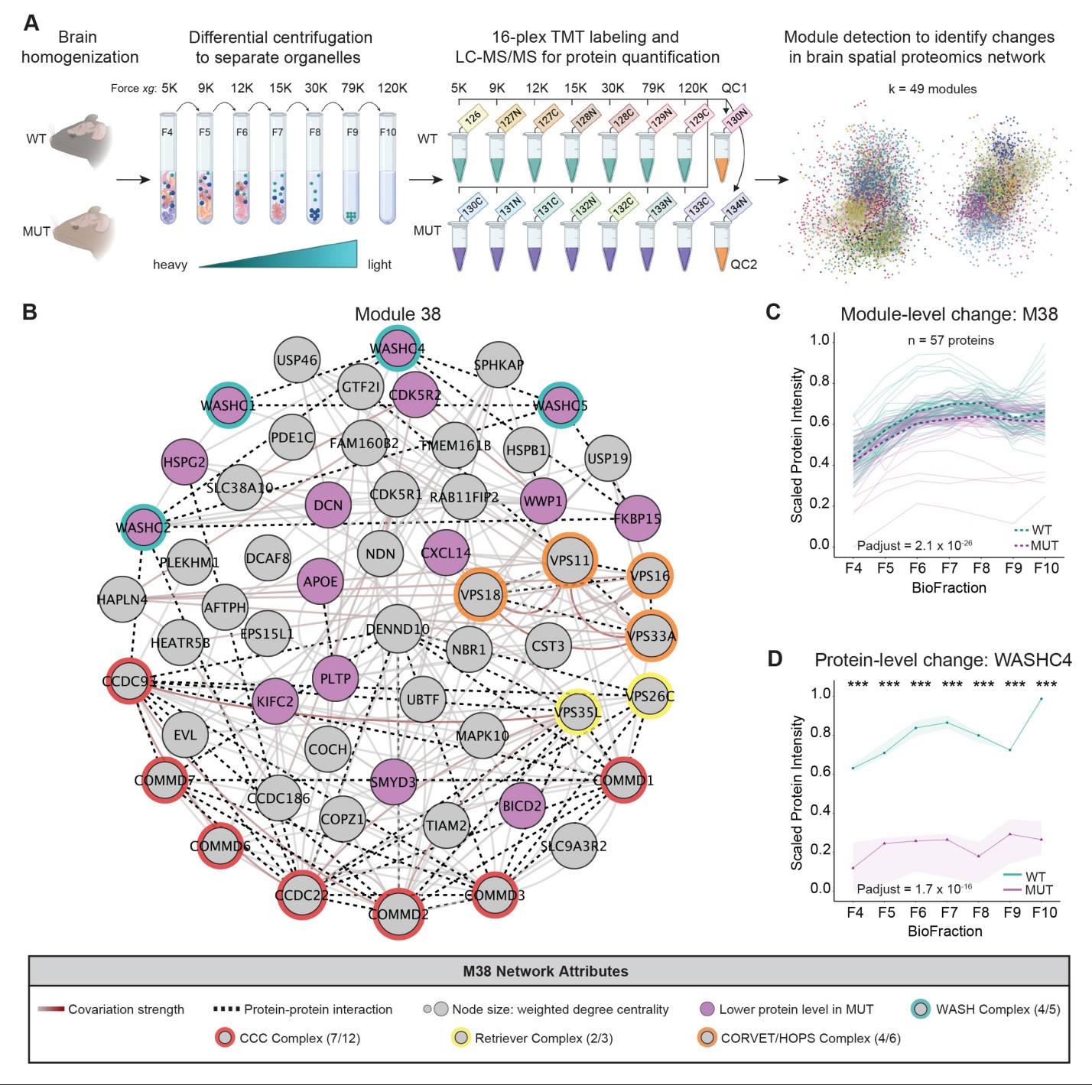

**Figure 3.** Spatial proteomics and network covariation analysis reveal significant disruptions to the WASH complex and an endosomal module in SWIP$^{P1019R}$ mutant mouse brain. (**A**) Tandem-mass-tag (TMT) spatial proteomics experimental design. Seven subcellular fractions were prepared from one WT and one MUT mouse (10mo). These samples, as well as two pooled quality control (QC) samples, were labeled with unique TMT tags and concatenated for simultaneous 16-plex LC–MS/MS analysis. This experiment was repeated three times (three WT and three MUT brains total). To detect network-level changes, proteins were clustered into modules, and linear mixed models (LMMs) were used to identify differences in module abundance between WT and MUT conditions. The network shows an overview of the spatial proteomics graph in which 49 different modules are indicated by colored nodes. (**B**) Protein module 38 (M38) contains subunits of the WASH, CCC, Retriever, and CORVET/HOPS complexes. Node size denotes its weighted degree centrality (~importance in module); purple node color indicates proteins with altered abundance in MUT brain relative to WT; red, yellow, orange, and green node borders highlight protein components of the CCC, Retriever, CORVET/HOPS, and WASH complexes obtained from the CORUM database; dashed black edges indicate experimentally determined protein-protein interactions; and gray-red edges denote the relative

*Figure 3 continued on next page*

*Figure 3 continued*

strength of protein covariation within a module (gray=weak, dark red=strong). (**C**) M38 displays decreased overall abundance in MUT brain. The aligned profiles of all M38 proteins are plotted together after sum normalization, and rescaling such that the maximum intensity is 1. Each solid line represents a single protein, measured in WT (teal) and MUT (purple) conditions. The estimated WT and MUT means are displayed in dashed teal and purple lines, respectively (WT-MUT Contrast log$_2$Fold-Change=−0.12, T=−11.14, DF=2324, p-adjust=2.078×10$^{-26}$; n=3 independent experiments). (**D**) Protein profile of WASHC4 (aka SWIP) plotted as relative (sum-normalized) protein intensity, rescaled to be in the range of 0–1 (WT-MUT Contrast log$_2$Fold-Change=−1.517, DF=26, p-adjust=1.72×10$^{-16}$; n=3 independent experiments). WT levels are depicted in teal, and MUT levels are depicted in purple. Shaded error bar represents the min-to-max values of all three experimental replicates. Significant differences in individual BioFraction WASHC4 levels are indicated with stars. ***p<0.001, MSstatsTMT p-value for intra-BioFraction comparisons with FDR correction.

The online version of this article includes the following figure supplement(s) for figure 3:

**Figure supplement 1.** Western blot analysis of LAMP1 abundance across subcellular brain fractions.
**Figure supplement 2.** Western blot analysis of EEA1 abundance across subcellular brain fractions.
**Figure supplement 3.** Western blot analysis of total protein abundance across subcellular brain fractions.
**Figure supplement 4.** Spatial proteomics experimental design and data analysis.
**Figure supplement 5.** Analysis of spatial brain proteome reveals conservation of organellar compartments found in LOPIT-DC dataset.

the module containing the WASH complex, M38, was predicted to have endosomal function by annotation of protein function (UniProt: 'Early Endosome', hypergeometric test p-adjust < 0.05, *Supplementary file 4*). Similar to the WASH iBioID proteome (*Figure 1*), M38 contained many endosomal proteins, including components of the CCC (CCDC22, CCDC93, COMMD1-3, and COMMD6-7) and Retriever sorting complexes (VPS26C and VPS35L) (*Figure 3B*). It also contained core subunits of the CORVET and HOPS vesicular tethering complexes, which enable fusion of vesicles within the endo-lysosomal system (VPS11, VPS16, VPS18, and VPS33A) (*van der Beek et al., 2019*). Across all fractions, the abundance of M38 was significantly lower in MUT brain compared to WT, providing evidence that the SWIP$^{P1019R}$ mutation reduces the stability of this protein subnetwork and impairs its function (*Figure 3C*).

We also observed another module, M36, that was enriched for lysosomal protein components (hypergeometric test p-adjust <0.05) (*Geladaki et al., 2019*) and contained all eight subunits of the exocyst complex (CORUM), a vesicular trafficking complex involved in lysosomal secretion (*Giurgiu et al., 2019*; *Sáez et al., 2019*). In contrast to the decreased abundance of the WASH complex/endosome module (M38), M36 exhibited increased abundance in MUT brain (*Figure 4C*). M36 (*Figure 4B*) contained several lysosomal cathepsin proteases (CTSA, CTSB, CTSS, and CTSL) as well as key lysosomal hydrolases (HEXA, GBA, GLB1, MAN2B1, and MAN2B2) (*Eng and Desnick, 1994*; *Mayor et al., 1993*; *Mok et al., 2003*; *Moon et al., 2016*; *Patel et al., 2018*; *Regier and Tifft, 1993*; *Rosenbaum et al., 2014*). Notably, M36 also contained the lysosomal glycoprotein progranulin (GRN), which is integral to proper lysosome function and whose loss is widely linked with neurodegenerative pathologies (*Baker et al., 2006*; *Pottier et al., 2016*; *Tanaka et al., 2017*; *Zhou et al., 2018*). The overall increase in abundance of module 36, and these key lysosomal proteins (*Figure 4C–E*), may therefore reflect an increase in flux through degradative lysosomal pathways in SWIP$^{P1019R}$ brain.

Besides these endo-lysosomal changes, module-level alterations were evident for an endoplasmic reticulum (ER) module (M6), supporting a shift in the proteostasis of mutant neurons (*Figure 4—figure supplement 1C–D*). Notably, within the ER module, M6, there was increased abundance of chaperones (e.g. HSPA5, PDIA3, PDIA4, PDIA6, and DNAJC3) that are commonly engaged in presence of misfolded proteins (*Bartels et al., 2019*; *Kim et al., 2020*; *Montibeller and de Belleroche, 2018*; *Synofzik et al., 2014*; *Wang et al., 2016*). This elevation of ER stress modulators can be indicative of neurodegenerative states, in which the unfolded protein response (UPR) is activated to resolve misfolded species (*Garcia-Huerta et al., 2016*; *Hetz and Saxena, 2017*). These data demonstrate that loss of WASH function not only alters endo-lysosomal trafficking, but also causes increased stress on cellular homeostasis.

In addition, we also observed a synaptic module (M27) that was reduced in MUT brain (*Figure 4—figure supplement 1E–F*). This module included core excitatory post-synaptic proteins such as SHANK2 and DLGAP4 (also identified in WASH1-BioID, *Figure 1*), consistent with endosomal WASH influencing synaptic regulation. Decreased abundance of these modules indicates that loss of the WASH complex may result in failure of these proteins to be properly trafficked to the synapse. In

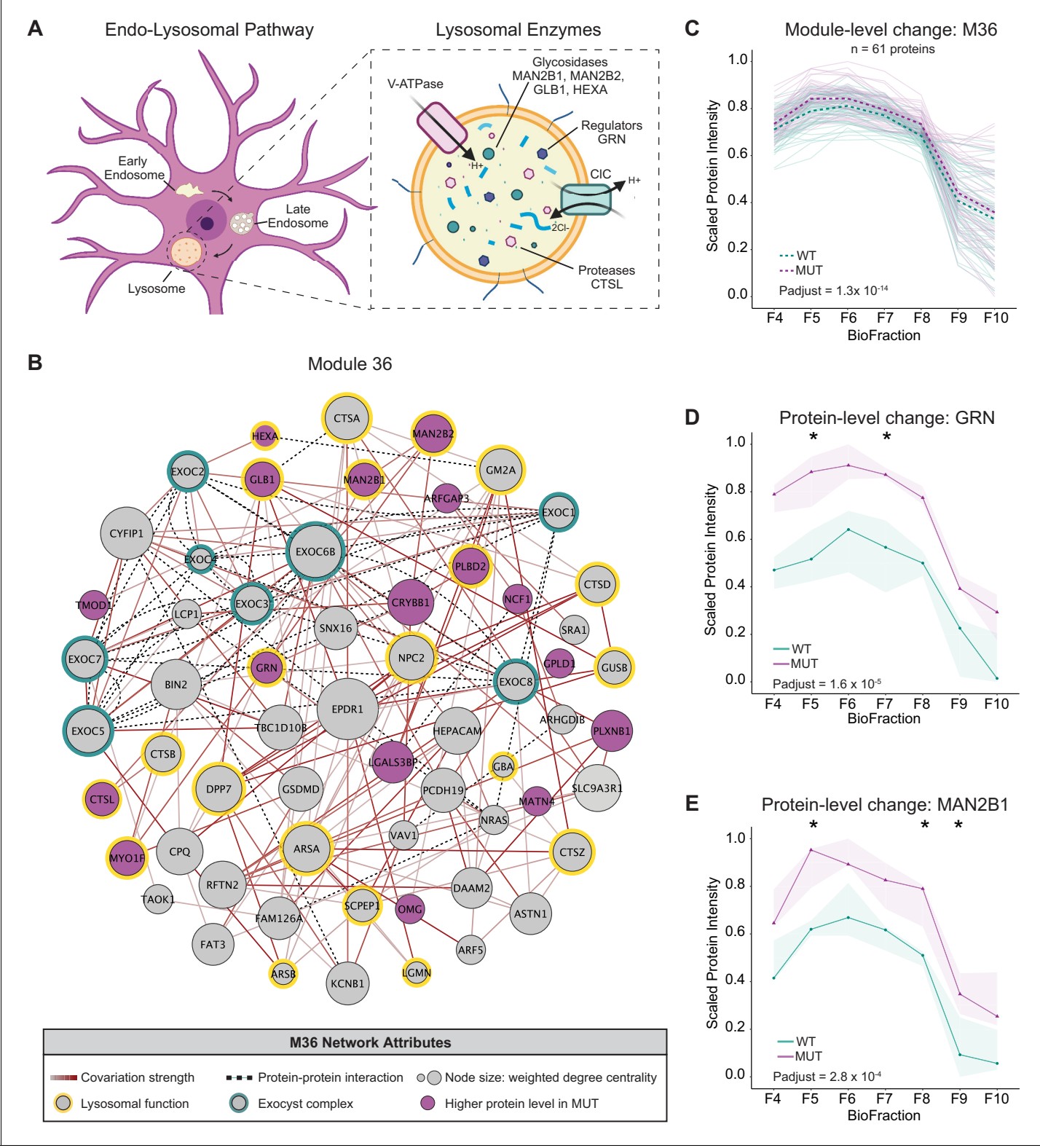

**Figure 4.** Disruption of a lysosomal protein network in SWIP[P1019R] mutant brain. (**A**) Simplified schematic of the endo-lysosomal pathway in neurons. Inset depicts representative lysosomal enzymes, such as proteases (CTSL), glycosidases (MAN2B1, MAN2B2, GLB1, HEXA), and key lysosomal regulators (GRN). (**B**) Network graph of module 36 (M36). M36 proteins that exhibit altered abundance in MUT brain include lysosomal proteins, HEXA, GLB1, MAN2B1, MAN2B2, GRN, and CTSL. Network attributes: Node size denotes its weighted degree centrality (~importance in module), node color indicates proteins with altered abundance in MUT brain relative to WT, yellow outlines highlight proteins identified as lysosomal in *Geladaki et al.,*

*Figure 4 continued on next page*

*Figure 4 continued*

*2019*, green outlines indicate members of the exocyst complex (CORUM), dashed black edges indicate experimentally determined protein-protein interactions (HitPredict), and gray-red edges denote the relative strength of protein covariation within a module (gray=weak, dark red=strong). (C) The scaled protein intensity of all M36 proteins plotted together as a module. Overall, there is a significant increase in the estimated mean of M36 relative to WT (WT-MUT log$_2$Fold-Change=0.086, T=8.25, DF=2488, p-adjust=1.29×10$^{-14}$; n=3 independent experiments). Light teal and purple lines denote scaled protein profiles for individual WT and MUT proteins, respectively. Dashed green and purple lines indicate mean scaled protein intensities for WT and MUT, respectively. (D) Protein profile of lysosomal protein progranulin, GRN (WT-MUT Contrast log$_2$Fold-Change=0.637, DF=28, p-adjust=1.58×10$^{-5}$; n=3 independent experiments). (E) Protein profile of lysosomal enzyme, MAN2B1 (WT-MUT Contrast log$_2$Fold-Change=0.319, DF=26, p-adjust=2.75×10$^{-4}$; n=3 independent experiments). For (D) and (E), WT levels are depicted in teal, and MUT levels are depicted in purple. Shaded error bar represents the min-to-max values of all three experimental replicates. Significant differences in individual BioFraction levels are indicated with stars. *p<0.05, MSstatsTMT p-value for intra-BioFraction comparisons with FDR correction (D,E).

The online version of this article includes the following figure supplement(s) for figure 4:

**Figure supplement 1.** Multiple protein networks display significant alterations in MUT brain compared to WT.
**Figure supplement 2.** Aged SWIP$^{P1019R}$ mutant mice display decreased excitatory synapse number in the motor cortex.

line with these findings, we observed fewer excitatory synapses in adult MUT brain compared to WT (*Figure 4—figure supplement 2*), validating that these module-level differences correlate with cellular alterations in vivo.

## Mutant neurons display structural abnormalities in endo-lysosomal compartments in vitro

Combined, the proteomics data strongly suggested that endo-lysosomal pathways are altered in adult SWIP$^{P1019R}$ mutant mouse brain. Next, we analyzed whether structural changes in this system were evident in primary neurons. Cortical neurons from littermate WT and MUT P0 pups were cultured for 15 days in vitro (DIV15, *Figure 5A*), then fixed and stained for established markers of early endosomes (early endosome antigen 1 [EEA1]; *Figure 5B and C*) and lysosomes (Cathepsin D [CathD]; *Figure 5D and E*). Reconstructed three-dimensional volumes of EEA1 and Cathepsin D puncta revealed that MUT neurons display larger EEA1$^+$ somatic puncta than WT neurons (*Figure 5G and J*), but no difference in the total number of EEA1$^+$ puncta (*Figure 5F*). This finding is consistent with a loss-of-function mutation, as loss of WASH activity prevents cargo scission from endosomes and leads to cargo accumulation (*Bartuzi et al., 2016*; *Gomez et al., 2012*). Conversely, MUT neurons exhibited significantly less Cathepsin D+ puncta than WT neurons (*Figure 5H*), but the remaining puncta were significantly larger than those of WT neurons (*Figure 5I and K*). These data support the finding that the SWIP$^{P1019R}$ mutation results in both molecular and morphological abnormalities in the endo-lysosomal pathway.

## SWIP$^{P1019R}$ mutant brains exhibit markers of abnormal endo-lysosomal structures and cell death in vivo

As there is strong evidence that dysfunctional endo-lysosomal trafficking and elevated ER stress are associated with neurodegenerative disorders, adolescent (P42) and adult (10 month old, 10mo) WT and MUT brain tissues were analyzed for the presence of cleaved caspase-3, a marker of apoptotic pathway activation, in four brain regions (*Boatright and Salvesen, 2003*; *Porter and Jänicke, 1999*). Very little cleaved caspase-3 staining was present in WT and MUT mice at adolescence (*Figure 6A and B*, and *Figure 6—figure supplement 1*). However, at 10mo, the MUT motor cortices displayed significantly greater cleaved caspsase-3 staining compared to age-matched WT littermate controls (*Figures 6D, E and H*). Furthermore, this difference appeared to be selective for the motor cortex, as we did not observe significant differences in cleaved caspase-3 staining at either age for hippocampal, striatal, or cerebellar regions (*Figure 6—figure supplement 1*). Consistent with these findings, there were no significant differences in dopaminergic cell number in the substantia nigra pars compacta or in dopaminergic innervation of the striatum in adult brain, suggesting that the motor cortex was the primary movement-related region altered in SWIP$^{P109R}$ brain (*Figure 6—figure supplement 2*). These data suggested that neurons of the motor cortex were particularly susceptible to disruption of endo-lysosomal pathways downstream of SWIP$^{P109R}$, perhaps because long-range corticospinal projections require high fidelity of trafficking pathways (*Blackstone et al., 2011*; *Slosarek et al., 2018*; *Wang et al., 2014*).

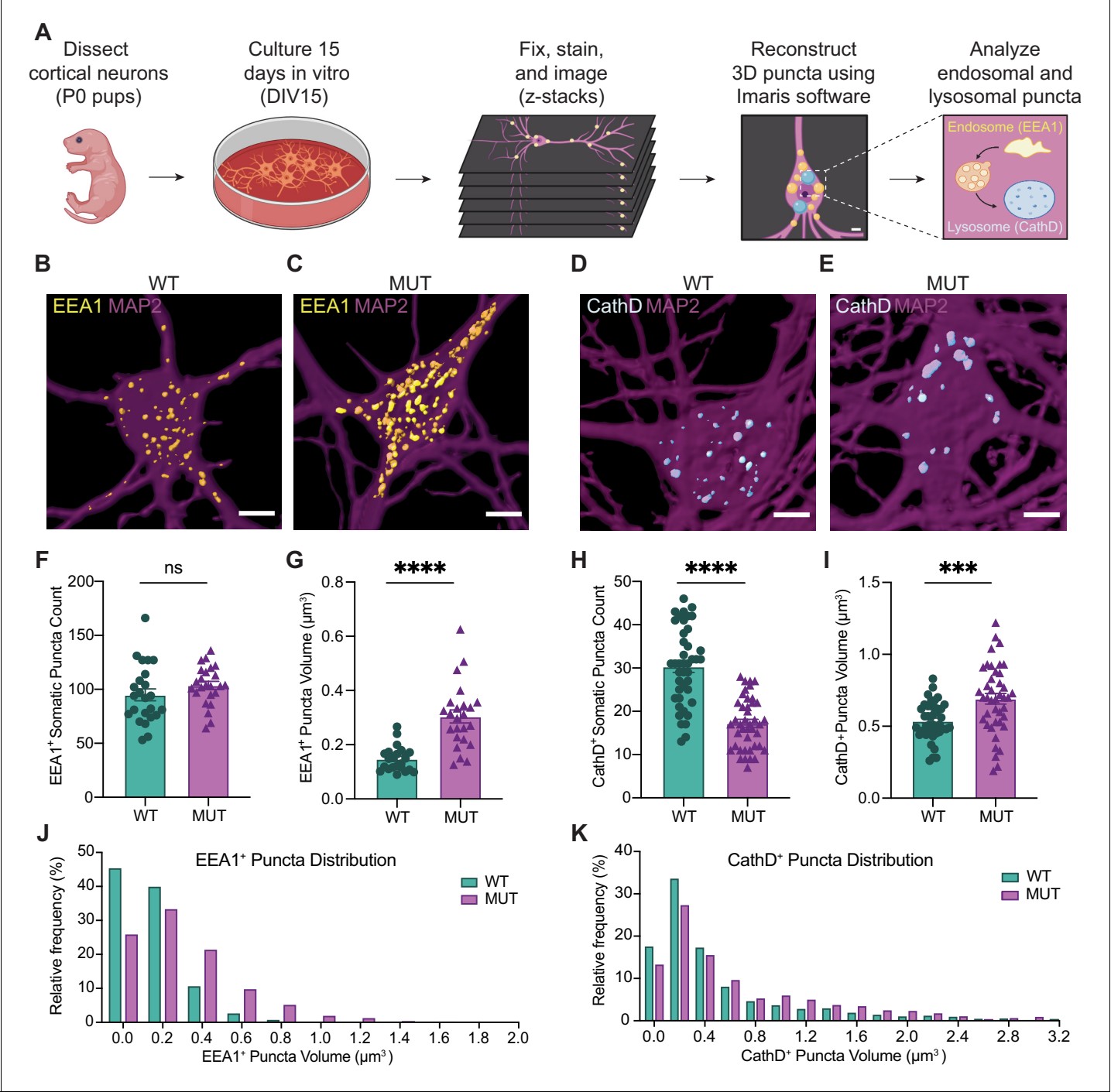

**Figure 5.** SWIP$^{P1019R}$ mutant neurons display structural abnormalities in endo-lysosomal compartments in vitro. (**A**) Experimental design. Cortices were dissected from P0 pups, and neurons were dissociated and cultured on glass coverslips for 15 days. Cultures were fixed, stained, and imaged using confocal microscopy. 3D puncta volumes were reconstructed from z-stack images using Imaris software. (**B,C**) Representative 3D reconstructions of WT and MUT DIV15 neurons (respectively) stained for EEA1 (yellow) and MAP2 (magenta). (**D,E**) Representative 3D reconstructions of WT and MUT DIV15 neurons (respectively) stained for Cathepsin D (cyan) and MAP2 (magenta). (**F**) Graph of the average number of EEA1$^+$ volumes per soma in each image (WT 95.0±5.5, n = 24 neurons; MUT 103.7±3.7, n = 24 neurons; U = 208.5, p=0.1024). (**G**) Graph of the average EEA1$^+$ volume size per soma shows larger EEA1$^+$ volumes in MUT neurons (WT 0.15±0.01 μm$^3$, n=24 neurons; MUT 0.30±0.02 μm$^3$, n=24 neurons; U=50, p<0.0001). (**H**) Graph of the average number of Cathepsin D+ volumes per soma illustrates less Cathepsin D+ volumes in MUT neurons (WT 30.4±1.4, n=42; MUT 17.2±0.9, n=42; U=204, p<0.0001). (**I**) Graph of the average Cathepsin D+ volume size per soma demonstrates larger Cathepsin D+ volumes in MUT neurons (WT 0.54±0.02 μm$^3$, n=42; MUT 0.69±0.04 μm$^3$, n=42; t$_{63}$=3.701, p=0.0005). (**J**) Histogram of EEA1$^+$ volumes illustrate differences in size distributions

*Figure 5 continued on next page*

*Figure 5 continued*

between MUT and WT neurons (D=0.2661, p<0.0001). (**K**) Histogram of CathD+ volumes show differences in size distributions between MUT and WT neurons (D=0.1307, p<0.0001). Analyses included at least three separate culture preparations. Scale bars, 5 µm (**B–E**). Data reported as mean ± SEM, error bars are SEM. ***p<0.001, ****p<0.0001, Mann–Whitney U test (**F–H**), two-tailed t-test (**I**), or Kolmogorov–Smirnov test (**J,K**).

To further examine the morphology of primary motor cortex neurons at a subcellular resolution, samples from age-matched 7-month-old WT and MUT mice (7mo, three animals each) were imaged by transmission electron microscopy (TEM). Strikingly, we observed large electron-dense inclusions in the cell bodies of MUT neurons (arrows, *Figure 6L*; pseudo-colored region, 6N). These dense structures were associated electron-lucent lipid-like inclusions (asterisk, *Figure 6N*), which supported the conclusion that these structures were lipofuscin accumulation at lysosomal residual bodies (*Poët et al., 2006*; *Valdez et al., 2017*; *Yoshikawa et al., 2002*). Lipofuscin is a by-product of lysosomal breakdown of lipids, proteins, and carbohydrates, which naturally accumulates over time in non-dividing cells such as neurons (*Höhn and Grune, 2013*; *Moreno-García et al., 2018*; *Terman and Brunk, 1998*). However, excessive lipofuscin accumulation is thought to be detrimental to cellular homeostasis by inhibiting lysosomal function and promoting oxidative stress, often leading to cell death (*Brunk and Terman, 2002*; *Powell et al., 2005*). As a result, elevated lipofuscin is considered a biomarker of neurodegenerative disorders, including Alzheimer's disease, Parkinson's disease, and neuronal ceroid lipofuscinoses (*Moreno-García et al., 2018*). Therefore, the marked increase in lipofuscin area and number seen in MUT electron micrographs (*Figure 6O and P*, respectively) is consistent with the increased abundance of lysosomal proteins observed by proteomics and likely reflects an increase in lysosomal breakdown of cellular material. Together these data indicate that SWIP^P1019R results in pathological lysosomal function that could lead to neurodegeneration.

## SWIP^P1019R mutant mice display persistent deficits in cued fear memory recall

To observe the functional consequences of the SWIP^P1019R mutation, we next studied WT and MUT mouse behavior. Given that children with homozygous SWIP^P1019R point mutations display intellectual disability (*Ropers et al., 2011*) and SWIP^P1019R mutant mice exhibit endo-lysosomal disruptions implicated in neurodegenerative processes, behavior was assessed at two ages: adolescence (P40–50), and mid-late adulthood (5.5–6.5 mo). Interestingly, MUT mice performed equivalently to WT mice in episodic and working memory paradigms, including novel object recognition and Y-maze alternations (*Figure 7—figure supplement 1*). However, in a fear conditioning task, MUT mice displayed a significant deficit in cued fear memory (*Figure 7*). This task tests the ability of a mouse to associate an aversive event (a mild electric footshock) with a paired tone (*Figure 7A*). Freezing behavior of mice during tone presentation is attributed to hippocampal or amygdala-based fear memory processes (*Goosens and Maren, 2001*; *Maren and Holt, 2000*; *Vazdarjanova and McGaugh, 1998*). Forty-eight hours after exposure to the paired tone and footshock, MUT mice showed a significant decrease in conditioned freezing to tone presentation compared to their WT littermates (*Figure 7B,C*). To ensure that this difference was not due to altered sensory capacities of MUT mice, we measured the startle response of mice to both electric foot shock and presented tones. In line with intact sensation, MUT mice responded comparably to WT mice in these tests (*Figure 7—figure supplement 2*). These data demonstrate that although MUT mice perceive footshock sensations and auditory cues, it is their memory of these paired events that is significantly impaired. Additionally, this deficit in fear response was evident at both adolescence and adulthood (top panels, and bottom panels, respectively, *Figure 7B and C*). These changes are consistent with the hypothesis that SWIP^P109R is the cause of cognitive impairments in humans.

## SWIP^P1019R mutant mice exhibit surprising motor deficits that are confirmed in human patients

Because SWIP^P1019R results in endo-lysosomal pathology consistent with neurodegenerative disorders in the motor cortex, we next analyzed motor function of the mice over time. First, we tested the ability of WT and MUT mice to remain on a rotating rod for 5 min (Rotarod, *Figure 8A–C*). At both adolescence and adulthood, MUT mice performed markedly worse than WT littermate controls

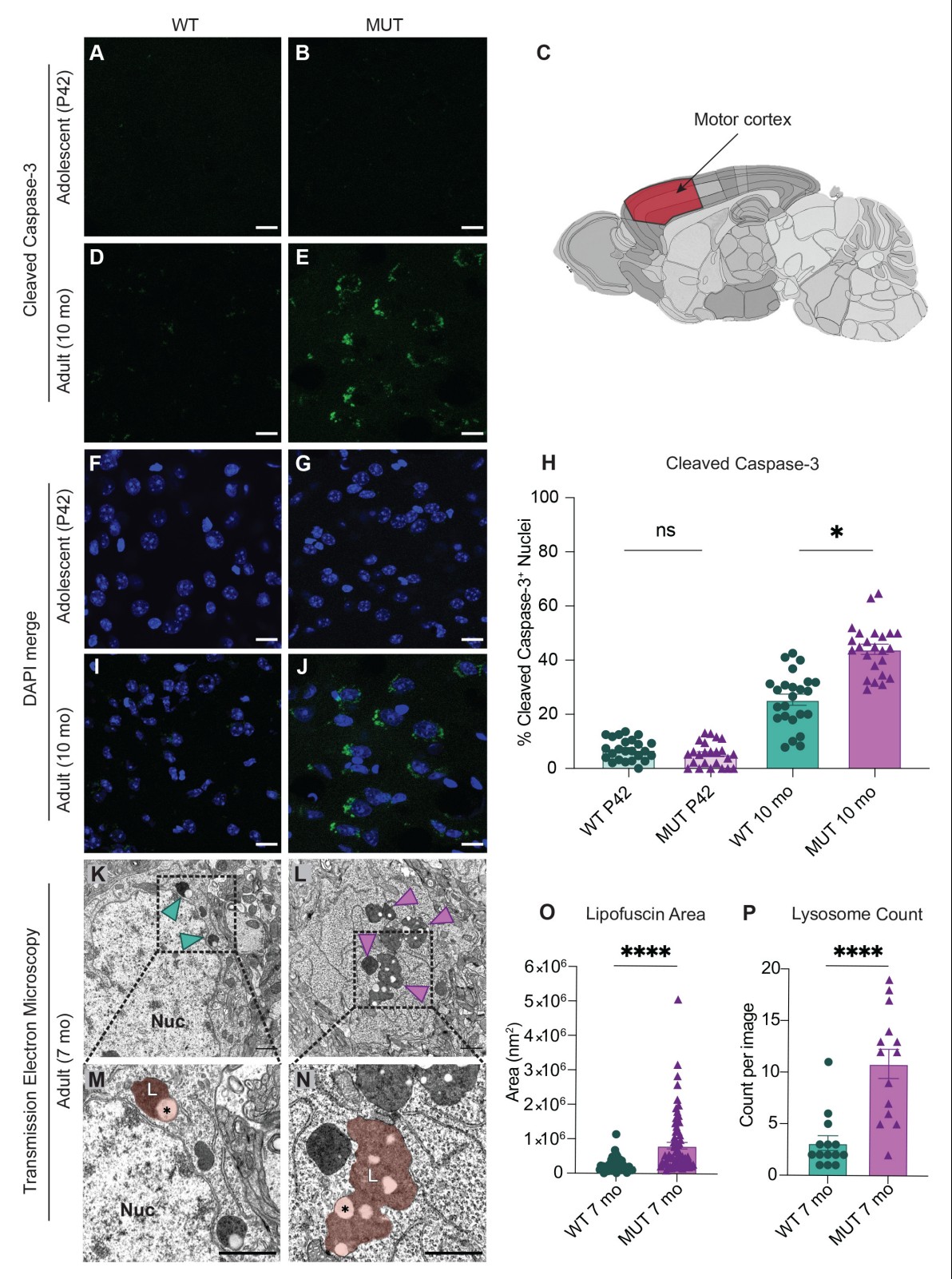

**Figure 6.** SWIP[P1019R] mutant brains exhibit markers of abnormal endo-lysosomal structures and cell death in vivo. (A,B) Representative images of adolescent (P42) WT and MUT motor cortex stained with cleaved caspase-3 (CC3, green). (C) Anatomical representation of mouse brain with motor cortex highlighted in red, adapted from the Allen Brain Atlas (*Oh et al., 2014*). (D,E) Representative image of adult (10 mo) WT and MUT motor cortex stained with CC3 (green). (F, G, I, and J) DAPI co-stained images for (A, B, D, and E, respectively). Scale bar for (A–J), 15 µm. (H) Graph depicting the

*Figure 6 continued on next page*

*Figure 6 continued*

normalized percentage of DAPI+ nuclei that are positive for CC3 per image. No difference is seen at P42, but the amount of CC3+ nuclei is significantly higher in aged MUT mice (P42 WT 6.97 ± 0.80%, P42 MUT 5.26 ± 0.90%, 10mo WT 25.38 ± 2.05%, 10mo MUT 44.01 ± 1.90%, H=74.12, p<0.0001). We observed no difference in number of nuclei per image between genotypes. (**K**) Representative transmission electron microscopy (TEM) image taken of soma from adult (7mo) WT motor cortex. Arrowheads delineate electron-dense lipofuscin material, Nuc=nucleus. (**L**) Representative transmission electron microscopy (TEM) image taken of soma from adult (7mo) MUT motor cortex. (**M**) Inset from (**K**) highlights lysosomal structure in WT soma. Pseudo-colored region depicts lipofuscin area, demarcated as L. (**N**) Inset from (**L**) highlights large lipofuscin deposit in MUT soma (L, pseudo-colored region) with electron-dense and electron-lucent lipid-like (asterisk) components. (**O**) Graph of areas of electron-dense regions of interest (ROI) shows increased ROI size in MUT neurons (WT $2.4 \times 10^5 \pm 2.8 \times 10^4$ nm$^2$, n=50 ROIs; MUT $8.2 \times 10^5 \pm 9.7 \times 10^4$ nm$^2$, n=75 ROIs; U=636, p<0.0001). (**P**) Graph of the average number of presumptive lysosomes with associated electron-dense material reveals increased number in MUT samples (WT 3.14±0.72 ROIs, n=14 images; MUT 10.86 ± 1.42 ROIs, n=14 images; U=17, p<0.0001). For (**O**) and (**P**), images were taken from multiple TEM grids, prepared from n=3 animals per genotype. Scale bar for all TEM images, 1 μm. Data reported as mean ± SEM, error bars are SEM. *p<0.05, ****p<0.0001, Kruskal–Wallis test (**H**), Mann–Whitney U test (**O,P**).

The online version of this article includes the following figure supplement(s) for figure 6:

**Figure supplement 1.** There is no significant difference in striatal, cerebellar, or hippocampal cell death between WT and MUT mice.

**Figure supplement 2.** Dopaminergic innervation is not significantly different between aged WT and MUT mice.

(*Figure 8C*). Mouse performance was not significantly different across trials, which suggested that this difference in retention time was not due to progressive fatigue, but more likely due to an overall difference in motor control (*Mann and Chesselet, 2015*).

To study the animals' movement at a finer scale, the gait of WT and MUT mice was also analyzed using a TreadScan system containing a high-speed camera coupled with a transparent treadmill (*Figure 8D*; *Beare et al., 2009*). Interestingly, while gait parameters of the mice were largely indistinguishable across genotypes at adolescence, a striking difference was seen when the same mice were aged to adulthood (*Figure 8E–G*). In particular, MUT mice took slower (*Figure 8E*), longer strides (*Figure 8F*), stepping closer to the midline of their body (track width, *Figure 8—figure supplement 1*), and their gait symmetry was altered, so that their strides were no longer perfectly out of phase (out of phase=0.5, *Figure 8G*). While these differences were most pronounced in the rear limbs (as depicted in *Figure 8E–G*), the same trends were present in front limbs (*Figure 8—figure supplement 1*). These findings demonstrate that SWIP$^{P1019R}$ results in progressive motor function decline that was detectable by the rotarod task at adolescence, but which became more prominent with age, as both gait and strength functions deteriorated.

These marked motor findings prompted us to re-evaluate the original reports of human SWIP$^{P1019R}$ patients (*Ropers et al., 2011*). While developmental delay or learning difficulties were the primary impetus for medical evaluation, all patients also exhibited motor symptoms (mean age=10.4 years old, *Figure 8H*). The patients' movements were described as 'clumsy' with notable fine motor difficulties, dysmetria, dysdiadochokinesia, and mild dysarthria on clinical exam (*Figure 8H*). Recent communication with the parents of these patients, who are now an average of 21 years old, revealed no notable symptom exacerbation. It is therefore possible that the SWIP$^{P1019R}$ mouse model either exhibits differences from human patients or may predict future disease progression for these individuals, given that we observed significant worsening at 5–6 months old in mice (which is thought to be equivalent to ~30–35 years old in humans) (*Dutta and Sengupta, 2016*; *Zhang et al., 2019*).

## Discussion

Taken together, the data presented here support a mechanistic model whereby SWIP$^{P1019R}$ causes a loss of WASH complex function, resulting in endo-lysosomal disruption and accumulation of neurodegenerative markers, such as upregulation of unfolded protein response modulators and lysosomal enzymes, as well as build-up of lipofuscin and cleaved caspase-3 over time. To our knowledge, this study provides the first mechanistic evidence of WASH complex impairment having direct and indirect organellar effects that lead to cognitive deficits and progressive motor impairments (*Figure 9*).

Using in vivo proximity-based proteomics in wild-type mouse brain, we found that the WASH complex closely interacts with the CCC (COMMD9 and CCDC93) and Retriever (VPS35L) cargo selective complexes (*Bartuzi et al., 2016*; *Singla et al., 2019*). Interestingly, we did not find

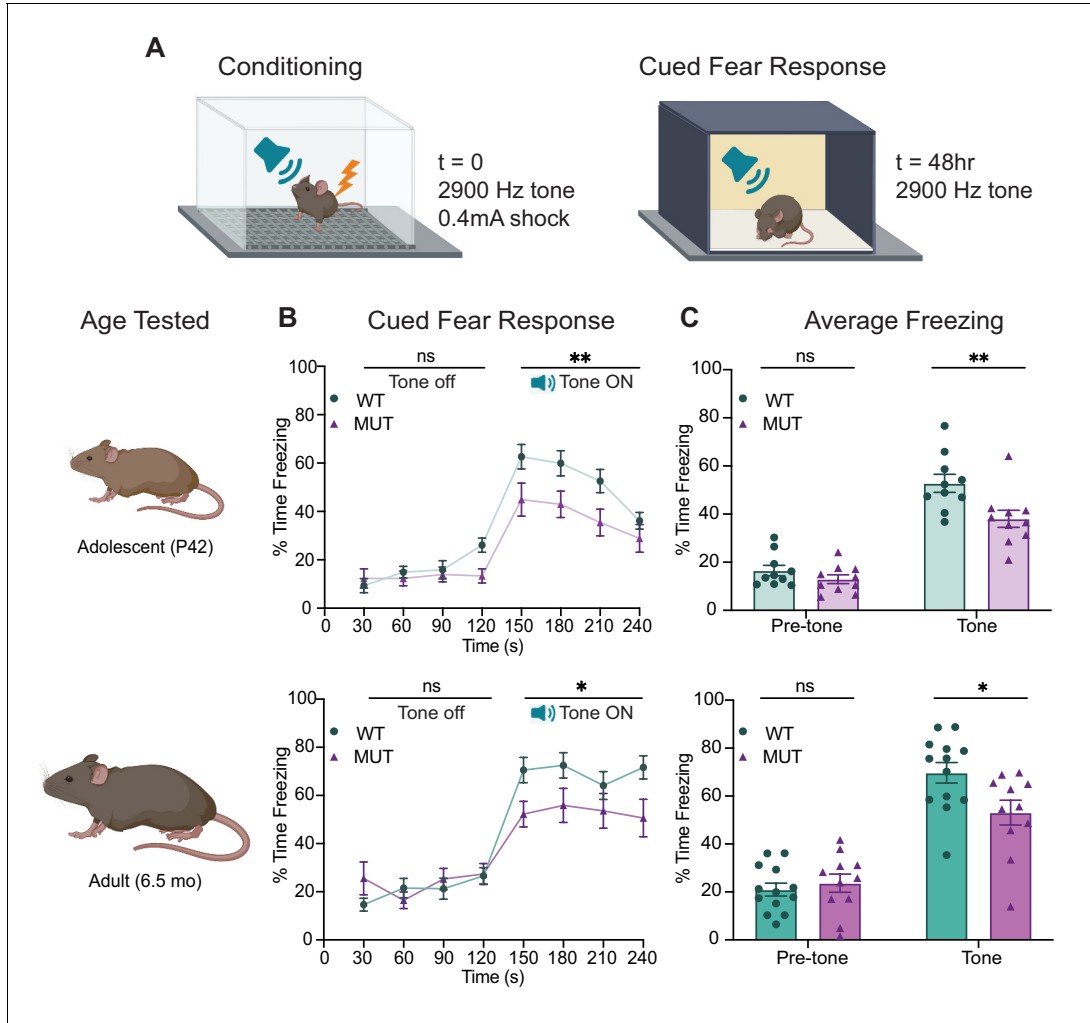

**Figure 7.** SWIP[P1019R] mutant mice display persistent deficits in cued fear memory recall. (**A**) Experimental fear conditioning paradigm. After acclimation to a conditioning chamber, mice received a mild aversive 0.4mA footshock paired with a 2900 Hz tone. 48 hr later, the mice were placed in a chamber with different tactile and visual cues. The mice acclimated for two minutes and then the 2900 Hz tone was played (no footshock) and freezing behavior was assessed. (**B**) Line graphs of WT and MUT freezing response during cued tone memory recall. Data represented as average freezing per genotype in 30 s time bins. The tone is presented after t=120 s, and remains on for 120 s (Tone ON). Two different cohorts of mice were used for age groups P42 (top) and 6.5mo (bottom). Two-way ANOVA analysis of average freezing during Pre-Tone and Tone periods reveal a Genotype x Time effect at P42 (WT n=10, MUT n=10, $F_{1,18}$=4.944, p=0.0392) and 6.5mo (WT n=13, MUT n=11, $F_{1,22}$=13.61, p=0.0013). (**C**) Graphs showing the average %time freezing per animal before and during tone presentation. Top: freezing is reduced by 20% in MUT adolescent mice compared to WT littermates (Pre-tone WT 16.5 ± 2.2%, n=10; Pre-tone MUT 13.0 ± 1.8%, n=10; $t_{36}$=0.8569, p=0.6366; Tone WT 52.8 ± 3.8%, n=10; Tone MUT 38.0 ± 3.6%, n=10; $t_{36}$=3.539, p=0.0023), Bottom: freezing is reduced by over 30% in MUT adult mice compared to WT littermates (Pre-tone WT 21.1 ± 2.7%, n=13; Pre-tone MUT 23.7±3.8%, n=11; $t_{44}$=0.4675, p=0.8721; Tone WT 69.7 ± 4.3%, n=13; Tone MUT 53.1 ± 5.2%, n=11; $t_{44}$=2.921, p=0.0109). Data reported as mean ± SEM, error bars are SEM. *p<0.05, **p<0.01, two-way ANOVAs (**B**) and Sidak's post hoc analyses (**C**).

The online version of this article includes the following figure supplement(s) for figure 7:

**Figure supplement 1.** SWIP[P1019R] mutant mice do not display deficits in spatial working memory or novel object recognition.

**Figure supplement 2.** SWIP[P1019R] mutant mice do not have significant deficits in contextual fear memory recall, auditory perception, or tactile sensation.

significant enrichment of the Retromer sorting complex, a well-known WASH interactor (*Figure 1*), which may be the by-product of using WASH1 rather than another WASH subunit for BioID tagging. Future studies on these protein candidates may clarify how these molecular interactions occur and influence WASH function in mouse brain.

These data are consistent with our spatial proteomics analyses of SWIP[P1019R] mutant brain, which clustered the WASH, CCC, Retriever, and CORVET/HOPS complexes together in M38 (*Figure 3*)

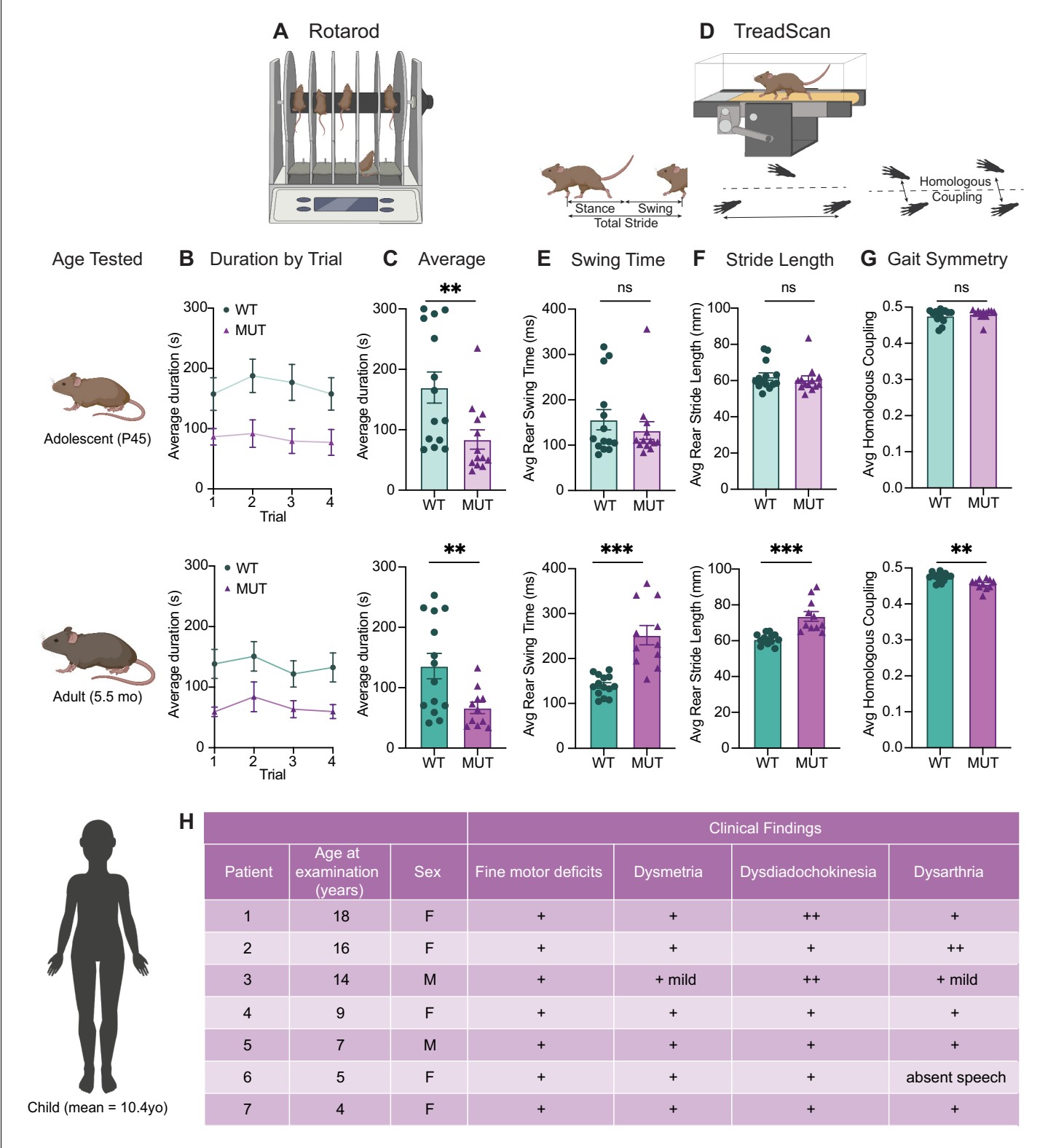

**Figure 8.** SWIP[P1019R] mutant mice exhibit surprising motor deficits that are confirmed in human patients. (**A**) Rotarod experimental setup. Mice walked atop a rod rotating at 32 rpm for 5 min, and the duration of time they remained on the rod before falling was recorded. (**B**) Line graph of average duration animals remained on the rod per genotype across four trials, with an inter-trial interval of 40 min. The same cohort of animals was tested at two different ages, P45 (top) and 5.5 months (bottom). Genotype had a significant effect on task performance at both ages (top, P45: genotype effect, $F_{1,25}=7.821$, $p=0.0098$. bottom, 5.5mo: genotype effect, $F_{1,23}=7.573$, $p=0.0114$). (**C**) Graphs showing the average duration each animal remained on the

*Figure 8 continued on next page*

*Figure 8 continued*

rod across trials. At both ages, the MUT mice exhibited an almost 50% reduction in their ability to remain on the rod (top, P45: WT 169.9 ± 25.7 s, MUT 83.8 ± 15.9 s, U=35, p=0.0054; bottom, 5.5mo: WT 135.9 ± 20.9 s, MUT 66.7 ± 9.5 s, $t_{18}$=3.011, p=0.0075). (D) TreadScan task. Mice walked on a treadmill for 20 s while their gate was captured with a high-speed camera. Diagrams of gait parameters measured in (E–G) are shown below the TreadScan apparatus. (E) Average swing time per stride for hindlimbs. At P45 (top), there is no significant difference in rear swing time (WT 156.2 ± 22.4 ms, MUT 132.3 ± 19.6 ms, U=83, p=0.7203). At 5.5mo (bottom), MUT mice display significantly longer rear swing time (WT 140 ± 6.2 ms, MUT 252.0 ± 21.6 ms, $t_{12}$=4.988, p=0.0003). (F) Average stride length for hindlimbs. At P45 (top), there is no significant difference in stride length (WT 62.3 ± 2.0 mm, MUT 60.5 ± 2.1 mm, U=75, p=0.4583). At 5.5mo (bottom), MUT mice take significantly longer strides with their hindlimbs (WT 60.8 ± 0.8 mm, MUT 73.6 ± 2.7 mm, $t_{11.7}$=4.547, p=0.0007). (G) Average homologous coupling for front and rear limbs. Homologous coupling is 0.5 when the left and right feet are completely out of phase. At P45 (top), WT and MUT mice exhibit normal homologous coupling (WT 0.48 ± 0.005, MUT 0.48 ± 0.004, U=76.5, p=0.4920). At 5.5 mo (bottom), MUT mice display decreased homologous coupling, suggestive of abnormal gait symmetry (WT 0.48 ± 0.003, MUT 0.46 ± 0.004, $t_{18.8}$=3.715, p=0.0015). At P45: n=14 WT, n=13 MUT; At 5.5mo: n=14 WT, n=11 MUT. (H) Table of motor findings in clinical exam of human patients with the homozygous SWIP[P1019R] mutation. All patients exhibit motor dysfunction (+ = symptom present). Data reported as mean ± SEM, error bars are SEM. *p<0.05, **p<0.01, ***p<0.001, ****p<0.0001, two-way repeated measure ANOVAs (B), Mann–Whitney U tests and two-tailed t-tests (C–G).

The online version of this article includes the following figure supplement(s) for figure 8:

**Figure supplement 1.** Progressive gait changes in SWIP[P1019R] mutant mice are not restricted to rear limbs.

and the Retromer complex in a different endosomal-enriched module, M22 (*Figure 4—figure supplement 1A*). Spatial proteomics analyses also revealed that disruption of these WASH–CCC–Retriever–CORVET/HOPS interactions may have multiple downstream effects on the endosomal machinery, since both endosomal-enriched modules exhibited significant decrease in SWIP[P1019R] brain (M38 and M22, *Figure 3* and *Figure 4—figure supplement 1A*). These modules include corresponding decreases in the abundance of endosomal proteins including Retromer subunits (VPS29, VPS26B, and VPS35; M22), associated sorting nexins (SNX27; M22), known WASH interactors (FKBP15; M38), and cargos (e.g. LRP1; M22) (*Figure 3* and *Figure 4—figure supplement 1*; *Del Olmo et al., 2019*; *Farfán et al., 2013*; *Fedoseienko et al., 2018*; *Halff et al., 2019*; *Harbour et al., 2012*; *McNally et al., 2017*; *Pan et al., 2010*; *Ye et al., 2020*; *Zimprich et al., 2011*). While previous studies have indicated that Retromer and CCC influence the endosomal localization of WASH (*Harbour et al., 2012*; *Phillips-Krawczak et al., 2015*; *Singla et al., 2019*), our findings demonstrate both protein- and module-level decreases in abundance of these proteins, pointing to a cascade of endosomal dysfunction. Future studies defining the hierarchical interplay between the WASH, Retromer, Retriever, and CCC complexes in neurons could provide clarity on how these mechanisms are organized.

Of note, some of the lysosomal enzymes with elevated levels in MUT brain (GRN, HEXA, and GLB1 – M36; *Figure 4*) are also implicated in lysosomal storage disorders, where they generally have decreased, rather than increased, function or expression (*Boles and Proia, 1995*; *Regier and Tifft, 1993*; *Smith et al., 2012*; *Ward et al., 2017*). This divergent lysosomal effect in our SWIP[P1019R] model compared to other degenerative models could represent either a distinct endo-lysosomal disruption that culminates in similar cellular pathology or a transient compensatory state that may ultimately lead to declined lysosomal function in SWIP[P1019R] neurons. We speculate that loss of WASH function in our mutant mouse model may lead to increased accumulation of cargo and associated machinery at early endosomes (as seen in *Figure 5*, enlarged EEA1[+] puncta), eventually overburdening early endosomal vesicles and triggering transition to late endosomes for subsequent fusion with degradative lysosomes (*Figure 9*). This would effectively increase delivery of endosomal substrates to the lysosome compared to baseline, resulting in enlarged, overloaded lysosomal structures, and elevated demand for degradative enzymes. For example, since mutant neurons display increased abundance of a lysosomal module (*Figure 4*), and larger lysosomal structures (*Figures 5* and *6*), they may require higher quantities of progranulin (GRN, M36; *Figure 4*) for sufficient lysosomal acidification (*Tanaka et al., 2017*).

Our findings that SWIP[P1019R] results in reduced WASH complex stability and function, which may ultimately drive lysosomal dysfunction, are supported by studies in non-mammalian cells. For example, expression of a dominant-negative form of WASH1 in amoebae impairs recycling of lysosomal V-ATPases (*Carnell et al., 2011*) and loss of WASH in *Drosophila* plasmocytes affects lysosomal acidification (*Gomez et al., 2012*; *Nagel et al., 2017*; *Zech et al., 2011*). Moreover, mouse embryonic

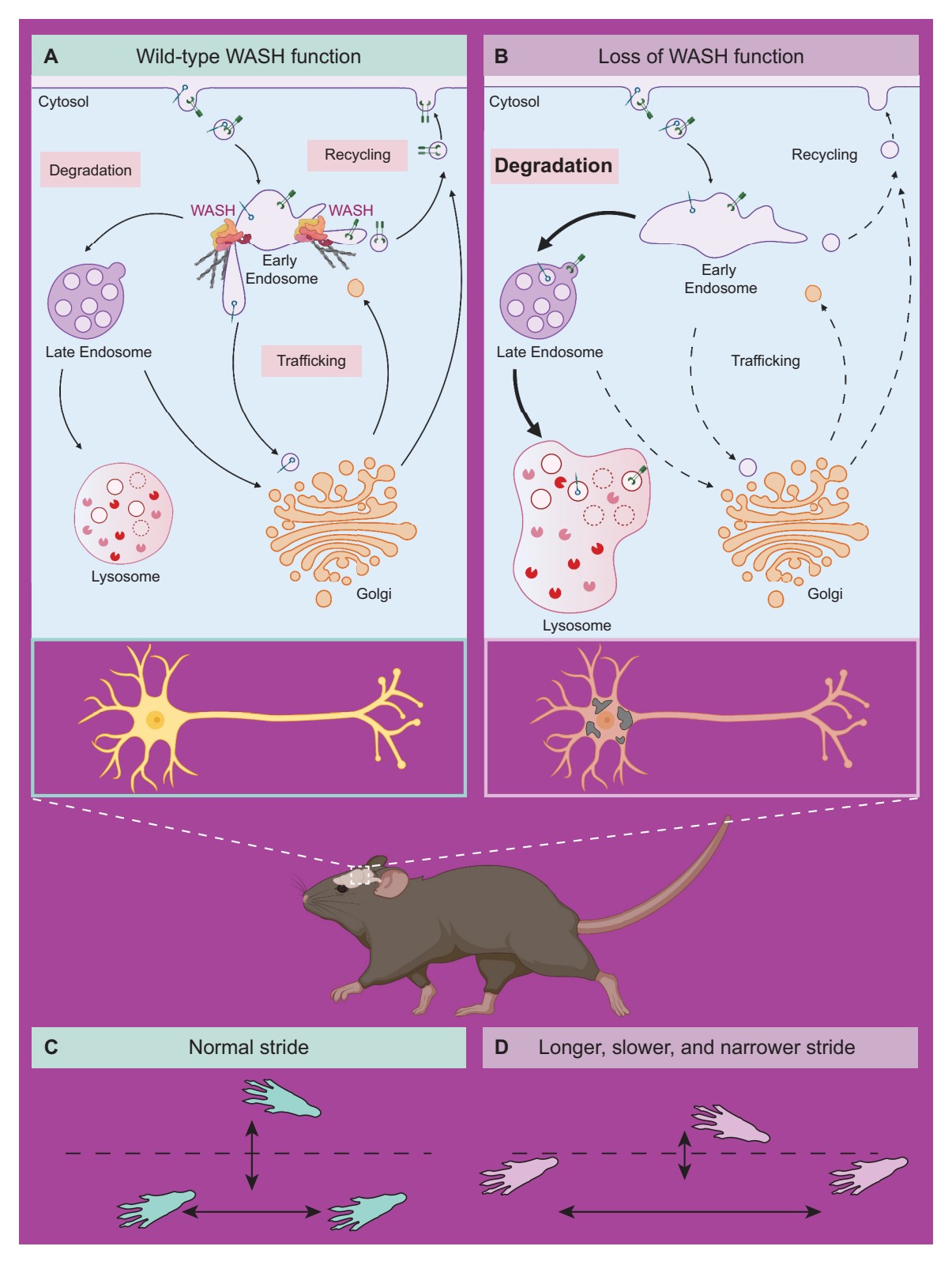

**Figure 9.** Model of neuronal endo-lysosomal pathology in SWIP^P1019R mutant mice. (**A**) Wild-type WASH function in mouse brain. Under normal conditions, the WASH complex interacts with many endosomal proteins and cytoskeletal regulators, such as the Arp2/3 complex. These interactions enable restructuring of the endosome surface (actin in gray) and allow for cargo segregation and scission of vesicles. Substrates are transported to the late endosome for lysosomal degradation, to the Golgi network for modification, or to the cell surface for recycling. (**B**) Loss of WASH function leads to

*Figure 9 continued on next page*

Figure 9 continued

increased lysosomal degradation in mouse brain. Destabilization of the WASH complex leads to enlarged endosomes and lysosomes, with increased substrate accumulation at the lysosome. This suggests an increase in flux through the endo-lysosomal pathway, possibly as a result of mis-localized endosomal substrates. (C) Wild-type mice exhibit normal motor function. (D) SWIP[P1019R] mutant mice display progressive motor dysfunction in association with these subcellular alterations.

fibroblasts lacking WASH1 display abnormal lysosomal morphologies, akin to the structures we observed in cultured SWIP[P1019R] MUT neurons (*Gomez et al., 2012*). Consistent with the idea that WASH regulates lysosomal V-ATPase function either directly or indirectly, we observed a significant decrease in the overall abundance of module M35, a module containing 6 of the 13 components of the vacuolar-associated ATPase complex subunits (CORUM: ATP6V1A, ATP6V1E1, ATP6V0C, ATP6V1F, ATP6V1C1, and ATP6V0A1; *Supplementary files 3–4*). The overall significant decrease in this module resonates with previous studies linking WASH to V-ATPase acidification of lysosomes.

In addition to lysosomal dysfunction, endoplasmic reticulum (ER) stress is commonly observed in neurodegenerative states, where accumulation of misfolded proteins disrupts cellular proteostasis (*Cai et al., 2016*; *Hetz and Saxena, 2017*; *Montibeller and de Belleroche, 2018*). This cellular strain triggers the adaptive unfolded protein response (UPR), which attempts to restore cellular homeostasis by increasing the cell's capacity to retain misfolded proteins within the ER, remedy mis-folded substrates, and trigger degradation of persistently misfolded species. Involved in this process are ER chaperones that we identified as increased in SWIP[P1019R] mutant brain including BiP (HSPA5), calreticulin (CALR), calnexin (CANX), and the protein disulfide isomerase family members (PDIA1, PDIA4, PDIA6; M6 *Figure 4—figure supplement 1C–D*; *Garcia-Huerta et al., 2016*). Many of these proteins were identified in the ER protein module found to be significantly altered in MUT mouse brain (M6), supporting a network-level change in the ER stress response (*Figure 4—figure supplement 1D*). One notable exception to this trend was the chaperone endoplasmin (HSP90B1, M22), which exhibited significantly decreased abundance in SWIP[P1019R] mutant brain (*Supplementary file 2*). This is surprising given that endoplasmin has been shown to coordinate with BiP in protein folding (*Sun et al., 2019*); however, it may highlight a possible compensatory mechanism. Additionally, prolonged UPR can stimulate autophagic pathways in neurons, where misfolded substrates are delivered to the lysosome for degradation (*Cai et al., 2016*). These data highlight a potential pathogenic relationship between ER and endo-lysosomal disturbances as an exciting avenue for future research.

Strikingly, we observed modules enriched for resident proteins corresponding to all 10 of the major subcellular compartments mapped by *Geladaki et al., 2019*: nucleus, mitochondria, golgi apparatus, ER, peroxisome, proteasome, plasma membrane, lysosome, cytoplasm, and ribosome; *Figure 3—figure supplement 5*. The greatest dysregulations, as quantified by log$_2$Fold-Change between genotypes, were in lysosomal, endosomal, ER, and synaptic modules, supporting the hypothesis that SWIP[P1019R] primarily results in disrupted endo-lysosomal trafficking. While analysis of these dysregulated modules informs the pathobiology of SWIP[P1019R], our spatial proteomics approach also identified numerous biologically cohesive modules, which remained unaltered (*Figure 3—figure supplement 5*). Given that many of these modules contained proteins of unknown function, we anticipate that future analyses of these modules and their protein constituents have great potential to inform our understanding of protein networks and their influence on neuronal cell biology.

It has become clear that preservation of the endo-lysosomal system is critical to neuronal function, as mutations in mediators of this process are implicated in neurological diseases such as Parkinson's disease, Huntington's disease, Alzheimer's disease, frontotemporal dementia, neuronal ceroid lipofuscinoses (NCLs), and hereditary spastic paraplegia (*Baker et al., 2006*; *Connor-Robson et al., 2019*; *Edvardson et al., 2012*; *Follett et al., 2019*; *Harold et al., 2009*; *Mukherjee et al., 2019*; *Pal et al., 2006*; *International Parkinsonism Genetics Network et al., 2013*; *Seshadri et al., 2010*; *Tachibana et al., 2019*; *Valdmanis et al., 2007*). These genetic links to predominantly neurodegenerative conditions have supported the proposition that loss of endo-lysosomal integrity can have compounding effects over time and contribute to progressive disease pathologies. In particular, mutations associated with Parkinson's disease have been found in a close endosomal interactor of the WASH complex – the retromer protein VPS35 (VPS35[D620N] and VPS35[R524W]) – and have been linked to pathological α-synuclein aggregation in vitro (*Chen et al., 2019*; *Follett et al., 2014*;

*Tang et al., 2015*). While α-synuclein (SNCA) was highly enriched in our WASH1-BioID assay in WT brain (*Figure 1*), its protein abundance was not found to be significantly different in SWIP[P1019R] mutant brain fractions compared to WT in our TMT spatial proteomic analysis (*Supplementary file 2*).

In addition, unlike many Parkinson's disease models, which display specific deficits in dopaminergic cells, we did not observe any dopaminergic cell-specific changes in SWIP[P1019R] brain (*Figure 6— figure supplement 2*). This suggests that the motor pathology of SWIP[P1019R] mice diverges from that of α-synuclein-driven Parkinson's mouse models. The more parsimonious explanation may be that α-synuclein's enrichment in the WASH1-BioID proteome results from its colocalization with the WASH complex at the endosome and throughout the endo-vesicular system in neurons (*Boassa et al., 2013*; *Bodain, 1965*; *Burre et al., 2010*; *Iwai et al., 1995*; *Lee et al., 2005*).

While our SWIP[P1019R] model appears to diverge in pathology from Parkinson's disease models, it does exhibit parallels to NCL models. NCLs are lysosomal storage disorders primarily found in children rather than adults, with heterogenous presentations and multigenic causations (*Mukherjee et al., 2019*). The majority of genes implicated in NCLs affect lysosomal enzymatic function or transport of proteins to the lysosome (*Mukherjee et al., 2019*; *Poët et al., 2006*; *Ramirez-Montealegre and Pearce, 2005*; *Yoshikawa et al., 2002*). Most patients present with marked neurological impairments, such as learning disabilities, motor abnormalities, vision loss, and seizures, and have the unifying feature of lysosomal lipofuscin accumulation upon pathological examination (*Mukherjee et al., 2019*). While the human SWIP[P1019R] mutation has not been classified as an NCL (*Ropers et al., 2011*), findings from our mutant mouse model suggest that loss of WASH complex function leads to phenotypes bearing strong resemblance to NCLs, including lipofuscin accumulation (*Figures 5–8*). As a result, our mouse model could provide the opportunity to study these pathologies at a mechanistic level, while also enabling preclinical development of treatments for their human counterparts.

Currently, there is an urgent need for greater mechanistic investigations of neurodegenerative disorders, particularly in the domain of endo-lysosomal trafficking. Despite the continual increase in identification of human disease-associated genes, our molecular understanding of how their protein equivalents function and contribute to pathogenesis remains limited. Here we employ a system-level analysis of proteomic datasets to uncover biological perturbations linked to SWIP[P1019R]. We demonstrate the power of combining in vivo proteomics and system network analyses with in vitro and in vivo functional studies to uncover relationships between genetic mutations and molecular disease pathologies. Applying this platform to study organellar dysfunction in other neurodegenerative and neurodevelopmental disorders may facilitate the identification of convergent disease pathways driving brain disorders.

# Materials and methods

**Key resources table**

| Reagent type (species) or resource | Designation | Source or reference | Identifiers | Additional information |
|---|---|---|---|---|
| Gene (*Homo sapiens*) | *WASHC4* | GenBank | Gene ID: 23325 | Aka SWIP |
| Gene (*Mus musculus*) | *Washc4* | Ensembl | ENSMUSG00000034560 | |
| Strain, strain background (*Mus musculus*) | SWIP[P1019R] | This paper, Duke Transgenic Mouse Facility | ENSMUSG00000034560 | Mouse line maintained by Soderling lab |
| Strain, strain background (*Mus musculus*) | B6SJLF1/J | Jackson Laboratories | Cat# 100012 RRID:IMSR_JAX:100012 | |
| Strain, strain background (*Mus musculus*) | C57BL/6J | Jackson Laboratories | Cat# 000664 RRID:IMSR_JAX:000664 | |

*Continued on next page*

*Continued*

| Reagent type (species) or resource | Designation | Source or reference | Identifiers | Additional information |
|---|---|---|---|---|
| Recombinant DNA reagent | pmCAG-SWIP-WT-HA (plasmid) | This paper | SWIP-WT | AAV construct to transfect and express the recombinant DNA Sequence |
| Recombinant DNA reagent | pmCAG-SWIP-MUT-HA (plasmid) | This paper | SWIP-MUT | AAV construct to transfect and express the recombinant DNA Sequence |
| Recombinant DNA reagent | phSyn1-WASH1-BioID2-HA (plasmid) | This paper | WASH1-BioID2 | Transduced AAV construct Sequence |
| Recombinant DNA reagent | phSyn1- solubleBioID2-HA (plasmid) | This paper | SolubleBioID2 control | Transduced AAV construct Sequence |
| Recombinant DNA reagent | pAd-DeltaF6 | Addgene | pAd-DeltaF6 | Helper plasmid for AAV2/9 viral preparation Sequence |
| Recombinant DNA reagent | pAAV2/9 | Addgene | pAAV2/9 | Viral capsid Sequence |
| Cell line (*Mus musculus*) | Primary mouse cortical cultures | This paper | SWIP WT, SWIP[P1019R] MUT neurons | Freshly isolated from wild-type or SWIP[P1019R] P0 mouse brains |
| Cell line (*Homo sapiens*) | Human Embryonic Kidney 293 T cells | Duke Cell Culture Facility | ATTC Cat# CRL-11268 RRID:CVCL_1926 | |
| Sequence-based reagent | *Washc4* CRISPR sgRNA | This paper | Oligonucleotide sequence | N20+ PAM sequence targeting mouse *Washc4* gene for introducing C/G mutation 5'ttgagaatactcacaagagg agg3' |
| Sequence-based reagent | *Washc4*_F repair | This paper | Forward repair oligonucleotide | Forward strand of the repair oligo used to introduce C/G mutation into mouse *Washc4* gene 5'atttcgaaggccaaag aatatacatctccgaaatt tctatatcattgttcgtcctctt gtgagtattctcaaaact agaagtgagttattgatggg tgttaatacagattcagtt tccataaagca3' |
| Sequence-based reagent | *Washc4*_R repair | This paper | Reverse repair oligonucleotide | Reverse strand of the repair oligo used to introduce C/G mutation into mouse *Washc4* gene 5'tgctttatggaaact gaatctgtattaacaccca tcaataactcacttctagtttt gagaatactcacaag aggacgaacaatgatatag aaatttcggagatgtat attctttggccttcgaaat3' |
| Sequence-based reagent | *Washc4*_F mutagenesis | This paper | Forward primer | *Washc4* C/G mutagenesis forward primer 5'ctacaaagttgagggtcagac ggggaacaattatatagaaa3' |
| Sequence-based reagent | *Washc4*_R mutagenesis | This paper | Reverse primer | *Washc4* C/G mutagenesis reverse primer 5'tttctatataattgttccccgtctga ccctcaactttgtag3' |

*Continued on next page*

*Continued*

| Reagent type (species) or resource | Designation | Source or reference | Identifiers | Additional information |
|---|---|---|---|---|
| Sequence-based reagent | *Washc4*_F genotyping | This paper | Forward primer | *Washc4* forward primer for genotyping SWIP[P1019R] mice 5'tgcttgtagatgtttttcct3' |
| Sequence-based reagent | *Washc4*_R genotyping | This paper | Reverse primer | *Washc4* reverse primer for genotyping SWIP[P1019R] mice 5'gttaacatgatcctatggcg3' |
| Antibody | Anti-human WASH1 (C-terminal, rabbit monoclonal) | Sigma Aldrich | Cat# SAB42200373 | WB (1:500) |
| Antibody | Anti-human Strumpellin (rabbit polyclonal) | Santa Cruz | Cat# sc-87442 RRID:AB_2234159 | WB (1:500) |
| Antibody | Anti-human EEA1 (rabbit monoclonal) | Cell Signaling Technology | Clone# C45B10 Cat# 3288 RRID:AB_2096811 | WB (1:1500) ICC (1:500) |
| Antibody | Anti-human LAMP1 (rabbit monoclonal) | Cell Signaling Technology | Clone# C54H11 Cat# 3243 RRID:AB_2134478 | WB (1:2000) |
| Antibody | Anti-human Beta Tubulin III (mouse monoclonal) | Sigma Aldrich | Clone# SDL.3D10 Cat# T8660 RRID:AB_477590 | WB (1:10,000) |
| Antibody | Anti-human HA (mouse monoclonal) | BioLegend | Clone# 16B12 Cat# MMS-101P RRID:AB_10064068 | WB (1:5000) |
| Antibody | Anti-mouse Cathepsin D (rat monoclonal) | Novus Biologicals | Clone# 204712 Cat# MAB1029 RRID:AB_2292411 | ICC (1:250) |
| Antibody | Anti-human MAP2 (guinea pig polyclonal) | Synaptic Systems | Cat# 188004 RRID:AB_2138181 | ICC (1:500) |
| Antibody | Anti-human Cleaved Caspase-3 (rabbit polyclonal) | Cell Signaling Technology | Specificity Asp175 Cat#9661 RRID:AB_2341188 | IHC (1:2000) |
| Antibody | Anti-human Calbindin (mouse monoclonal) | Sigma Aldrich | Clone# CB-955 Cat# C9848 RRID:AB_476894 | IHC (1:2000) |
| Antibody | Anti-human HA (rat monoclonal) | Sigma Aldrich | Clone# 3F10 Cat# 11867423001 RRID:AB_390918 | IHC (1:500) |
| Antibody | Anti-human Bassoon (mouse monoclonal) | Abcam | Clone# SAP7F407 Cat# ab82958 RRID:AB_1860018 | IHC (1:500) |
| Antibody | Anti-human Homer1 (rabbit polyclonal) | Synaptic Systems | Cat# 160002 RRID:AB_2120990 | IHC (1:500) |
| Antibody | Anti-human Tyrosine Hydroxylase (chicken polyclonal) | Abcam | Cat# ab76442 RRID:AB_1524535 | IHC (1:1000) |
| Antibody | Anti-human NeuN (mouse monoclonal) | Abcam | Clone# 1B7 Cat# ab104224 RRID:AB_10711040 | IHC (1:1000) |
| Commercial assay or kit | TMTpro 16plex Label Reagent | Thermo Fisher | Cat# A44520 | |
| Commercial assay or kit | NeutrAvidin Agarose Resins | Thermo Fisher | Cat# 29201 | |
| Commercial assay or kit | S-Trap Binding Buffer | Profiti | Cat# K02-micro-10 | |

*Continued on next page*

*Continued*

| Reagent type (species) or resource | Designation | Source or reference | Identifiers | Additional information |
|---|---|---|---|---|
| Commercial assay or kit | QuikChange XL Site-Directed Mutagenesis Kit | Agilent | Cat# 200517 | |
| Software, algorithm | Courtland et al., source code | GitHub | | |
| Software, algorithm | MSstatsTMT | GitHub | | PubMed |
| Software, algorithm | leidenalg Python Library | conda | | Version 0.8.1 |
| Software, algorithm | Cytoscape | https://cytoscape.org/ | RRID:SCR_003032 | Version 3.7.2 |
| Software, algorithm | Imaris | Oxford Instruments | RRID:SCR_007370 | Version 9.2.0 |
| Software, algorithm | Zen | Zeiss | RRID:SCR_018163 | Version 2.3 |
| Software, algorithm | Fiji | https://fiji.sc/ | RRID:SCR_002285 | Version 2.0.0-rc-69/1.52 p |
| Software, algorithm | GraphPad Prism | GraphPad Software | RRID:SCR_002798 | Version 8.0 |
| Software, algorithm | Proteome Discoverer | Thermo Fisher | RRID:SCR_014477 | Versions 2.2 and 2.4 |
| Software, algorithm | TreadScan NeurodegenScanSuite | CleverSysInc | | |
| Software, algorithm | EthoVision XT | Noldus Information Technology | RRID:SCR_000441 | Version 11.0 |
| Software, algorithm | Rotarod apparatus for mouse | Med Associates | Cat# ENV-575M | |
| Software, algorithm | Fear conditioning chamber | Med Associates | Cat# VFC-008-LP | |
| Software, algorithm | FreezeScan software | CleverSysInc | RRID:SCR_014495 | |
| Software, algorithm | Startle reflex chamber and software | Med Associates | Cat# MED-ASR-PRO1 | |
| Other | Geladaki et al.'s, LOPIT-DC protocol | PubMed | PMCID:PMC6338729 | Subcellular fractionation protocol |
| Other | Orbitrap Fusion Lumos Tribrid Mass Spectrometer | Duke Proteomics and Metabolomics Shared Resource | | Mass spectrometer used for spatial proteomics |
| Other | Thermo QExactive HF-X Mass Spectrometer | Duke Proteomics and Metabolomics Shared Resource | | Mass spectrometer used for iBioID |
| Other | Zeiss 710 LSM confocal microscope | Duke Light Microscopy Core Facility (LCMF) | RRID:SCR_018063 | Confocal microscope used for image acquisition of ICC and IHC samples |
| Other | Reichert Ultracut E Microtome | Duke Department of Pathology | | Microtome used to prepare TEM samples |
| Other | Phillips CM12 Electron Microscope | Duke Department of Pathology | | Transmission electron microscope used for TEM image acquisition |
| Other | Beckman XL-90 Centrifuge and Ti-70 rotor | Duke Department of Cell Biology | | Ultracentrifuge used for AAV virus preparation |
| Other | Beckman TLA-100 Ultracentrifuge and TLA-55 rotor | Duke Department of Cell Biology | | Ultracentrifuge used for spatial proteomics sample preparation |
| Other | DAPI stain | Thermo Fisher | Cat# D1306 RRID:AB_2629482 | (1 µg/mL) |

## Animals

We generated *Washc4* mutant (SWIP[P1019R]) mice in collaboration with the Duke Transgenic Core Facility to mimic the de novo human variant at amino acid 1019 of human *WASHC4.* A CRISPR-induced CCT>CGT point mutation was introduced into exon 29 of *Washc4*. Fifty nanograms per microliter of the sgRNA (5′-ttgagaatactcacaagaggagg-3′), 100 ng/µL Cas9 mRNA, and 100 ng/µL of a repair oligonucleotide containing the C>G mutation were injected into the cytoplasm of B6SJLF1/J mouse embryos (Jax #100012) (see Key Resources Table for the sequence of the repair oligonucleotide). Mice with germline transmission were then backcrossed into a C57BL/6J background (Jax #000664). At least five backcrosses were obtained before animals were used for behavior. We bred heterozygous SWIP[P1019R] mice together to obtain age-matched mutant and wild-type genotypes for cell culture and behavioral experiments. Genetic sequencing was used to screen for germline transmission of the C>G point mutation (*FOR:* 5′-tgcttgtagatgttttttcct-3′, *REV*: 5′-gttaacat-gatcctatggcg-3′). All mice were housed in the Duke University's Division of Laboratory Animal Resources or Behavioral Core facilities at two to five animals per cage on a 14:10 hr light:dark cycle. All experiments were conducted with a protocol approved by the Duke University Institutional Animal Care and Use Committee in accordance with NIH guidelines.

## Human subjects

We retrospectively analyzed clinical findings from seven children with homozygous *WASHC4*[c.3056C>G] mutations (obtained by Dr. Rajab in 2010 at the Royal Hospital, Muscat, Oman). The original report of these human subjects and parental consent for data use can be found in *Ropers et al., 2011*.

## Cell lines

HEK293T cells (ATCC #CRL-11268) were purchased from the Duke Cell Culture facility and were tested for mycoplasma contamination. HEK239T cells were used for co-immunoprecipitation experiments and preparation of AAV viruses.

## Primary neuronal culture

Primary neuronal cultures were prepared from P0 mouse cortex. P0 mouse pups were rapidly decapitated and cortices were dissected and kept individually in 5 mL Hibernate A (Thermo #A1247501) supplemented with 2% B27(Thermo #17504044) at 4°C overnight to allow for individual animal genotyping before plating. Neurons were then treated with Papain (Worthington #LS003120) and DNAse (VWR #V0335)-supplemented Hibernate A for 18 min at 37°C and washed twice with plating medium (plating medium: Neurobasal A [Thermo #10888022] supplemented with 10% horse serum, 2% B-27, and 1% GlutaMAX [Thermo #35050061]), and triturated before plating at 250,000 cells/well on poly-L-lysine-treated coverslips (Sigma #P2636) in 24-well plates. Plating medium was replaced with growth medium (Neurobasal A, 2% B-27, 1% GlutaMAX) 2 hr later. Cell media was supplemented and treated with AraC at DIV5 (5 uM final concentration/well). Half-media changes were then performed every 4 days.

## Plasmid DNA constructs

For immunoprecipitation experiments, a pmCAG-SWIP-WT-HA construct was generated by PCR amplification of the human *WASHC4* sequence, which was then inserted between NheI and SalI restriction sites of a pmCAG-HA backbone generated in our lab. Site-directed mutagenesis (Agilent #200517) was used to introduce a C>G point mutation into this pmCAG-SWIP-WT-HA construct for generation of a pmCAG-SWIP-MUT-HA construct (*FOR:* 5′-ctacaaagttgagggtcagacggggaacaattata-tagaaa-3′, *REV*: 5′-tttctatataattgttccccgtctgaccctcaactttgtag-3′). For iBioID experiments, an AAV construct expressing hSyn1-WASH1-BioID2-HA was generated by cloning a *Washc1* insert between SalI and HindIII sites of a pAAV-hSyn1-Actin Chromobody-Linker-BioID2-pA construct (replacing Actin Chromobody) generated in our lab. This backbone included a 25 nm GS linker-BioID2-HA fragment from Addgene #80899, generated by *Kim et al., 2016*. An hSyn1-solubleBioID2-HA construct was created similarly, by removing Actin Chromobody from the above construct. Oligonucleotide sequences are reported in Key Resources Table. Links to sequences of the plasmid DNA constructs are available in *Supplementary file 5*.

## AAV viral preparation

AAV preparations were performed as described previously (*Uezu et al., 2016*). The day before transfection, HEK293T cells were plated at a density of $1.5 \times 10^7$ cells per 15 cm$^2$ plate in DMEM media with 10% fetal bovine serum and 1% Pen/Strep (Thermo #11965–092, Sigma #F4135, Thermo #15140–122). Six HEK293T 15 cm$^2$ plates were used per viral preparation. The next day, 30 μg of pAd-DeltaF6 helper plasmid, 15 μg of AAV2/9 plasmid, and 15 μg of an AAV plasmid carrying the transgene of interest were mixed in OptiMEM with PEI-MAX (final concentration 80 μg/mL, Polysciences #24765). Two milliliters of this solution were then added dropwise to each of the 6 HEK293T 15 cm$^2$ plates. Eight hours later, the media was replaced with 20 mL DMEM+10%FBS. Seventy-two hours post-transfection, cells were scraped and collected in the media, pooled, and centrifuged at 1500 rpm for 5 min at RT. The final pellet from the six cell plates was resuspended in 5 mL of cell lysis buffer (15 mM NaCl, 5 mM Tris–HCl, pH 8.5), and freeze-thawed three times using an ethanol/dry ice bath. The lysate was then treated with 50 U/mL of Benzonase (Novagen #70664), for 30 min in a 37°C water bath, vortexed, and then centrifuged at 4500 rpm for 30 min at 4°C. The resulting supernatant containing AAV particles was added to the top of an iodixanol gradient (15%, 25%, 40%, 60% top to bottom) in an Optiseal tube (Beckman Coulter #361625). The gradient was then centrifuged using a Beckman Ti-70 rotor in a Beckman XL-90 ultracentrifuge at 67,000 rpm for 70 min, 18°C. The purified viral solution was extracted from the 40%/60% iodixanol interface using a syringe and placed into an Amicon 100 kDa filter unit (#UFC910024). The viral solution was washed in this filter three times with 1× ice-cold PBS by adding 5 mL of PBS and centrifuging at 4900 rpm for 45 min at 4°C to obtain a final volume of approximately 200 μL of concentrated virus that was aliquoted into 5–10 μL aliquots and stored at −80°C until use.

## Immunocytochemistry

### Primary antibodies

Rabbit anti-EEA1 (Cell Signaling Technology #C45B10, 1:500), Rat anti-Cathepsin D (Novus #204712, 1:250), and Guinea Pig anti-MAP2 (Synaptic Systems #188004, 1:500).

### Secondary antibodies

Goat anti-Rabbit Alexa Fluor 568 (Invitrogen #A11036, 1:1000), Goat anti-Guinea Pig Alexa Fluor 488 (Invitrogen #A11073, 1:1000), Goat anti-Rat Alexa Fluor 488 (Invitrogen #A11006, 1:1000), and Goat anti-Guinea Pig Alexa Fluor 555 (Invitrogen #A21435, 1:1000).

At DIV15, neurons were fixed for 15 min using ice-cold 4%PFA/4% sucrose in 1× PBS, pH 7.4 (for EEA1 staining), or 30 min with 50% Bouin's solution/4% sucrose (for CathepsinD staining, Sigma #HT10132), pH 7.4 (*Cheng et al., 2018*). Fixed neurons were washed with 1× PBS, then permeabilized with 0.25% TritonX-100 in PBS for 8 min at room temperature (RT), and blocked with 5%normal goat serum/0.2%Triton-X100 in PBS (blocking buffer) for 1 hr at RT with gentle rocking. For EEA1/MAP2 staining, samples were incubated with primary antibodies diluted in blocking buffer at RT for 1 hr. For CathepsinD/MAP2 staining, samples were incubated with primary antibodies diluted in blocking buffer overnight at 4°C. For both conditions, samples were washed three times with 1× PBS and incubated for 30 min at RT with secondary antibodies, protected from light. After secondary antibody staining, coverslips were washed three times with 1× PBS and mounted with Fluoro-Save mounting solution (Sigma #345789). See antibody section for primary and secondary antibody concentrations.

## Immunohistochemistry

### Primary antibodies

Rabbit anti-Cleaved Caspase-3 (Cell Signaling Technology #9661, 1:2000), Mouse anti-Calbindin (Sigma #C9848, 1:2000), Rat anti-HA 3F10 (Sigma #12158167001, 1:500), Mouse anti-Bassoon (Abcam #ab82958, 1:500), Rabbit anti-Homer1 (Synaptic Systems #160002, 1:500), Chick anti-Tyrosine Hydroxylase (Abcam #ab76442, 1:1000), and Mouse anti-NeuN (Abcam #ab104224, 1:1000).

### Secondary antibodies

Donkey anti-Rabbit Alexa Fluor 488 (Invitrogen #A21206, 1:2000), Goat anti-Mouse Alexa Fluor 594 (Invitrogen #A11032, 1:2000), Goat anti-Rat Alexa Fluor 488 (Invitrogen #A11006, 1:5000),

Streptavidin Alexa Fluor 594 conjugate (Invitrogen #S32356, 1:5000), Goat anti-Mouse Alexa Fluor 594 (Invitrogen #A11032, 1:500), Donkey anti-Rabbit Alexa Fluor 488 (Invitrogen #A21206, 1:500), Goat anti-Chick Alexa Fluor 568 (Invitrogen #A11041, 1:1000), Goat anti-Mouse Alexa Fluor 647 (Invitrogen #A21235, 1:1000), and 4′,6-diamidino-2-phenylindole (DAPI, Sigma #D9542, 1:1000 for 10 min at RT).

Mice were deeply anesthetized with isoflurane and then transcardially perfused with ice-cold heparinized PBS (25 U/mL) by gravity flow. After clearing of liver and lungs (~2 min), perfusate was switched to ice-cold 4% PFA in 1× PBS (pH 7.4) for 15 min. Brains were dissected, post-fixed in 4% PFA overnight at 4°C, and then cryoprotected in 30% sucrose/1× PBS for 48 hr at 4°C. Fixed brains were then mounted in OTC (Sakura TissueTek #4583) and stored at −20°C until cryosectioning. Every third sagittal section (30 μm thickness) was collected from the motor cortex and striatal regions. Free-floating sections were then permeabilized with 1%TritonX-100 in 1X PBS at RT for 2 hr, and blocked in 1X blocking solution (Abcam #126587) diluted in 0.2%TritonX-100 in 1× PBS for 1 hr at RT. Sections were then incubated in primary antibodies diluted in the 1× blocking solution for two overnights at 4°C. After three washes with 0.2%TritonX-100 in 1× PBS, the sections were then incubated in secondary antibodies diluted in 1× blocking buffer for one overnight at 4°C. Sections were then washed four times with 0.2%TritonX-100 in 1× PBS at RT and mounted onto coverslips with FluoroSave mounting solution (Sigma #345789).

## Western blotting
### Primary antibodies
Rabbit anti-Strumpellin (Santa Cruz #sc-87442, 1:500), Rabbit anti-WASH1 c-terminal (Sigma #SAB4200373, 1:500), Mouse anti-Beta Tubulin III (Sigma #T8660, 1:10,000), Mouse anti-HA (BioLegend #MMS-101P, 1:5000), Rabbit anti-LAMP1 (Cell Signaling Technology #C54H11, 1:2000), and Rabbit anti-EEA1 (Cell Signaling Technology #C45B10, 1:1500).

### Secondary antibodies
Donkey anti-Rabbit-HRP (GE Life Sciences #NA934, 1:5,000), Goat anti-mouse-HRP (GE Life Sciences #NA931, 1:5000), Goat anti-Rabbit IR Dye 800CW (LI-COR # 926–32211).

### Western blotting of whole-brain lysates (*Figure 2*)
Ten micrograms of each sample were electrophoresed through a 12-well, 4–20% SDS–PAGE gel (Bio-Rad #4561096) at 100V for 1 hr at RT, transferred onto a nitrocellulose membrane (GE Life Sciences #GE10600002) at 100 V for 70 min at RT on ice, and blocked with 5% nonfat dry milk in TRIS-buffered saline containing 0.05% Tween-20 (TBST, pH 7.4). Gels were saved for Coomassie staining at RT for 30 min. Membranes were probed with one primary antibody at a time for 24 hr at 4°C and then washed four times with TBST at RT before incubating with the corresponding species-specific secondary antibody at RT for 1 hr. Membranes were washed with TBST, and then enhanced chemiluminescence (ECL) substrate was added (Thermo Fischer #32109). Membranes were exposed to autoradiography films and scanned with an Epson 1670 at 600dpi. We probed each membrane with one antibody at a time, stripped the membrane with stripping buffer (Thermo Fischer #21059) for 10 min at RT, and then blocked for 1 hr at RT before probing with the next antibody. Order of probes: Strumpellin, β-tubulin, and then WASH1. We determined the optical density of the bands using Image J software (NIH). Data obtained from three independent experiments were plotted and statistically analyzed using GraphPad Prism (version 8) software.

### Western blotting of subcellular brain fractions (*Figure 3—figure supplements 1–3*)
Eight micrograms of each sample were electrophoresed through a 15-well, 4–20% SDS–PAGE gel (Bio-Rad #4561096) at 100 V for 1 hr at RT and transferred onto a nitrocellulose membrane (GE Life Sciences #GE10600002) at 100 V for 70 min at RT on ice. Membranes were incubated with Total Protein Stain for 5 min at RT (LI-COR # 926–11015), rinsed, and imaged at 700 nm using an Odyssey Fc imaging system (LI-COR) to determine protein loading. Membranes were then briefly incubated with REVERT solution and blocked with Odyssey blocking buffer (LI-COR #927–50000) for 1 hr at RT. Membranes were probed with one primary antibody at a time for 24 hr at 4°C and then washed four

times with 1× TBST before incubating with secondary antibody at RT for 1 hr. Membranes were washed four times with TBST and then imaged with an Odyssey Fc imaging system at 700 nm and 800 nm. Order of probes: LAMP1 then EEA1. We determined the optical density of the bands using Odyssey Fc Image Studio software. Data obtained from three independent experiments and statistically analyzed using GraphPad Prism (version 8) software.

## Immunoprecipitation

HEK293T cells were transfected with pmCAG-SWIP-WT-HA or pmCAG-SWIP-MUT-HA constructs for three days, as previously described (*Mason et al., 2011*). Cells were lysed with lysis buffer (25 mM HEPES, 150 mM NaCl, 1 mM EDTA, 1% NonidetP-40, pH 7.4) containing protease inhibitors (5 mM NaF, 1 mM orthovanadate, 1 mM AEBSF, and 2 µg/mL leupeptin/pepstatin) and centrifuged at 1700 g for 5 min. Collected supernatant was incubated with 30 µL of pre-washed anti-HA agarose beads (Sigma #A2095) on a sample rotator (15 rpm) for 2 hr at 4°C. Beads were then washed three times with lysis buffer, and sample buffer was added before subjecting to immunoblotting as described above. The protein-transferred membrane was probed individually for WASH1, Strumpellin, and HA. Data were collected from four separate preparations of WT and MUT conditions.

## Electron microscopy

Adult (7mo) WT and MUT SWIP$^{P1019R}$ mice were deeply anesthetized with isoflurane and then transcardially perfused with warmed heparinized saline (25 U/mL heparin) for 4 min, followed by ice-cold 0.15 M cacodylate buffer pH 7.4 containing 2.5% glutaraldehyde (Electron Microscopy Sciences #16320), 3% paraformaldehyde, and 2 mM CaCl$_2$ for 15 min. Brain samples were dissected and stored on ice in the same fixative for 2 hr before washing in 0.1 M sodium cacodylate buffer (three changes for 15 min each). Samples were then post-fixed in 1.0% OsO$_4$ in 0.1 M sodium cacodylate buffer for 1 hr on a rotator. Samples were then washed in three 15 min changes of 0.1 M sodium cacodylate. Samples were then placed into *en bloc* stain (1% uranyl acetate) overnight at 4°C. Subsequently, samples were dehydrated in a series of ascending acetone concentrations including 50%, 70%, 95%, and 100% for three cycles with 15 min incubation at each concentration change. Samples were then placed in a 50:50 mixture of epoxy resin (Epon) and acetone overnight on a rotator. This solution was then replaced twice with 100% fresh Epon for at least 2 hr at room temperature on a rotator. Samples were embedded with 100% Epon resin in BEEM capsules (Ted Pella) for 48 hr at 60°C. Samples were ultrathin sectioned to 60–70 nm on a Reichert Ultracut E ultramicrotome. Harvested grids were then stained with 2% uranyl acetate in 50% ethanol for 30 min and Sato's lead stain for 1 min. Micrographs were acquired using a Phillips CM12 electron microscope operating at 80 kV, at 1700× magnification. Micrographs were analyzed in Adobe Photoshop 2019, using the 'magic wand' tool to demarcate and measure the area of electron-dense and electron-lucent regions of interest (ROIs). Statistical analyses of ROI measurements were performed in GraphPad Prism (version 8) software. The experimenter was blinded to genotype for image acquisition and analysis.

## iBioID Protein Sample Preparation

AAV2/9 viral probes, hSyn1-WASH1-BioID2-HA or hSyn1-solubleBioID2-HA, were injected into wild-type CD1 mouse brains using a Hamilton syringe (#7635–01) at age P0–P1 to ensure viral spread throughout the forebrain (*Glascock et al., 2011*). Fifteen days post-viral injection, biotin was subcutaneously administered at 24 mg/kg for seven consecutive days for biotinylation of proteins in proximity to BioID2 probes. Whole brains were extracted on the final day of biotin injections, snap frozen, and stored in liquid nitrogen until protein purification. Seven brains were used for protein purification of each probe, and each purification was performed three times independently (21 brains total for WASH1-BioID2, 21 for solubleBioID2).

We performed all homogenization and protein purification on ice. A 2 mL Dounce homogenizer was used to individually homogenize each brain in a 1:1 solution of Lysis-R:2X-RIPA buffer solution with protease inhibitors (Roche cOmplete tablets #11836153001). Each sample was sonicated three times for 7 s and then centrifuged at 5000 g for 5 min at 4°C. Samples were transferred to Beckman Coulter 1.5 mL tubes (#344059) and then spun at 45,000 rpm in a Beckman Coulter tabletop ultracentrifuge (TLA-55 rotor) for 1 hr at 4°C. SDS was added to supernatants (final 1%), and samples were then boiled for 5 min at 95°C. We next combined supernatants from the same condition

together (WASH1-BioID2 vs. solubleBioID2) in 15 mL conical tubes to rotate with 30 µL high-capacity NeutrAvidin beads overnight at 4°C (Thermo #29204).

The following day, all steps were performed under a hood with keratin-free reagents. Samples were spun down at 6000 rpm, 4°C for 5 min to pellet the beads and remove supernatant. The pelleted beads then went through a series of washes, each for 10 min at RT with 500 µL of solvent, and then spun down on a tabletop centrifuge to pellet the beads for the next wash. The washes were as follows: 2% SDS twice, 1% TritonX100–1% deoxycholate-25 mM $LiCl_2$ once, 1 M NaCL twice, 50 mM ammonium bicarbonate (Ambic) five times. Beads were then mixed 1:1 with a 2× Laemmli sample buffer that contained 3 mM biotin/50 mM Ambic, boiled for 5 min at 95°C, vortexed three times, and then biotinylated protein supernatants were stored at −80°C until LC–MS/MS.

## LC–MS/MS for iBioID

We gave the Duke Proteomics and Metabolomics Shared Resource (DPMSR) six eluents from streptavidin resins (3× WASH1-BioID2, 3× solubleBioID2), stored on dry ice. Samples were reduced with 10 mM dithiolthreitol for 30 min at 80°C and alkylated with 20 mM iodoacetamide for 30 min at room temperature. Next, samples were supplemented with a final concentration of 1.2% phosphoric acid and 256 µL of S-Trap (Protifi) binding buffer (90% MeOH/100 mM triethylammonium bicarbonate [TEAB]). Proteins were trapped on the S-Trap, digested using 20 ng/µL sequencing grade trypsin (Promega) for 1 hr at 47°C, and eluted using 50 mM TEAB, followed by 0.2% formic acid (FA), and lastly using 50% acetonitrile (ACN)/0.2% FA. All samples were then lyophilized to dryness and resuspended in 20 µL 1%TFA/2% ACN containing 25 fmol/µL yeast alcohol dehydrogenase (UniProtKB P00330; ADH_YEAST). From each sample, 3 µL was removed to create a pooled QC sample (SPQC) which was run analyzed in technical triplicate throughout the acquisition period.

Quantitative LC/MS/MS was performed on 2 µL of each sample, using a nanoAcquity UPLC system (Waters) coupled with a Thermo QExactive HF-X high-resolution accurate mass tandem mass spectrometer (Thermo) via a nanoelectrospray ionization source. Briefly, the sample was first trapped on a Symmetry C18 20 mm × 180 µm trapping column (5 µL/min at 99.9/0.1 vol/vol water/ACN), after which the analytical separation was performed using a 1.8 µm Acquity HSS T3 C18 75 µm × 250 mm column (Waters) with a 90 min linear gradient of 5–30% ACN with 0.1% formic acid at a flow rate of 400 nL/min with a column temperature of 55°C. Data collection on the QExactive HF-X mass spectrometer was performed in a data-dependent acquisition (DDA) mode of acquisition with a r=120,000,000 (@ m/z 200) full MS scan from m/z 375–1600 with a target AGC value of $3e^6$ ions followed by 30 MS/MS scans at r=15,000,000 (@ m/z 200) at a target AGC value of $5e^4$ ions and 45 ms. A 20 s dynamic exclusion was employed to increase depth of coverage. The total analysis cycle time for each sample injection was approximately 2 hr.

## LOPIT-DC subcellular fractionation

We performed three independent fractionation experiments with one adult SWIP mutant brain and one WT mouse brain fractionated in each experiment. Each mouse was sacrificed by isoflurane inhalation and its brain was immediately extracted and placed into a 2 mL Dounce homogenizer on ice with 1 mL isotonic TEVP homogenization buffer (320 mM sucrose, 10 mM Tris base, 1 mM EDTA, 1 mM EGTA, 5 mM NaF, pH7.4 [*Hallett et al., 2008*]). A complete mini protease inhibitor cocktail tablet (Sigma #11836170001) was added to a 50 mL TEVP buffer aliquot immediately before use. Brains were homogenized for 15 passes with a Dounce homogenizer to break the tissue, and then this lysate was brought up to a 5 mL volume with additional TEVP buffer. Lysates were then passed through a 0.5 mL ball-bearing homogenizer for two passes (14 µm ball, Isobiotec) to release organelles. Final brain lysate volumes were approximately 7.5 mL each. Lysates were then divided into replicate microfuge tubes (Beckman Coulter #357448) to perform differential centrifugation, following Geladaki et. al's LOPIT-DC protocol (*Geladaki et al., 2019*). Centrifugation was carried out at 4°C in a tabletop Eppendorf 5424 centrifuge for spins at 200 g, 1000 g, 3000 g, 5000 g, 9000 g, 12,000 g, and 15,000 g. To isolate the final three fractions, a tabletop Beckman TLA-100 ultracentrifuge with a TLA-55 rotor was used at 4°C with speeds of 30,000 g, 79,000 g, and 120,000 g, respectively. Samples were kept on ice at all times, and pellets were stored at −80°C. Pellets from seven fractions (5000 g–120,000g) were used for proteomic analyses.

## 16-plex TMT LC–MS/MS

The Duke Proteomics and Metabolomics Shared Resource (DPMSR) processed and prepared fraction pellets from all 42 frozen samples simultaneously (seven fractions per brain from three WT and three MUT brains). Due to volume constraints, each sample was split into three tubes, for a total of 126 samples, which were processed in the following manner: 100 µL of 8 M urea was added to the first aliquot then probe sonicated for 5 s with an energy setting of 30%. This volume was then transferred to the second and then third aliquots after sonication in the same manner. All tubes were centrifuged at 10,000 g, and any residual volume from tubes 1 and 2 were added to tube 3. Protein concentrations were determined by BCA on the supernatant in duplicate (5 µL each assay). Total protein concentrations for each replicate ranged from 1.1 mg/mL to 7.8 mg/mL with total protein quantities ranging from 108.3 to 740.81 µg. 60 µg of each sample was removed and normalized to 52.6 µL with 8 M urea and 14.6 µL 20% SDS. Samples were reduced with 10 mM dithiolthreitol for 30 min at 80°C and alkylated with 20 mM iodoacetamide for 30 min at room temperature. Next, they were supplemented with 7.4 µL of 12% phosphoric acid and 574 µL of S-Trap (Protifi) binding buffer (90% MeOH/100 mM TEAB). Proteins were trapped on the S-Trap, digested using 20 ng/µL sequencing grade trypsin (Promega) for 1 hr at 47°C, and eluted using 50 mM TEAB, followed by 0.2% FA, and lastly using 50% ACN/0.2% FA. All samples were then lyophilized to dryness.

Each sample was resuspended in 120 µL 200 mM TEAB, pH 8.0. From each sample, 20 µL was removed and combined to form a pooled quality control sample (SPQC). Fresh TMTPro reagent (0.5 mg for each 16-plex reagent) was resuspended in 20 µL 100% ACN and was added to each sample. Samples were incubated for 1 hr at RT. After the 1 hr reaction, 5 µL of 5% hydroxylamine was added and incubated for 15 min at room temperature to quench the reaction. Each 16-plex TMT experiment consisted of the WT and MUT fractions from one mouse, as well as the two SPQC samples. Samples corresponding to each experiment were concatenated and lyophilized to dryness.

Samples were resuspended in 800 µL 0.1% formic acid. 400 µg was fractionated into 48 unique high-pH reversed-phase fractions using pH 9.0 20 mM ammonium formate as mobile phase A and neat ACN as mobile phase B. The column used was a 2.1 mm × 50 mm XBridge C18 (Waters), and fractionation was performed on an Agilent 1100 HPLC with G1364C fraction collector. Throughout the method, the flow rate was 0.4 mL/min and the column temperature was 55°C. The gradient method was set as follows: 0 min, 3%B; 1 min, 7% B; 50 min, 50%B; 51 min, 90% B; 55 min, 90% B; 56 min, 3% B; 70 min, 3% B. 48 fractions were collected in equal time segments from 0 to 52 min, then concatenated into 12 unique samples using every 12th fraction. For instance, fractions 1, 13, 25, and 37 were combined, fractions 2, 14, 26, and 38 were combined, etc. Fractions were frozen and lyophilized overnight. Samples were resuspended in 66 µL 1% TFA/2% ACN prior to LC–MS analysis.

Quantitative LC/MS/MS was performed on 2 µL (1 µg) of each sample, using a nanoAcquity UPLC system (Waters) coupled with a Thermo Orbitrap Fusion Lumos high-resolution accurate mass tandem mass spectrometer (Thermo) equipped with a FAIMS Pro ion-mobility device via a nanoelectrospray ionization source to enhance precursor ion selectivity and quantitative accuracy without losing the depth of coverage. Briefly, the sample was first trapped on a Symmetry C18 20 mm × 180 µm trapping column (5 µL/min at 99.9/0.1 vol/vol water/ACN), after which the analytical separation was performed using a 1.8 µm Acquity HSS T3 C18 75 µm × 250 mm column (Waters) with a 90 min linear gradient of 5–30% ACN with 0.1% formic acid at a flow rate of 400 nL/min with a column temperature of 55°C. Data collection on the Fusion Lumos mass spectrometer was performed for three different compensation voltages (CV: −40 V, −60 V, −80 V). Within each CV, a DDA mode of acquisition with a r=120,000,000 (@ m/z 200) full MS scan from m/z 375–1600 with a target AGC value of $4e^5$ ions was performed. MS/MS scans were acquired in the Orbitrap at r=50,000,000 (@ m/z 200) from m/z 100 with a target AGC value of $1e^5$ and max fill time of 105 ms. The total cycle time for each CV was 1 s, with total cycle times of 3 s between like full MS scans. A 45 s dynamic exclusion was employed to increase depth of coverage. The total analysis cycle time for each sample injection was approximately 2 hr.

Following UPLC–MS/MS analyses, data were imported into Proteome Discoverer 2.4 (Thermo Scientific). The MS/MS data were searched against a SwissProt Mouse database (downloaded November 2019) plus additional common contaminant proteins, including yeast alcohol dehydrogenase (ADH), bovine casein, bovine serum albumin, as well as an equal number of reversed-sequence

'decoys' for FDR determination. Mascot Distiller and Mascot Server (v 2.5, Matrix Sciences) were utilized to produce fragment ion spectra and to perform the database searches. Database search parameters included fixed modification on Cys (carbamidomethyl) and variable modification on Met (oxidation), Asn/Gln (deamidation), Lys (TMTPro), and peptide N-termini (TMTPro). Data were searched at 5 ppm precursor and 0.02 product mass accuracy with full trypsin enzyme rules. Reporter ion intensities were calculated using the Reporter Ions Quantifier algorithm in Proteome Discoverer. Percolator node in Proteome Discoverer was used to annotate the data at a maximum 1% protein FDR.

## Mouse behavioral assays

Behavioral tests were performed on age-matched WT and homozygous SWIP$^{P1019R}$ mutant littermates. Male and female mice were used in all experiments. Testing was performed at two time points: P42–55 days old as a young adult age and 5.5 months old as mid-adulthood, so that we could compare disease progression in this mouse model to human patients (*Ropers et al., 2011*). The sequence of behavioral testing was as follows: Y-maze (to measure working memory), object novelty recognition (to measure short- and long-term object recognition memory), TreadScan (to assess gait), and steady-speed rotarod (to assess motor control and strength) for 40–55 day old mice. Testing was performed over 1.5 weeks, interspersed with rest days for acclimation. This sequence was repeated with the same cohort at 5.5–6 months old, with three additional measures added to the end of testing: fear conditioning (to assess associative fear memory), a hearing test (to measure tone response), and a shock threshold test (to assess somatosensation). Of note, a separate, second cohort of mice was evaluated for fear conditioning, hearing, and shock threshold testing at adolescence. After each trial, equipment was cleaned with Labsan to remove residual odors. The experimenter was blinded to genotype for all behavioral analyses.

## Y-maze

Working memory was evaluated by measuring spontaneous alternations in a three-arm Y-maze under indirect illumination (80–90 lux). A mouse was placed in the center of the maze and allowed to freely explore all arms, each of which had different visual cues for spatial recognition. Trials were 5 min in length, with video data and analyses captured by EthoVision XT 11.0 software (Noldus Information Technology). Entry to an arm was define as the mouse being >1 body length into a given arm. An alternation was defined as three successive entries into each of the different arms. Total % alternation was calculated as the total number of alternations/the total number of arm entries minus $2 \times 100$.

## Novel object recognition

One hour before testing, mice were individually exposed to the testing arena (a 48×22×18 cm white opaque arena) for 10 min under 80–100 lux illumination without any objects. The test consisted of three phases: training (day 1), short-term memory test (STM, day 1), and long-term memory test (LTM, day 2). For the training phase, two identical objects were placed 10 cm apart, against opposing walls of the arena. A mouse was placed in the center of the arena and given full access to explore both objects for 5 min and then returned to its home cage. For STM testing, one of the training objects remained (the now familiar object), and a novel object replaced one of the training objects (similar in size, different shapes). The mouse was returned to the arena 30 min after the training task and allowed to explore freely for 5 min. For LTM testing, the novel object was replaced with another object, and the familiar object remained unchanged. The LTM test was also 5 min in duration, conducted 24 hr after the training task. Behavior was scored using Ethovision 11.0 XT software (Noldus) and analyzed by a blind observer. Object contact was defined as the mouse's nose within 1 cm of the object. We analyzed both number of nose contacts with each object and duration of contacts. Preference scores were calculated as follows: (duration contact$_{novel}$− duration contact$_{familiar}$)/ total duration contact$_{novel+familiar}$. Positive scores signified a preference for the novel object, whereas negative scores denoted a preference for the familiar object, and scores approaching zero indicated no preference.

## TreadScan

A TreadScan forced locomotion treadmill system (CleverSys Inc, Reston, VA) was used for gait recording and analysis. Each mouse was recorded walking on a transparent treadmill at 45 days old and again at 5.5 months old. Mice were acclimated to the treadmill chamber for 1 min before the start of recording to eliminate exploratory behavior confounding normal gait. Trials were 20 s in length, with mice walking at speeds between 13.83 and 16.53 cm/s (P45 WT average 15.74 cm/s; P45 MUT average 15.80 cm/s; 5.5mo WT average 15.77 cm/s; 5.5mo MUT average 15.85 cm/s). A high-speed digital camera attached to the treadmill-captured limb movement at a frame rate of 100 frames/s. We used TreadScan software (CleverSys) and representative WT and MUT videos to generate footprint templates, which were then used to identify individual paw profiles for each limb. Parameters such as stance time, swing time, step length, track width, and limb coupling were recorded for the entire 20 s duration for each animal. Output gait tracking was verified manually by a blinded experimenter to ensure consistent limb tracking throughout the duration of each video.

## Steady speed rotarod

A 5-lane rotarod (Med Associates, St. Albans, VT) was used for steady-speed motor analysis. The rod was run at a steady speed of 32 rpm for four 5 min trials, with a 40 min inter-trial interval. We recorded mouse latency to fall by infrared beam break or manually for any mouse that completed two or more rotations on the rod without walking. Mice were randomized across lanes for each trial.

## Fear conditioning

Animals were examined in contextual and cued fear conditioning as described by *Rodriguiz and Wetsel, 2006*. Two separate cohorts of mice were used in testing the two age groups. A 3-day testing paradigm was used to assess memory: conditioning on day 1, context testing 24 hr post-conditioning on day 2, and cued tone testing 48 hr post-conditioning on day 3. All testing was conducted in fear conditioning chambers (Med Associates). In the conditioning phase, mice were first acclimated to the test chamber for 2 min under ~100 lux illumination. Then a 2900 Hz, 80 dB tone (conditioned stimulus, CS) played for 30 s, which terminated with a paired 0.4 mA, 2 s scrambled foot shock (unconditioned stimulus, US). Mice were removed from the chamber and returned to their home cage 30 s later. In the context testing phase, mice were placed in the same conditioning chamber and monitored for freezing behavior for a 5 min trial period, in the absence of the CS and US. For cued tone testing, the chambers were modified to different dimensions and shapes, contained different floors and wall textures, and lighting was adjusted to 50 lux. Mice acclimated to the chamber for 2 min, and then the CS was presented continuously for 3 min. Contextual and cued fear memory was assessed by freezing behavior, captured by automated video software (CleversSys).

## Hearing test

We tested mouse hearing using a startle platform (Med Associates) connected to Startle Pro Software in a sound-proof chamber. Mice were placed in a ventilated restraint cylinder connected to the startle response detection system to measure startle to each acoustic stimulus. After 2 min of acclimation, mice were assessed for an acoustic startle response to seven different tone frequencies, 2 kHz, 3 kHz, 4 kHz, 8 kHz, 12 kHz, 16 kHz, and 20 kHz that were randomly presented three times each at four different decibels, 80, 100, 105, and 110 dB, for a total of 84 trials. A random inter-trial interval of 15–60 s (average 30 s) was used to prevent anticipation of a stimulus. An animal's reaction to the tone was recorded as startle reactivity in the first 100 ms of the stimulus presentation, which was transduced through the platform's load cell and expressed in arbitrary units (AU).

## Startle response (somatosensation)

Mouse somatosensation was tested by placing mice in a startle chamber (Med Associates) connected to Startle Pro Software. Mice were placed atop a multi-bar cradle within a ventilated plexiglass restraint cylinder, which allows for horizontal movement within the chamber, but not upright rearing. After 2 min of acclimation, each mouse was exposed to 10 different scrambled shock intensities, ranging from 0 to 0.6 mA with randomized inter-trial intervals of 20–90 s. Each animal's startle reactivity during the first 100 ms of the shock was transduced through the platform's load cell and recorded as area under the curve (AUC) in AU.

## Quantification and statistical analysis

Experimental conditions, number of replicates, and statistical tests used are stated in each figure legend. Each experiment was replicated at least three times (or on at least three separate animals) to assure rigor and reproducibility. Both male and female age-matched mice were used for all experiments, with data pooled from both sexes. Data compilation and statistical analyses for all non-proteomic data were performed using GraphPad Prism (version 8, GraphPad Software, CA), using a significance level of alpha=0.05. Prism provides exact p-values unless p<0.0001. All data are reported as mean ± SEM. Each data set was tested for normal distribution using a D'Agostino–Person normality test to determine whether parametric (unpaired Student's t-test, one-way ANOVA, two-way ANOVA) or non-parametric (Mann–Whitney, Kruskal–Wallis, Kolmogorov–Smirnov) tests should be used. Parametric assumptions were confirmed with the Shapiro–Wilk test (normality) and Levine's test (error variance homogeneity) for ANOVA with repeated-measures testing. The analysis of iBioID and TMT proteomics data are described below. All proteomic data and analysis scripts are available online (see key resources table).

## Imaris 3D reconstruction

For EEA1[+] and CathepsinD+ puncta analyses, coverslips were imaged on a Zeiss LSM 710 confocal microscope. Images were sampled at a resolution of 1024 × 1024 pixels with a dwell time of 0.45 µs using a 63×/1.4 oil immersion objective, a 2.0 times digital zoom, and a z-step size of 0.37 µm. Images were saved as '.lsm' formatted files, and quantification was performed on a POGO Velocity workstation in the Duke Light Microscopy Core Facility using Imaris 9.2.0 software (Bitplane, South Windsor, CT). For analyses, we first used the 'surface' tool to make a solid fill surface of the MAP2-stained neuronal soma and dendrites, with the background subtraction option enabled. We selected a threshold that demarcated the neuron structure accurately while excluding background. For EEA1 puncta analyses, a 600 × 800 µm selection box was placed around the soma in each image and surfaces were created for EEA1 puncta within the selection box. Similarly, for CathepsinD puncta analyses, a 600 × 600 µm selection box was placed around the soma(s) in each image for surface creation. The same threshold settings were used across all images, and individual surface data from each soma were exported for aggregate analyses. The experimenter was blinded to sample conditions for both image acquisition and analysis.

## Cleaved caspase-3 image analysis

Z-stack images were acquired on a Zeiss 710 LSM confocal microscope. Images were sampled at a resolution of 1024 × 1024 pixels with a dwell time of 1.58 µs, using a 63×/1.4 oil immersion objective (for cortex, striatum, and hippocampus) or 20×/0.8 dry objective (cerebellum), a 1.0 times digital zoom, and a z-step size of 0.67 µm. Images were saved as '.lsm' formatted files and then converted into maximum intensity projections (MIPs) using Zen 2.3 SP1 software. Quantification of CC3 colocalization with DAPI was performed on the MIPs using the Particle Analyzer function in FIJI ImageJ software. The experimenter was blind to sample conditions for both image acquisition and analysis.

## Synapse quantification image analysis

Z-stack images of the motor cortex were acquired on a Zeiss 710 LSM confocal microscope. Images were sampled at a resolution of 1024 × 1024 pixels with a dwell time of 1.58 µs, using a 63×/1.4 oil immersion objective, a 1.0 times digital zoom, and a z-step size of 0.34 µm, acquiring five steps per image. Images were saved as '.lsm' formatted files and then converted into maximum intensity projections (MIPs) using Zen 2.3 SP1 software. We selected 250 µm x 250 µm regions in the MIPs for analyses. Quantification of bassoon and homer1 colocalization was performed using the Particle Analyzer function of FIJI ImageJ software. The experimenter was blind to sample conditions for both image acquisition and analysis.

## Tyrosine hydroxylase image analysis

Z-stack images of the substantia nigra and striatum were acquired on a Zeiss 710 LSM confocal microscope. Images were sampled at a resolution of 1024 × 1024 pixels with a dwell time of 1.58 µs, using a 40×/1.3 oil immersion objective or 10×/0.45 dry objective, a 1.0 times digital zoom, and a z-step size of 0.67 µm, acquiring five steps per image. Images were saved as '.lsm' formatted

files and then converted into maximum intensity projections (MIPs) using Zen 2.3 SP1 software. Quantification of Tyrosine Hydroxlyase$^+$ (TH$^+$) neurons was performed using the Particle Analyzer function of FIJI ImageJ software. Quantification of dopaminergic innervation of the striatum was obtained by measuring the mean TH$^+$ signal intensity for each image. The experimenter was blind to sample conditions for both image acquisition and analysis.

## iBioID quantitative analysis

Following UPLC–MS/MS analyses, data was imported into Proteome Discoverer 2.2 (Thermo Scientific Inc) and aligned based on the accurate mass and retention time of detected ions ('features') using Minora Feature Detector algorithm in Proteome Discoverer. Relative peptide abundance was calculated based on AUC of the selected ion chromatograms of the aligned features across all runs. The MS/MS data was searched against the SwissProt *Mus musculus* database (downloaded in April 2018) with additional proteins, including yeast ADH1, bovine serum albumin, as well as an equal number of reversed-sequence 'decoys' for false discovery rate (FDR) determination. Mascot Distiller and Mascot Server (v 2.5, Matrix Sciences) were utilized to produce fragment ion spectra and to perform the database searches. Database search parameters included fixed modification on Cys (carbamidomethyl), variable modifications on Meth (oxidation), and Asn and Gln (deamidation) and were searched at 5 ppm precursor and 0.02 Da product mass accuracy with full trypsin enzymatic rules. Peptide Validator and Protein FDR Validator nodes in Proteome Discoverer were used to annotate the data at a maximum 1% protein FDR.

Protein-level intensities were exported from Proteome Discoverer and processed using custom R scripts. Carboxylases, keratins, and mitochondrial proteins (*Calvo et al., 2016*) were removed from the identified proteins as known contaminants. Sample loading normalization was performed to account for technical variation between the nine individual MS runs. In brief, this is done by multiplying intensities from each MS run by a scaling factor, such that total run intensities are equal. We created a pooled QC sample by pooling equivalent aliquots of peptides from each biological replicate and analyzed this in technical duplicate in each experiment. We performed sample pool normalization to SPQC samples to standardize protein measurements across all samples and correct for batch effects between MS analyses. Sample pool normalization adjusts the protein-wise mean of all biological replicates to be equal to the mean of all SPQC replicates. Finally, proteins that were identified by a single peptide and/or identified in less than 50% of samples were removed. Any remaining missing values were inferred to be missing not-at-random due to the left shifted distribution of proteins with missing values and imputed using the k-nearest neighbors algorithm using the impute.knn function in the R package impute (impute::impute.knn). Normalized protein data were then fit with a simple linear model to derive a model-based statistical comparison of WASH iBioID and soluble-BioID2 control groups (*Huang et al., 2020*). To consider a protein enriched in the WASH interactome, we required that a protein exhibit a fold-change greater than 4 over the negative control with a Benjamini−Hochberg false discovery rate (FDR) less than 0.05 (*Benjamini and Hochberg, 1995*). With these criteria, 175 proteins were identified as WASH1 interactome proteins. The statistical results can be found in *Supplementary file 1*.

Proteins that function together often interact directly. We compiled experimentally determined protein–protein interactions (PPIs) among the WASH1 interactome from the HitPredict database (*López et al., 2015*) using a custom R package, getPPIs (available online at twesleyb/getPPIs). We report PPIs among the WASH1 interactome in *Supplementary file 1*.

Bioinformatic GO analysis was conducted by manual annotation of identified proteins and confirmed with Metascape analysis (*Zhou et al., 2019*) of WASH1-BioID2-enriched proteins using the 2102 proteins identified in the mass spectrometry analysis as background.

## Protein-level statistical inference with MSstatsTMT

PSM-level data were exported from Proteome Discover 2.2 and prepared for analysis with MSstatsTMT, an R package for data normalization and hypothesis testing in multiplex TMT proteomics experiments (*Huang et al., 2020*). MSstatsTMT performs statistical inference in two steps. First, each protein in the dataset is fit with a LMM expressing the major sources of variation in the experimental design. Second, given the fitted model, a model-based comparison is made between pairs of treatment conditions.

Given our experimental design, MSstatsTMT fits the following LMM to each protein-level subset of the data:

$$Y_{mcbt} = \mu + Condition_c + Mixture_m + \epsilon_{mcb} \tag{1}$$

The model's constraints delimit the response as a function of fixed and mixed effects:

$$\sum_{c=1}^{C} Condition_c = 0$$
$$\frac{Mixture_m \sim iid\ N\left(0, \sigma_M^2\right)}{\epsilon_{mcb} \sim iid\ N(0, \sigma^2)} \tag{2}$$

Condition is a fixed effect and represents the 14 combinations of Genotype and BioFraction in our experimental design. The term Mixture is a mixed effect and represents variation between the three TMT mixtures. Mixed effects are normally and independently distributed (i.i.d.). The term epsilon (ε) is a mixed effect and represents both biological and technical variations, quantifying any remaining error.

Pairwise contrasts between MUT and WT conditions are obtained by comparing estimates obtained from the LMM fit by restricted maximum likelihood (*Bates et al., 2015*). We are interested in testing the null hypothesis: $l^T * \beta = 0$. Where the contrast, $l^T$ is a vector of sum 0 specifying the positive and negative coefficients of the contrast. Beta (β) is the model-based estimates of Mutant and Control conditions. A test statistic for such a two-way contrast is given by *Kuznetsova et al., 2017*:

$$t = \frac{l^T \hat{\beta}}{\sqrt{l\sigma^2 \hat{V} l^T}} \tag{3}$$

We obtain the model's estimates (β), error ($\sigma^2$), and variance–covariance matrix ($V$) from the fit LMM. The numerator is the log fold-change of a comparison. Together, the denominator represents the standard error of the comparison. The degrees of freedom for the contrast are derived using the Satterthwaite moment of approximation method (*Kuznetsova et al., 2017*). Finally, a p-value is calculated given the t-statistic and degrees of freedom. p-values for all tests of a given contrast are adjusted using the FDR method (*Huang et al., 2020*). Using MSstatsTMT we assessed two types of contrasts. Statistical results for both intra-BioFraction and the overall 'Mutant-Control' comparison are found in *Supplementary file 2*.

## Quantitative TMT proteomics analysis for spatial proteomics

Peptide-level data from the spatial proteomics analysis of SWIP[P1019R] WT and MUT brain were exported from Proteome Discoverer (version 2.4) and analyzed using custom R scripts. Peptides from contaminant and non-mouse proteins were removed. First, we performed sample loading normalization, normalizing the total ion intensity for each TMT channel within an experiment to be equal. Sample loading normalization corrects for small differences in the amount of sample analyzed and labeling reaction efficiency differences between individual TMT channels within an experiment.

We found that in each TMT experiment there were a small number of missing values (mean percent missing=1.6±0.17%). Missing values were inferred to be missing at random imputed using the k-nearest neighbor algorithm in the R package impute (impute::impute.knn). Missing values for SPQC samples were not imputed. Peptides with any missing SPQC data were removed.

Following sample loading normalization, SPQC replicates within each experiment should yield identical measurements. As peptides with irreproducible QC measurements are unlikely to be quantitatively robust, and their inclusion may bias downstream normalization, we sought to remove them. To assess intra-batch peptide variability, we adapted the method described by *Ping et al., 2018*. Briefly, peptides were binned into five groups based on the average intensity of the two SPQC replicates. For each pair of SPQC measurements, the log ratio of SPQC intensities was calculated. To identify outlier QC peptides, we plotted the distribution of these log ratios binned into five intensity bins. Peptides with ratios that were more than 4 standard deviations away from the mean of its intensity bin were considered outliers and removed.

Proteins were summarized as the sum of all unique peptide intensities corresponding to a unique UniProtKB Accession identifier, and sample loading normalization was performed across all three

experiments to account for inter-experimental technical variability. In a TMT experiment, the peptides selected for MS2 fragmentation are partially random, especially at lower signal-to-noise ratios. This stochasticity means that proteins are typically quantified by different peptides in each experiment. Thus, although SPQC samples should yield identical protein measurements in each of the three experiments (as it is the same sample analyzed in each experiment), the observed protein measurements exhibit variability due to their quantification by different peptides. To account for this protein-level bias, we utilized the internal reference scaling (IRS) approach described by *Plubell et al., 2017*. IRS normalization scales the protein-wise geometric average of all SPQC measurements across all experiments to be equal and simultaneously adjusts biological replicates. In brief, each protein is multiplied by a scaling factor, which adjusts its intra-experimental SPQC values to be equal to the geometric mean of all SPQC values for the three experiments. This normalization step effectively standardizes protein measurements between different mass spectrometry experiments.

Before downstream analyses, we removed irreproducible proteins. This included proteins that were quantified in less than 50% of all samples, proteins that were identified by a single peptide, and proteins that had missing SPQC values. Across all 42 biological replicates, we observed that a small number of proteins had potential outlier measurements that were either several orders of magnitude greater or less than the mean of its replicates. In order to identify and remove these proteins, we assessed the reproducibility of protein measurements within a fraction in the same manner used to identify and filter SPQC outlier peptides. A small number of proteins were identified as outliers if the average log ratio of their three QC technical replicates was more than 4 standard deviations away from the mean of its intensity bin. In total, we retained 5897 proteins in the final spatial proteomics dataset.

## Spatial proteomics network construction

To construct a protein covariation graph, we assessed the pairwise covariation (correlation) between all 5897 proteins quantified in all 42 biological samples using the Pearson correlation statistic (*Freedman et al., 2007*). The resulting complete, signed, weighted, and symmetric adjacency matrix was then re-weighted using 'Network Enhancement'. We implemented network enhancement in R based on microbma's translation of the original Matlab code (https://github.com/microbma/neten). Network enhancement removes noise from the graph and facilitates downstream community detection (*Wang et al., 2018*).

## Clustering the spatial proteomics network

The enhanced adjacency matrix was clustered in Python using the Leiden algorithm (*Traag et al., 2019*), a recent extension and improvement of the well-known Louvain algorithm (*Mucha et al., 2010*). The Leiden algorithm functions to optimize the partition of a graph into modules by maximizing a quality statistic. We utilized the 'Surprise' quality statistic (*Traag et al., 2015*) to identify optimal partitions of the protein covariation graph. Clustering of the network resulted in the identification of 49 modules.

## Module-level statistical inference

To evaluate modules that were changing between WT and MUT genotypes, we extended MSstatsTMT's LMM framework. In this statistical design, we were interested in the average effect of genotype on the common response of all proteins in a module. After scaling normalized protein intensity measurements, we fit each module-level subset of the data with a linear mixed-model expressing the term Protein as a random effect:

$$\log_2(Relative\ Protein\ Intensity) = \mu + Condition + Protein + \epsilon \qquad (4)$$

When fitting the module-level models, we omitted the term Mixture as the variance attributable to Mixture after normalization is negligible (*Figure 3—figure supplement 4*). The response variable is the $\log_2$-transformed scaled (sum-normalized) protein intensity measurements for all proteins in a spatial proteomics module. An overall comparison is assessed given the fitted model and a contrast vector specifying a comparison between WT and MUT groups, as described above, for protein-wise comparisons. We utilized the Bonferroni method to adjust p-values for k=49 module comparisons

and considered modules with an adjusted p-value less than 0.05 significant (n=23). For plotting, $log_2$-transformed relative protein intensity measurements were scaled into the range of 0–1 to avoid plotting negative numbers.

## Module gene set enrichment analysis

Modules were analyzed for enrichment of the WASH interactome (this paper, *Figure 1*), Retriever complex (*McNally et al., 2017*), CORUM protein complexes (*Giurgiu et al., 2019*), and subcellular predictions generated by *Geladaki et al., 2019* using the hypergeometric test with Bonferroni p-value correction for multiple comparisons. The union of all clustered and pathway proteins was used as background for the hypergeometric test. In addition to analysis of these general cellular pathways, we analyzed modules for enrichment of neuron-specific subcellular compartments – this included the presynapse (*Takamori et al., 2006*), excitatory post-synapse (*Uezu et al., 2016*), and inhibitory post-synapse (*Uezu et al., 2016*). Gene set enrichment results are found in *Supplementary file 4* and are available online at https://github.com/soderling-lab/SwipProteomics.

## Network visualization

Network graphs were visualized in Cytoscape (Version 3.7.2). Node location was manually adjusted to visualize the module more compactly. Node size was set to be proportional to the weighted degree centrality of a node in its module subgraph. Node size thus reflects node importance in the module. Visualizing co-expression or covariation networks is challenging because every node is connected to every other node (the graph is complete). To aid visualization of module topology, we removed weak edges from the graphs. A threshold for each module was set to remove the maximal number of edges before the module subgraph split into multiple components. This strategy enables visualization of the strongest paths in a network.

## Acknowledgements

We are very grateful for the human *WASHC4*$^{c.3056C>G}$ patients' contributions to this study. We would like to thank Dr. Richard Weinberg (University of North Carolina, Chapel Hill) for his expert analysis of our electron microscopy data, as well as Dr. Peter J Mucha (University of North Carolina, Chapel Hill) for his kind guidance and redirection of our clustering approach to utilize the Leiden algorithm. We also are very grateful for the statistical expertise and MSstatsTMT guidance provided by Ting Huang and Dr. Olga Vitek (Northeastern University). We thank Dr. Phillip Wilmarth (Oregon Health and Science University) for his correspondence and guidance for IRS normalization. We also thank the Duke Transgenic Core Facility for their work in generating the SWIP$^{P1019R}$ mutant mice, as well as the Duke Behavioral Core Facility, the Duke Proteomics and Metabolomics Shared Resource, the Duke Electron Microscopy Core Facility, and the Duke Light Microscopy Core Facility for their support in completing these experiments. We also greatly appreciate everyone who provided advice on this project and reviewed this manuscript, including Drs. Alicia Purkey, Shataakshi Dube, Anne West, Cagla Eroglu, and Nicole Calakos (Duke University). We generated experimental schematics using a personal academic BioRender.com license. This work was supported by a Translating Duke Health Neuroscience Initiative Grant to SHS, NIH grants (MH111684 and DA047258) to SHS, an NIH grant (MH117429) and NARSAD young investigator grant (25163) to IHK, NIH F30 fellowship funding (MH117851) and MSTP training grant support (GM007171) for JLC, and NIH F31 fellowship funding (5F31NS113738-03) to TWAB.

## Additional information

### Funding

| Funder | Grant reference number | Author |
|---|---|---|
| School of Medicine, Duke University | Translating Duke Health Neuroscience | Scott Soderling |
| National Institute of Mental Health | MH111684 | Scott Soderling |

| | | |
|---|---|---|
| National Institute on Drug Abuse | DA047258 | Scott Soderling |
| National Institute of Mental Health | MH117429 | Il Hwan Kim |
| Brain and Behavior Research Foundation | NARSAD 25163 | Il Hwan Kim |
| National Institute of Mental Health | MH117851 | Jamie L Courtland Greg Waitt |
| National Institute of General Medical Sciences | GM007171 | Greg Waitt Greg Waitt |
| National Institute of Neurological Disorders and Stroke | NS113738 | Tyler W Bradshaw |

The funders had no role in study design, data collection and interpretation, or the decision to submit the work for publication.

### Author contributions
Jamie L Courtland, Conceptualization, Data curation, Formal analysis, Funding acquisition, Investigation, Visualization, Methodology, Writing - original draft, Writing - review and editing; Tyler WA Bradshaw, Conceptualization, Data curation, Software, Formal analysis, Funding acquisition, Investigation, Visualization, Methodology, Writing - original draft, Writing - review and editing; Greg Waitt, Data curation, Methodology; Erik J Soderblom, Data curation, Formal analysis, Methodology; Tricia Ho, Data curation; Anna Rajab, Data curation, Formal analysis, Investigation, Writing - review and editing; Ricardo Vancini, Data curation, Investigation; Il Hwan Kim, Data curation, Supervision, Funding acquisition, Writing - review and editing; Scott H Soderling, Conceptualization, Resources, Supervision, Funding acquisition, Writing - original draft, Project administration, Writing - review and editing

### Author ORCIDs
Jamie L Courtland (iD) https://orcid.org/0000-0001-6846-3552
Tyler WA Bradshaw (iD) https://orcid.org/0000-0003-4480-1187
Scott H Soderling (iD) https://orcid.org/0000-0001-7808-197X

### Ethics
Human subjects: The original report of these human subjects and parental consent for data use can be found in Ropers et al., 2011.
Animal experimentation: All experiments were conducted with a protocol approved by the Duke University Institutional Animal Care and Use Committee (A2241709) in accordance with NIH guidelines.

### Decision letter and Author response
Decision letter https://doi.org/10.7554/eLife.61590.sa1
Author response https://doi.org/10.7554/eLife.61590.sa2

## Additional files

### Supplementary files
• Supplementary file 1. WASH iBioID statistical results. Raw data and statistical analyses for the in vivo BioID (iBioID) experiment (*Figure 1*). Normalized protein data were fit with a linear model to derive a model-based statistical comparison of WASH1-iBioID and solubleBioID2 control groups (*Huang et al., 2020*). 175 proteins were considered enriched in the WASH interactome by exhibiting a fold-change greater than four over the negative control with a Benjamini–Hochberg false discovery rate (FDR) less than 0.05 (*Benjamini and Hochberg, 1995*).

• Supplementary file 2. SWIP[P1019R] TMT protein-level statistical results. Raw data and protein-level statistical analyses for the spatial proteomics TMT experiment (*Figure 3*). MSstatsTMT was used to assess two types of contrasts: intra-BioFraction comparisons between Mutant and Control conditions, and overall 'Mutant-Control' comparisons.

• Supplementary file 3. Module-level results from network analysis of SWIP[P1019R] proteomics. Module membership and module-level statistical analyses for the spatial proteomics TMT experiment (*Figure 3*). Each module was fit with a linear mixed model. Contrasts between Mutant and Control groups were assessed to determine which modules had significantly altered levels between genotypes.

• Supplementary file 4. Module-level gene set enrichments of SWIP[P1019R] proteomics. Gene set enrichment statistics for the identified modules (*Figure 3*, *Supplementary file 3*). Biological pathway information, including source pathway membership, is provided.

• Supplementary file 5. Plasmid constructs. Description of plasmids used with corresponding DNA sequence files.

• Transparent reporting form

### Data availability

Further information and requests for resources and reagents should be directed to and will be fulfilled by the Lead Contact, Scott Soderling (http://scott.soderling@duke.edu). Plasmids generated by this study are available upon request from corresponding author Scott H Soderling (http://scott.soderling@duke.edu). The data and source code used in this study are available online at https://github.com/twesleyb/SwipProteomics.

The following datasets were generated:

| Author(s) | Year | Dataset title | Dataset URL | Database and Identifier |
|---|---|---|---|---|
| Courtland JL, Bradshaw TWA, Waitt G, Soderblom EJ, Ho T, Rajab A, Vancini R, Kim IH, Soderling SH | 2021 | WASH BioID | https://github.com/twesleyb/SwipProteomics/blob/master/data/BioID.zip | GitHub, BioID.zip |
| Courtland JL, Bradshaw TWA, Waitt G, Soderblom EJ, Ho T, Rajab A, Vancini R, Kim IH, Soderling SH | 2021 | SWIP TMT | https://github.com/twesleyb/SwipProteomics/blob/master/data/TMT.zip | GitHub, TMT.zip |

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
