## [Decision Letter]

**Acceptance summary:**

Your work shows how SWIP-P1019R drives endo-lysosomal expansion (a phenotype associated Lysosomal Storage Diseases) and leads to cognitive-movement impairments in mice and humans. In addition, the WASH proteome will be a valuable resource for the community.

**Decision letter after peer review:**

Thank you for submitting your article "Genetic Disruption of WASHC4 Drives Endo-lysosomal Dysfunction and Cognitive-Movement Impairments in Mice and Humans" for consideration by *eLife*. Your article has been reviewed by three peer reviewers, and the evaluation has been overseen by a Reviewing Editor and Huda Zoghbi as the Senior Editor. The reviewers have opted to remain anonymous.

The reviewers have discussed the reviews with one another and the Reviewing Editor has drafted this decision to help you prepare a revised submission.

All three reviewers agreed that Courtland et al. present and interesting and important aspect about the interactome of WASH complex in the brain. Their analysis of the effects of SWIP P1019R in endolysosomal pathway and in neurodegenerative disease is novel. All reviewers agreed that this manuscript does warrant publication in *eLife*. However, the following major concerns needed to be addressed.

A) The authors need to provide more information about their experimental design and data analysis to address the following concerns

Reviewer 1:

1) Why was WASH1 used as the fusion target for BioID? Is BioID2 alone a good control for this experiment? Had the authors considered using WASH1 with its N-terminal WASH assembly domain removed as a better control?

2) Why was SPS-MS3 not used as the mass spectrometry approach? More accurate data may have been achieved even with the application of FAIMS.

3) There are currently no plots of the data and the authors use a bespoke data analysis pipeline. It is not clear whether the experiment was successful and the results crucially depend on successful separation of the organelles. The paper is currently missing western blots to confirm separation of the organelles, upon which the analysis pipeline depends.

4) To the reader the power of using a spatial proteomics approach such as the LOPIT-DC method of Geladaki et al. seems to be a bit lost here. The authors need to be clear what added information this experimental design gives over simply just looking at changes in the total proteome. It is hard to reconcile the data with organelle re-localization without firstly showing total abundance changes between the MUT and wildtype, and secondly giving some indication that the necessary organelle resolution has been achieved (as above). This should be clarified in the text. For example, if a protein resides in more than one location, and there is a change in the abundance of this protein in one of these locations, this may manifest itself and a perceived change is localization.

5) Furthermore, if endosomal properties are changed as a result of the mutation it might result in the endosomal protein pelting at different speeds. This will complicate the statistical analysis because fractions cannot be compared like for like.

6) Was the final cytosolic (supernatant) fraction was discarded? If the abundance of the cytosolic pool of these proteins is changing then the observed results here could simply be due to that, rather than any interpreted changes at the endosome.

7) The Authors claim “In addition to highlighting the neuronal roles of WASH in CCC- and Retriever mediated endosomal sorting, our proteomics approach also identified protein modules 21 with increased abundance in SWIP P1019R mutant brain.” This is very confusing as the protein abundance change are not shown, what is known is that the more of the proteins is likely to be associated with a complex, but not the overall abundance of the complex components in the cell. These arguments persist throughout the Discussion section. If the authors take only a proportion of each fraction and this is not consistent across fractions or WT and MUT, then they do not know what the total abundance changes are.

B) Lack of important controls- the authors need to experimentally address these concerns.

Reviewer 1

1) The authors should confirm that the endosomal enriched fraction is the same in both WT and mutation experiments.

Reviewer 2

1) In their proteome data, the authors argue that 37 out of 255 modules exhibit significant differences in WT and MUT brains. These data indicates that in addition to endo-lysosomal modules, many other pathways are also affected in the MUT brains. These include endoplasmic reticulum (ER) module (M83), synaptic modules (M35 and M248) and many others that the authors did not specify…did the authors observe any defects in other organelles or cellular compartments, such as ER, mitochondria, synapse…etc?

2) Finally, in their TEM images in Figure 5, the authors argue that the electrical-dense inclusions in the cell bodies of MUT neurons are "visually" consistent with lipofusine accumulation. The authors need to use biochemical or histological methods to prove their point. This will significantly strengthen their arguments.

Reviewer 3

1) It would be beneficial if the authors could do some IPs with the WT and mutant SWIP vectors to validate the proteomic data

C) Concerns about their statistical methods and data analysis that needed to be addressed.

Reviewer 1

1) The analysis method used was chosen as previous approaches to deal with spatial proteomics data in the literature make use of well curated organelle markers. The authors claim that they did not have access to a robust set of marker lists, but other studies have used mouse neuronal cell lines (Itzhak DN et al., 2017) and also mouse ES cells (Christoforou. A et al., Nature Commun. 2016). These lists could easily have been adapted and used to visualize organelle separation using straightforward approaches such as PCA.

2) The analysis of the (spatial) proteomics data is currently not clear and there is some confusion. Firstly, edgeR was originally developed to handle RNA-sequencing data, not scRNA-sequencing data. Furthermore, RNA-sequencing data are indeed interpreted as counts and a negative binomial distribution is appropriate. This is not the case for proteomics data, as an integral under the isotopic envelope is involved in computing the intensity. Thus the analysis is not appropriate for the task. LIMMA, DEP, MSqRob, DeqMS, MSstatsTMT would all be appropriate methods.

3) The GLM framework for differential protein abundance between modules is not quite clear and the analysis is not quite correct. Instead of summarizing a module as the sum of the proteins, linear models should be fit on the data directly with a global module term and a factor for each protein. The protein factor will probably need to be encoded as a random effect. Lme4 and gam packages in R should be able to do this analysis. This section would gain a lot of clarity from some more precise descriptions.

4) From the figure it looks like the spatial proteomics data was normalized so that the max intensity in the most intense fraction is 1 – is that the case? Usually spatial proteomics data are normalized so that protein intensity sums to 1 across the fraction. I also find all the normalization and filtering for the TMT analysis quite confusing – a table might help with the desired effect in a column. Why did the authors not summarize peptides to proteins via the median or sum and then normalize so that proteins sum to 1 across the fractions?

5) There are no plots of the finally normalized spatial proteomics data to see whether the experiment was successful or not.

D) Overstated conclusions- the authors should tune down their argument or provide more data as suggested to support their conclusions.

Reviewer 1

1) The connection with human findings is a bit overstated. The findings suggest that the clinical phenotypes between humans and mice are similar. However, the mechanistic insights are only shown in mouse models. The text is slightly overstated and the mechanistic insights in humans should be toned down.

Reviewer 2

1) In their proteome data, the authors argue that 37 out of 255 modules exhibit significant differences in WT and MUT brains. These data indicates that in addition to endo-lysosomal modules, many other pathways are also affected in the MUT brains. These include endoplasmic reticulum (ER) module (M83), synaptic modules (M35 and M248) and many others that the authors did not specify. A major concern is that whether the endo-lysosomal dysfunction is the only factor that contributes to the behavioral defects? A rescue experiment can solve most of this concern. It has been shown recently the R33, a retromer chaperone, can strengthen retromer function and improves memory in a mouse model of AD (PMID: 31964406). The authors can consider testing this drug in their model.

Reviewer 3

1) The authors suggest that the WASH complex may not interact as closely with retromer as it does in other cells. This is a bold statement to make based on BioID and given the existing literature associated with the retromer-WASH axis. For example, the VPS35-D620N disruption of binding to FAM21. Could the authors expand on this? Is it possible that the retromer complex is not present in the proximity-based proteomics due to the use of WASHC1?

E) The authors need to comment on the following two suggestions.

Reviewer 3

1) Could the authors expand more on their result showing that many of the lysosomal protein interactors are enriched in the SWIP mutant condition compared to the WT when many of these proteins have been shown to be lost in neurodegenerative disease? Do the authors think that if they looked at longer aged animal they would see a drop as the lysosomes become impaired and that their model is looking at how the cells try to compensate for the endosomal dysfunction (ie early stages of neurodegeneration)?

2) It is interesting that the SWIP(P1019R) mutant mice exhibit such significant progressive motor deficits. The authors found no difference in the cleaved caspase 3 staining in the striatum but did they look at whether there was a loss of dopaminergic neurons in the substantia nigra pars compacta (or a loss of dopaminergic innervation or dopamine levels in the striatum) to account for these motor deficits? I would expect there to be a drastic loss of dopamine due to the significant motor deficits shown. Interestingly, SNCA is also present as an interactor of the WASHC1. Could the authors expand on whether they think α synuclein could therefore, also be playing a role (particularly as the authors also suggest an elevation of ER stress modulators in the SWIP mutant mice proteomics?)

---

## [Author Response]

All three reviewers agreed that Courtland et al. present and interesting and important aspect about the interactome of WASH complex in the brain. Their analysis of the effects of SWIP P1019R in endolysosomal pathway and in neurodegenerative disease is novel. All reviewers agreed that this manuscript does warrant publication in eLife. However, the following major concerns needed to be addressed.

All reviewers raised questions that we have addressed in this revision. These critiques focused on three main areas: (1) the computational analysis of our spatial proteomics experiment, (2) verification of the reproducibility and biological relevance of our proteomic data, and (3) additional studies on the molecular basis of motor dysfunction in SWIP^P1019R^ mice. In response to these concerns we have made changes to data in the following figures: Figure 1 panels G-M; Figure 2 panels B-C; Figure 3 panels A-D; Figure 4 panels A-E; Figure 5 panels F, H, J, K; Figure 6 panels H, P; Figure 1—figure supplement 1 panels A-D; Figure 3—figure supplement 1 panels A-C; Figure 3—figure supplement 2 panels A-C; Figure 3—figure supplement 3 panels A-C; Figure 3—figure supplement 4 panels A-F; Figure 3—figure supplement 5 panels A-H; Figure 4—figure supplement 1 panels A-F; Figure 4—figure supplement 2 panels A-G; Figure 6—figure supplement 2 panels A-T; Supplementary file 1; Supplementary file 2; Supplementary file 3; and Supplementary file 4.

In summary, we have re-analyzed our spatial proteomics experiment using more appropriate statistical methods for analysis of proteomics data (Figures 3-4). We have also provided additional information on our analysis workflow and data reproducibility (Figure 3—figure supplements 1-5). In addition to these changes, we performed immunohistochemical experiments to quantify the number of excitatory synapses in the motor cortex (Figure 4—figure supplement 2), as well as dopaminergic cells in the substantia nigra pars compacta and striatum (Figure 6—figure supplement 2). These experiments address key questions raised about synaptic alterations, as well as changes to the dopaminergic system, in SWIP^P1019R^ mice. We have adjusted our written descriptions to reflect these changes, as well as reviewer-suggested clarifications. We believe these improvements have made the manuscript much stronger and have answered the reviewers’ concerns.

A) The authors need to provide more information about their experimental design and data analysis to address the following concernsReviewer 1:1) Why was WASH1 used as the fusion target for BioID? Is BioID2 alone a good control for this experiment? Had the authors considered using WASH1 with its N-terminal WASH assembly domain removed as a better control?

Due to packaging limits of adeno-associated viruses (AAVs, approximate size limit = 4.7 kb), we utilized WASHC1, rather than WASHC4 (protein: SWIP) as our iBioID probe. When accounting for the size of other elements in our vector, such as the AAV inverted terminal repeat sequences (ITRs), hSyn1 promoter, biotin ligase BioID2, Woodchuck Hepatitis Virus Post-transcriptional Regulatory Element (WPRE), and poly-adenylation tail (polyA), we could only use a WASH complex expression construct that was 2.3kb or smaller. Of the WASH subunits, WASH1 fit this constraint at 1.8kb, whereas WASHC4, our protein of interest for the remainder of the paper, was above packaging limits at 5.9kb.

We have used soluble BioID2 as a negative control for other in vivo BioID experiments (Spence et al., 2019; Takano et al., 2020; Uezu et al., 2016) to identify background cellular biotinylation. While previous reports have shown that the N-terminus of WASH1 (amino acids 1-51) is required for interactions with the other four WASH subunits (Jia et al., 2010), overexpressing this mutant construct may have unintended detrimental effects and may still bind to some proteins relevant to the WASH complex. Therefore, we preferred a soluble BioID2 probe that more uniformly controls for background biotinylation throughout the cell.

2) Why was SPS-MS3 not used as the mass spectrometry approach? More accurate data may have been achieved even with the application of FAIMS.

FAIMS was used to enhance precursor ion selectivity and quantitative accuracy in MS2 TMT experiments without losing the depth of coverage that one would experience with SPS-MS3 TMT experiments (~25% at the protein level; Schweppe et al., 2019). We have clarified the resolution of MS data acquisition in the “16-plex TMT LC-MS/MS” Materials and methods section.

3) There are currently no plots of the data and the authors use a bespoke data analysis pipeline. It is not clear whether the experiment was successful and the results crucially depend on successful separation of the organelles. The paper is currently missing western blots to confirm separation of the organelles, upon which the analysis pipeline depends.

Thank you for this important suggestion. We have attached supplemental figures displaying Western blots of our spatial proteomic samples (Figure 3—figure supplements 1-3). These data highlight the pattern of protein abundance for the early endosomal marker, EEA1, and lysosomal marker, LAMP1. We have also included total protein stains for each membrane, which were used to normalize EEA1 and LAMP1 band intensity measurements. These figures demonstrate that the patterns of EEA1 and LAMP1 across all biological fractions are consistent between experiments and between WT and MUT samples, supporting adequate technical reproducibility and highlighting a lack of genotypic influence on fraction pelleting, respectively. Of note, these blots include all biological fractions (1-11) obtained in each experiment, while our mass spectrometric analyses were restricted to fractions 4-10. We believe these data demonstrate that our subcellular fractionation approach separated organelles as anticipated. In addition, we have included a PCA plot of our mass spectrometry data, which similarly demonstrates the separation of biological fractions from each brain (Figure 3—figure supplement 4C). This plot demonstrates that fractions were significantly different from one another, reflecting successful organelle separation.

4) To the reader the power of using a spatial proteomics approach such as the LOPIT-DC method of Geladaki et al. seems to be a bit lost here. The authors need to be clear what added information this experimental design gives over simply just looking at changes in the total proteome. It is hard to reconcile the data with organelle re-localization without firstly showing total abundance changes between the MUT and wildtype, and secondly giving some indication that the necessary organelle resolution has been achieved (as above). This should be clarified in the text. For example, if a protein resides in more than one location, and there is a change in the abundance of this protein in one of these locations, this may manifest itself and a perceived change is localization.

Using subcellular fractionation to separate organelles and their corresponding proteins into abundance profiles across fractions allowed us to achieve a level of resolution and depth of coverage that would not have been possible had we just compared differences in total protein abundance between genotypes from lysate. By separating organelles, we were able to group proteins that were covarying across fractions together, enabling us to identify protein modules that we would not have been able to classify in an unfractionated, bulk protein analysis. We have re-written the description of this experiment to clarify the motivation and rationale behind using this approach, and have included scaled abundance profiles for individual proteins and protein modules in Figures 3, 4, and Figure 4—figure supplement 1.

5) Furthermore, if endosomal properties are changed as a result of the mutation it might result in the endosomal protein pelting at different speeds. This will complicate the statistical analysis because fractions cannot be compared like for like.

We agree, and initially thought that this may be a confounding factor in our experiment. Western blot analysis of our subcellular brain fractions demonstrated no significant difference in endosomal EEA1 patterning between control and WASHC4 mutant conditions, suggesting that pelleting of endosomal proteins was consistent across genotypes (see Figure 3—figure supplement 2). Furthermore, this endosomal pattern was distinct and separable from lysosomal LAMP1 patterning (Figure 3—figure supplement 1), suggesting that this approach sufficiently separated organelles. Moreover, modeling of the variance attributable to Genotype, BioFraction, and Mixture for each protein in our spatial proteomics dataset demonstrates that only 1-2% of the variance for each protein is explained by Genotype. Most (~80%) variation is explained by BioFraction. After normalization, the variance attributable to Mixture is negligible (Figure 3—figure supplement 4D). We believe these data indicate that the potential influence of Genotype on subcellular fractionation is small, supporting the use of this approach to measure differences in protein/module abundance between genotypes.

6) Was the final cytosolic (supernatant) fraction was discarded? If the abundance of the cytosolic pool of these proteins is changing then the observed results here could simply be due to that, rather than any interpreted changes at the endosome.

The final cytosolic (supernatant) fraction was not included in the TMT 16-plex mass spectrometric analysis (see Figure 3—figure supplements 1-3). While the cytosolic abundance of proteins may be altered between MUT and WT, this should not detract from, or conflict with, significant differences in abundance between MUT and WT for a given subcellular fraction. For instance, if in MUT brain a protein was located in the cytosol rather than in a given subcellular organelle, such as the Golgi apparatus, this would be seen as a decrease in its abundance at the Golgi. While this does not mean its overall levels within the cell has changed, it does mean its Golgi levels have changed. We accounted for these two types of measurements (fraction-level abundance and overall abundance) in the different levels of analyses we performed. We utilized MSstatsTMT to assess both intra-BioFraction and overall “Mutant-Control” comparisons. The statistical comparisons are now illustrated in Figure 3—figure supplement 4F.

7) The Authors claim “In addition to highlighting the neuronal roles of WASH in CCC- and Retriever mediated endosomal sorting, our proteomics approach also identified protein modules 21 with increased abundance in SWIP P1019R mutant brain.” This is very confusing as the protein abundance change are not shown, what is known is that the more of the proteins is likely to be associated with a complex, but not the overall abundance of the complex components in the cell. These arguments persist throughout the Discussion section. If the authors take only a proportion of each fraction and this is not consistent across fractions or WT and MUT, then they do not know what the total abundance changes are.

We apologize our description of our module-level findings was unclear. For our experiments we performed quantitative mass spectrometry for each biological fraction obtained from WT and MUT brain to directly compare the relative levels of peptides (summarized to proteins). Our Western Blot data indicate there were no gross changes in the relative fractionation of organelles. Further, our modules are identified based on co-variation of protein levels across fractions for both WT and MUT samples. For the modules highlighted in Figures 3, 4, and Figure 4—figure supplement 1, the estimated mean abundance of all proteins in a module was significantly different between MUT and WT brain, and these data are depicted as scaled intensity plots in Figures 3C, Figure 4C, and Figure 4—figure supplement 1B,D,F. What we did not intend to convey were claims regarding changes in molecular interactions, or “likelihood of a protein associating with a complex,” as the reviewer mentioned. The only instance in which we provide evidence for a protein interacting with a complex is in our Western blot analyses of WASH subunit levels in WT vs. MUT brain (Figure 2), and co-IPs of WT and MUT constructs in cell lines (Figure 2—figure supplement 1). We have clarified the text to reflect these conclusions.

B) Lack of important controls- the authors need to experimentally address these concerns.Reviewer 1The authors should confirm that the endosomal enriched fraction is the same in both WT and mutation experiments.

As mentioned above in our response to critique #3 in Section A, we have provided a supplemental figure (Figure 3—figure supplement 2) demonstrating endosomal protein, EEA1, levels across fractions. From these data, as well as analyses of our mass spectrometric data, the pattern of endosomal proteins across fractions did not change significantly with genotype, suggesting that endosomal compartments pelleted at the same speeds from WT and MUT samples.

Reviewer 21) In their proteome data, the authors argue that 37 out of 255 modules exhibit significant differences in WT and MUT brains. These data indicates that in addition to endo-lysosomal modules, many other pathways are also affected in the MUT brains. These include endoplasmic reticulum (ER) module (M83), synaptic modules (M35 and M248) and many others that the authors did not specify…did the authors observe any defects in other organelles or cellular compartments, such as ER, mitochondria, synapse…etc?

In our re-analysis of the data, we identify a total of 49 modules. All module-level statistics are reported in Supplementary file 3. We find 23 of these modules have an overall significant difference between WT and MUT conditions (Bonferroni P-adjust < 0.05). Among the 49 modules identified in our dataset, all 10 LOPIT-DC compartments defined by Geladaki *et al.*, are represented, as determined by significant enrichment > 2-fold (hypergeometric test, Bonferroni P-adjust < 0.05). Results for module gene set enrichment are reported in Supplementary file 4. We highlight LOPIT-DC-enriched compartments with insignificant change in Figure 3—figure supplement 5. Our interpretation of the data is that the most prominent changes identified by our analysis is in endosomal-enriched M38 and lysosome-enriched M36. In line with WASHC4-mediated endosomal dysfunction, we observed another endosome module, M22 that exhibited decreased overall abundance. In addition to these endo-lysosomal changes, we also observe significant changes in M6, a module enriched for ER proteins including prominent markers of ER stress. We highlight these modules in Figure 4—figure supplement 1.

Consistent with the observed enrichment of synaptic proteins in the WASH-iBioID proteome, we observed a synaptic module, enriched for proteins identified by WASH1-iBioID (fold enrichment > 2-fold, hypergeometric test Bonferroni P-adjust < 0.05), that is significantly decreased (Figure 4—figure supplement 1E-F). As prompted by the reviewers, we followed up on this change by performing immunohistochemistry to quantify excitatory synapse density in WT and MUT brain tissue at adolescence and adulthood (see Figure 4—figure supplement 2). We focused our studies to the motor cortex, given the previous molecular changes we observed in this brain region by CC3 staining and TEM imaging. Brains were stained for excitatory pre-synaptic marker, bassoon, and post-synaptic marker, homer1, and imaged using confocal microscopy, where co-localization of the two markers is known to represent a synapse (Burrus et al., 2020; Dzyubenko et al., 2016; Stogsdill et al., 2017). We found no difference in excitatory synapse number at adolescence, but did observe a 20% decrease in excitatory synapse number in adult MUT mice compared to WT. These data are in line with the decrease in abundance of synaptic modules that we observed via spatial proteomics in adult mouse brain, given that these modules primarily contained excitatory post-synaptic proteins. We have added these findings to our description of our module analysis and believe this represents a nice example of how the module analysis helped to identify unexpected cellular changes associated with the SWIP^P1019R^ mutation.

2) Finally, in their TEM images in Figure 5, the authors argue that the electrical-dense inclusions in the cell bodies of MUT neurons are "visually" consistent with lipofusine accumulation. The authors need to use biochemical or histological methods to prove their point. This will significantly strengthen their arguments.

In the original manuscript, we used conservative language to describe what was likely lipofuscin accumulation at lysosomes, calling them “consistent with lipofuscin”. We arrived at lipofuscin as the likely substrate after comparing our results with electron-dense accumulations found in other studies of neuronal dysfunction (Gilissen and Staneva-Dobrovski, 2013; Poët et al., 2006; Ward et al., 2017; Yoshikawa et al., 2002). The reviewers make a good point that a secondary validation method would strengthen our argument that lipofuscin is accumulating at lysosomes. While some immunohistochemical techniques can detect lipofuscin, it is often difficult to probe due to its heterogenous composition, consisting of cross-linked oxidized proteins, lipids, and sugars (Moreno-García et al., 2018). We attempted to use one of these techniques, Sudan black staining, but were unsuccessful in getting the approach to work. Despite this, electron microscopy is the most sensitive method for detecting lipofuscin, and is considered the best confirmatory test for disorders associated with lipofuscin accumulation, such as neuronal ceroid lipofuscinoses (Aungaroon et al., 2016; Mukherjee et al., 2019; Simonati et al., 2014).

Therefore, as further validation, we asked Dr. Richard Weinberg, an expert in neuronal electron microscopy, to review our images in a blinded fashion to genotype or conclusion. He concluded that the electron densities were “likely lipofuscin granules, presumably representing an abnormality in the lysosomal pathway”. This independent review by an expert in the field is consistent with our interpretation of the data. We have modified the text to reflect his conclusions.

Reviewer 31) It would be beneficial if the authors could do some IPs with the WT and mutant SWIP vectors to validate the proteomic data

Thank you for this suggestion. We agree that additional immunoprecipitation experiments would be beneficial, however given that in vivo BioID identifies both direct and indirect interactors, we believe these additional experiments are outside of the scope of this experiment. We do note that the BioID experiment did identify 13 proteins previously known to associate with the WASH complex (Figure 1), supporting the overall approach. We have also shown data supporting our proteomics findings in Figure 2B and Figure 2—figure supplement 1, which both illustrate decreased abundance of WASH complex components, Strumpellin and WASH1, in cells or brain tissue expressing SWIP^P1019R^. We have included additional text in the revised version stating the limitation of the BioID dataset and that further validation is needed.

C) Concerns about their statistical methods and data analysis that needed to be addressed.Reviewer 11) The analysis method used was chosen as previous approaches to deal with spatial proteomics data in the literature make use of well curated organelle markers. The authors claim that they did not have access to a robust set of marker lists, but other studies have used mouse neuronal cell lines (Itzhak DN et al., 2017) and also mouse ES cells (Christoforou. A et al., Nature Commun. 2016). These lists could easily have been adapted and used to visualize organelle separation using straightforward approaches such as PCA.

We thank the reviewer for this suggestion. We did implement the pRloc analysis for our dataset, however, in our hands it had difficulty classifying endosomal proteins (Crook et al., 2019, 2018; Geladaki et al., 2019). Therefore, we used an alternative bottom-up approach, as exemplified by recent work by Orre et al., (Orre et al., 2019), where we first clustered the spatial proteomics network to identify groups of proteins with similar profiles, and then predicted the subcellular compartment that each module represents based on enrichment of LOPIT-DC organelle markers (Geladaki et al., 2019). As stated above, all 10 LOPIT-DC compartments defined by Geladaki *et al.*, are represented, as determined by significant enrichment > 2-fold (hypergeometric test, Bonferroni P-adjust < 0.05). Results for module gene set enrichment are reported in Supplementary file 4, and some of these organelle-enriched modules are highlighted in Figure 3—figure supplement 5.

2) The analysis of the (spatial) proteomics data is currently not clear and there is some confusion. Firstly, edgeR was originally developed to handle RNA-sequencing data, not scRNA-sequencing data. Furthermore, RNA-sequencing data are indeed interpreted as counts and a negative binomial distribution is appropriate. This is not the case for proteomics data, as an integral under the isotopic envelope is involved in computing the intensity. Thus the analysis is not appropriate for the task. LIMMA, DEP, MSqRob, DeqMS, MSstatsTMT would all be appropriate methods.

We greatly appreciate the reviewer’s insight on this and in response have completely re-analyzed our dataset in collaboration with the laboratory of Dr. Olga Vitek. We revised our statistical approach and re-analyzed our data, making use of Huang et al. (2020)’s recently published R package MSstatsTMT. MSstatsTMT uses a flexible linear mixed-model (LMM) statistical framework which we extend to re-evaluate both protein- and module-level statistical comparisons in our SWIP-TMT spatial proteomics dataset.

Our previous method for analysis of our TMT proteomics dataset can be summarized as the Sum + IRS approach, as described (Huang et al., 2020). Following protein summarization and internal reference scaling (IRS) normalization (Plubell et al., 2017), we applied edgeR (McCarthy et al., 2012) to assess differential abundance of individual proteins and protein-groups. The use of edgeR for protein-level comparisons was based on work by Plubell et al. (2017) who described IRS normalization and the use of edgeR for statistical testing in TMT MS experiments (Plubell et al., 2017). However, we failed to consider the overall adequacy of edgeR’s negative binomial (NB) model for TMT MS data.

Statistical inference in edgeR is performed for each gene or protein using a NB generalized linear model (GLM) framework. The data are assumed to be adequately described by a NB distribution parameterized by a dispersion parameter, phi. The dispersion parameter accounts for mean-variance relationships in proteomics and transcriptomics data. As signal intensity in protein MS is fundamentally related to the number of ions generated from an ionized, fragmented protein, we incorrectly inferred that TMT mass spectrometry data can be modeled as NB count data.

We address these concerns by re-analyzing the data with a more appropriate tool, MSstatsTMT. MSstatsTMT utilizes a linear-mixed model framework, and thus does not depend upon the NB assumption of edgeR’s GLM. All of these new analyses are included in the Materials and methods section and Figures 3, 4, and Figure 3—figure supplements 1-5. We also provide the statistical results in Supplementary files 2-3. Importantly, we find the results of this new analysis do not change the prior conclusions of the manuscript, but rather strengthens them.

3) The GLM framework for differential protein abundance between modules is not quite clear and the analysis is not quite correct. Instead of summarizing a module as the sum of the proteins, linear models should be fit on the data directly with a global module term and a factor for each protein. The protein factor will probably need to be encoded as a random effect. Lme4 and gam packages in R should be able to do this analysis. This section would gain a lot of clarity from some more precise descriptions.

We agree that the statistical framework developed by MSstatsTMT uses a more appropriate linear-mixed model framework (LMM) for analysis of MS data. Moreover, we realized that this LMM framework could be conceptually extended to perform analysis of protein-groups (modules). The strength of mixed-models lies in their flexible ability to model multiple sources of variation in the estimation of a response variable modeled as fixed and mixed effects. Given a map partitioning the proteome into modules of covarying proteins, we assessed the module-level difference between WT and SWIP^P1019R^ conditions. We fit the data for each module in the dataset with a linear mixed-model expressing the mixed-effect term Protein, which captures variation among a module’s constituent proteins. The response variable is the log_2_-transformed, sum-normalized protein intensity measurements for all proteins in a given module. As indicated by the reviewers, it is essential to perform protein sum-normalization to scale protein measurements, so protein profiles are comparable (e.g. (Itzhak et al., 2019)). Additional description of our LMM approach to test for module-level differential abundance is given in our updated Materials and methods section. Code to reproduce the entire analysis is available on Github (https://github.com/soderling-lab/SwipProteomics).

4) From the figure it looks like the spatial proteomics data was normalized so that the max intensity in the most intense fraction is 1 – is that the case? Usually spatial proteomics data are normalized so that protein intensity sums to 1 across the fraction. I also find all the normalization and filtering for the TMT analysis quite confusing – a table might help with the desired effect in a column. Why did the authors not summarize peptides to proteins via the median or sum and then normalize so that proteins sum to 1 across the fractions?

Our revised spatial proteomics analysis now does exactly as the reviewer suggests. After normalization and protein summarization, the protein data are sum normalized in the manner described by several of the spatial proteomics references, including Geladaki et al. (Geladaki et al., 2019). This is essential to align protein profiles for a module—scaling proteins within a module into the same intensity scale. We refer to these values as relative intensity, as they are relative to a proteins total measured intensity. The log_2_-transformed relative intensity values are then fit with equation 4 (see Materials and methods) for module-level modeling. Note that to avoid plotting negative log_2_ intensities, the values are then scaled to be in the range of 0 to 1 for plotting.

5) There are no plots of the finally normalized spatial proteomics data to see whether the experiment was successful or not.

We have now included several plots showing the overall quality of the normalized data in Figure 1—figure supplement 1 and Figure 3—figure supplements 1-4. The PCA plot in Figure 3—figure supplement 4C shows the resolution of 7 subcellular fractions. The network overview in Figure 3A shows one graphical representation of our spatial proteomics network. The PCA plot in Figure 3—figure supplement 4E shows another representation of the SWIP spatial proteome partitioned into 49 modules in PCA space.

D) Overstated conclusions- the authors should tune down their argument or provide more data as suggested to support their conclusions.Reviewer 11) The connection with human findings is a bit overstated. The findings suggest that the clinical phenotypes between humans and mice are similar. However, the mechanistic insights are only shown in mouse models. The text is slightly overstated and the mechanistic insights in humans should be toned down.

We have adjusted the wording used in our Abstract and Introduction. We believe these restated conclusions more accurately reflect our findings in mice and humans.

Reviewer 21) In their proteome data, the authors argue that 37 out of 255 modules exhibit significant differences in WT and MUT brains. These data indicates that in addition to endo-lysosomal modules, many other pathways are also affected in the MUT brains. These include endoplasmic reticulum (ER) module (M83), synaptic modules (M35 and M248) and many others that the authors did not specify. A major concern is that whether the endo-lysosomal dysfunction is the only factor that contributes to the behavioral defects? A rescue experiment can solve most of this concern. It has been shown recently the R33, a retromer chaperone, can strengthen retromer function and improves memory in a mouse model of AD (PMID: 31964406). The authors can consider testing this drug in their model.

We agree it is very possible that disruption of the endo-lysosomal pathway is not the sole driver of behavioral changes in the SWIP^P1019R^ mutant mice and have included the possibility outlined by the reviewer in the revised Discussion. Thank you for suggesting this rescue experiment, we will attempt to use this compound in future studies.

Reviewer 31) The authors suggest that the WASH complex may not interact as closely with retromer as it does in other cells. This is a bold statement to make based on BioID and given the existing literature associated with the retromer-WASH axis. For example, the VPS35-D620N disruption of binding to FAM21. Could the authors expand on this? Is it possible that the retromer complex is not present in the proximity-based proteomics due to the use of WASHC1?

Reviewer 3 makes a sound suggestion that the lack of retromer proteins in our iBioID study may be influenced by the WASH subunit we chose to tag with BioID2 and the criteria we used for determining protein enrichment (at least 4-fold higher in WASH1-BIOID samples over soluble control, identified by at least two peptides in all biological replicates, fewer than 50% missing values, and P-adjust < 0.05). Given that our module-level analysis of the TMT dataset also resulted in no clustering of retromer components with the WASH proteins, we hypothesized that there was less of an interaction between WASH and retromer in SWIP^P1019R^ mouse brain, but this assumption could certainly be incorrect. Given that there is precedence for disruption of WASH-retromer interactions in in vitro models of neurodegenerative disease (such as in cells expressing the retromer mutation VSP35^D620N^), we have edited our comments on WASH-Retromer interactions in the Discussion (McGough et al., 2014; Zavodszky et al., 2014).

E) The authors need to comment on the following two suggestions.Reviewer 31) Could the authors expand more on their result showing that many of the lysosomal protein interactors are enriched in the SWIP mutant condition compared to the WT when many of these proteins have been shown to be lost in neurodegenerative disease? Do the authors think that if they looked at longer aged animal they would see a drop as the lysosomes become impaired and that their model is looking at how the cells try to compensate for the endosomal dysfunction (ie early stages of neurodegeneration)?

This is an interesting question. While our SWIP^P1019R^ mouse model displays increased lysosomal proteins, many other neurodegenerative disorders are thought to be the result of decreased lysosomal proteins, particularly lysosomal enzymes (referenced in the Discussion). We think this increased level of lysosomal proteins may be a compensatory mechanism in SWIP^P1019R^ mutant brain, perhaps an attempt to manage mis-trafficked cargo. It is feasible that this increase could be transient, and may subside over time if the neurons are incapable of handling prolonged strain on the endo-lysosomal system. Alternatively, divergence between our mouse model and others may instead reflect two different etiologies of endo-lysosomal dysfunction that produce similar behavioral outcomes. In the latter case, either inhibition of lysosomal function or elevated lysosomal demand may ultimately lead to toxic substrate accumulation and neurodegeneration. We have added these comments to our Discussion.

2) It is interesting that the SWIP(P1019R) mutant mice exhibit such significant progressive motor deficits. The authors found no difference in the cleaved caspase 3 staining in the striatum but did they look at whether there was a loss of dopaminergic neurons in the substantia nigra pars compacta (or a loss of dopaminergic innervation or dopamine levels in the striatum) to account for these motor deficits? I would expect there to be a drastic loss of dopamine due to the significant motor deficits shown. Interestingly, SNCA is also present as an interactor of the WASHC1. Could the authors expand on whether they think α synuclein could therefore, also be playing a role (particularly as the authors also suggest an elevation of ER stress modulators in the SWIP mutant mice proteomics?)

To further investigate the molecular basis of motor dysfunction in MUT mice, we analyzed brain tissue from 8-month-old adult WT and MUT mice for the presence of tyrosine hydroxylase (TH)^+^ neurons, a marker of dopaminergic neurons. We analyzed the number of TH^+^ cells within the substantia nigra pars compacta, the primary brain structure containing dopaminergic cells that are affected in degenerative disorders such as Parkinson’s disease, as well as these cells’ projections to the striatum. In both cases, there was no significant difference between WT and MUT samples, suggesting that dopaminergic cells are not specifically affected by the SWIP^P1019R^ mutation (see Figure 6—figure supplement 2). This is in agreement with the cleaved caspase-3 staining we performed, where we did not observe a significant difference between WT and MUT samples in the striatum. These findings have been added to the Results.

Reviewer 3 makes an interesting suggestion that the SWIP^P1019R^ mutation might be connected to, or influenced by, α-synuclein-driven neurodegeneration. Α-synuclein (α-synuclein) aggregation and mis-localization are highly implicated in neurodegenerative disorders, particularly Parkinson’s disease (Baba et al., 1998; Burré et al., 2018; Spillantini et al., 1997). Moreover, mutations in the endosomal retromer complex subunits, VPS35^D620N^ and VPS35^R524W^, have also been associated with Parkinson’s disease and α-synuclein aggregation in vitro. These data support the idea that endo-lysosomal trafficking disruption may be a mechanism by which α-synuclein contributes to neurodegenerative pathology (Follett et al., 2016, 2014; Tang et al., 2015). However, in vivo analyses of animal models harboring these mutations have not consistently recapitulated these findings, so whether altered endo-lysosomal trafficking is causative or just correlated with α-synuclein aggregation remains an open question (Chen et al., 2019). For example, VPS35^D620N^ mice do not display α-synuclein aggregate accumulation in Lewy bodies (pathological structures found in neurodegenerative brain tissue), despite aggregates being found in cultured SNc neurons expressing VPS35^D620N^ (Chen et al., 2019; Tang et al., 2015).

While α-synuclein (SNCA) was highly enriched in our WASH1-BioID assay in WT brain (Figure 1) we did not find evidence that its levels were different in the SWIP^P1019R^ mutant brain (Supplementary file 2). However, we cannot conclude whether or not α-synuclein aggregation occurs in SWIP^P1019R^ mutant brain without further investigation. In addition, unlike many Parkinson’s disease models, which display specific deficits in dopaminergic cells, we did not observe any dopaminergic cell-specific changes in SWIP^P1019R^ brain (see Figure 6—figure supplement 2). This suggests that the motor pathology of SWIP^P1019R^ mice diverges from that of α-synuclein-driven Parkinson’s mouse models. The more parsimonious explanation may be that α-synuclein’s enrichment in the WASH1-BioID proteome results from its colocalization with the WASH complex at the endosome and throughout the endo-vesicular system in neurons (Boassa et al., 2013; Bodain, 1965; Burré et al., 2010; Iwai et al., 1995; Lee et al., 2005). These points have been added to the Discussion section of our manuscript.